# Genome-wide Cas9-mediated screening of essential non-coding regulatory elements via libraries of paired single-guide RNAs

Yufeng Li [1,7], Minkang Tan [1,7], Almira Akkari-Henić [1,7], Limin Zhang [1,8], Maarten Kip [1,8], Shengnan Sun [1,8], Jorian J. Sepers [1,8], Ningning Xu[1,8], Yavuz Ariyurek[2], Susan L. Kloet [2], Richard P. Davis [3], Harald Mikkers [1], Joshua J. Gruber [4], Michael P. Snyder [5] ✉, Xiao Li [6] ✉ & Baoxu Pang [1] ✉

The functions of non-coding regulatory elements (NCREs), which constitute a major fraction of the human genome, have not been systematically studied. Here we report a method involving libraries of paired single-guide RNAs targeting both ends of an NCRE as a screening system for the Cas9-mediated deletion of thousands of NCREs genome-wide to study their functions in distinct biological contexts. By using K562 and 293T cell lines and human embryonic stem cells, we show that NCREs can have redundant functions, and that many ultra-conserved elements have silencer activity and play essential roles in cell growth and in cellular responses to drugs (notably, the ultra-conserved element PAX6_Tarzan may be critical for heart development, as removing it from human embryonic stem cells led to defects in cardiomyocyte differentiation). The high-throughput screen, which is compatible with single-cell sequencing, may allow for the identification of druggable NCREs.

Protein-coding genes represent only less than 2% of the human genome and the rest is non-coding, many of which contain regulatory elements that guide the transcription of genes at the right time and within the right tissue[1,2]. On the basis of the known functions, the non-coding regulatory elements (NCREs) are categorized into various segments, such as non-coding RNAs, promoters, enhancers, silencers, insulators and so on[2–7]. Assigning and understanding the function of the non-coding regulatory genome has been a main focus of genetic research during the past decades[2]. With the efforts of individual research groups and large consortia such as ENCODE and Roadmap Epigenomics, putative biological roles have been assigned to many NCREs[8–11]. In general, different types of NCRE have unique combinations of epigenetic modifications. Therefore, using next-generation sequencing methods such as chromatin immunoprecipitation sequencing (ChIP-seq) against various histone modifications or transcription factors, or methods to profile accessible chromatins such as DNase I hypersensitivity site sequencing (DNase-seq), formaldehyde-assisted isolation of regulatory elements with sequencing (FAIRE-seq) and assay for transposase-accessible chromatin using sequencing (ATAC-seq), many NCREs have been mapped in the human genome[12,13]. For instance, insulator regions are often enriched for CTCF binding sites[14,15], while enhancer regions are usually decorated with a combination of H3K27ac and H3K4me1 (ref. 16).

[1]Department of Cell and Chemical Biology, Leiden University Medical Center, Leiden, the Netherlands. [2]Leiden Genome Technology Center, Department of Human Genetics, Leiden University Medical Center, Leiden, the Netherlands. [3]Department of Anatomy and Embryology, The Novo Nordisk Foundation Center for Stem Cell Medicine (reNEW), Leiden University Medical Center, Leiden, the Netherlands. [4]Department of Internal Medicine, University of Texas Southwestern Medical Center, Dallas, TX, USA. [5]Department of Genetics, Stanford University, Stanford, CA, USA. [6]Department of Biochemistry, The Center for RNA Science and Therapeutics, Department of Computer and Data Sciences, Case Western Reserve University, Cleveland, OH, USA. [7]These authors contributed equally: Yufeng Li, Minkang Tan, Almira Akkari-Henić. [8]These authors contributed equally: Limin Zhang, Maarten Kip, Shengnan Sun, Jorian J. Sepers, Ningning Xu. ✉e-mail: mpsnyder@stanford.edu; xiao.li9@case.edu; b.pang@lumc.nl

Simultaneously, the recent development of massively parallel reporter systems also facilitates the direct biological activity measurement and identification of enhancers and silencers[4,17,18]. However, all these methods do not measure the biological functions of these NCREs in their endogenous environment.

Recent advances in clustered regularly interspaced short palindromic repeats (CRISPR) genome editing have paved a new venue to study both the coding and non-coding parts of the genome[19–22]. CRISPR–Cas9 (CRISPR-associated protein 9) recognizes 20 bp genomic regions followed by PAM (5′-NGG-3′) and typically introduces genetic changes of a few nucleotide deletions or insertions around the targeting sites in the genome[23]. This is especially useful in studying gene functions as mutation-induced frameshifts in the coding regions would render the proteins non-functional. On the other hand, NCREs often range from 50 to 200 bp in length, with multiple transcription factor (TF) binding sites[24]. Therefore, a single CRISPR–Cas9-mediated genome editing must completely destroy the TF binding sites to abolish the function of NCRE[25,26], which is also limited by the possible CRISPR–Cas9 targeting sites throughout the human genome. Due to such limitations, single-guide RNAs tiling an entire testing region were often used to study NCREs regulating a few important genes. The modified CRISPR system that uses catalytic inactive Cas9 proteins (dCas9), linked to either a transcription activation or repression system, can also be used to study the functions of enhancers and insulators[26,27]. However, previous knowledge of NCRE functions is needed to select the proper CRISPR–dCas9-mediated activation or repression systems[27,28]. Such selection could be further complicated by recent evidence showing that certain NCREs could be bifunctional, for example, functioning as enhancers or silencers depending on the cellular context[7,18]. Similarly, the dCas9 system is also limited by the availability of single-guide RNAs at the targeting regions. Several dual-CRISPR systems have been used to study non-coding RNAs or enhancers, but their throughputs are still limited due to either the barcoding system or the design of the screening system; therefore, only limited regions were targeted[29,30]. Thus far, no systematic study of the NCREs in a genome-wide fashion, especially focusing on enhancers and silencers, has been performed.

We have developed a new dual-CRISPR screening system that could delete thousands of NCREs in a systematic and genome-wide fashion. This dual-CRISPR screening system is easy to construct and sequence, without the need for additional barcoding. As target regions of more than 200 bp in size are removed from the genome, NCREs could be studied irrespective of their specific biological functions. We designed dual-CRISPR libraries targeting 4,047 ultra-conserved elements (UCEs) in the human genome from UCNEbase[31], 1,527 in vivo-validated conserved enhancers from VISTA Enhancer Browser[32] and all 13,539 predicted enhancers in K562 cells from ENCODE[16]. Using this system, we studied the biological functions of the UCEs in the human genome and identified regions that would affect cell survival and drug response in K562 and 293T cells. We found that many UCEs have silencer activities, and many enhancers show dual functions. Furthermore, clusters of NCREs that play important functions could also be identified. Identified UCE region PAX6_Tarzan showed a key function in cardiomyocyte differentiation from human embryonic stem cells (hESCs), underscoring the feasibility of our approach to gain insight into the function of UCEs. Here we provide a versatile tool and pipeline to study the function of NCREs and other non-coding parts of the genome.

## Results

### Development of the dual-CRISPR system
NCREs could be positive transcriptional regulators (that is, enhancers), negative transcriptional regulators (that is, silencers) or genome structure regulators (that is, insulators). To systematically study NCREs irrespective of their different biological functions in their endogenous chromosome context, a versatile high-throughput dual-CRISPR screening system was developed. In this dual-CRISPR system, two different

RNA polymerase III promoters, U6 and H1, are positioned in a convergent orientation to drive the transcription of two CRISPR guide RNAs (Extended Data Fig. 1a). To test this plasmid system, two guide RNAs targeting the 5′ and 3′ ends of one DNA fragment in the genome are inserted. After transfecting the cells, two functional guide RNAs are expressed and able to delete the targeted regions from the human genome (Extended Data Fig. 1b,c).

To target thousands of potential NCREs and test their functions in different contexts, a lentiviral system and cloning strategies analogous to the method described above were developed. Briefly, after NCREs were selected, all potential single-guide RNAs that target both ends of each NCRE were designed. Guide RNAs were then selected on the basis of their targeting efficiency and off-target potential. After that, guide RNAs were paired to be able to remove the targeting regions in the presence of Cas9. Paired 20 nucleotide (nt) CRISPR RNA (crRNA) protospacer sequences were then properly orientated to follow the direction of the convergent promoters. Restriction enzyme recognition sites were placed between the paired crRNA protospacer sequences to open the plasmids to insert the guide RNA scaffolds in the subsequent steps. After the oligo pool was ordered, a two-step cloning strategy was then used to assemble the full functional dual-CRISPR library (Methods). In brief, the oligo pool that contains only the paired 20 nt crRNA sequences was cloned into the lentiviral vector. After propagation, the plasmids containing the paired 20 nt crRNA sequences were opened up by restriction enzyme digestion, followed by the second round of cloning to insert the two trans-acting CRISPR RNA (tracrRNA) scaffold sequences. The final plasmid libraries then contained paired functional guide RNAs that would remove the respective individual NCREs (Extended Data Fig. 1d). There are several advantages of the new dual-CRISPR system. Arranging guide RNAs in a convergent orientation allows direct polymerase chain reaction (PCR) steps to amplify the fragments containing the paired guide RNA sequences from the infected cells, which is compatible with high-throughput paired-end sequencing (Extended Data Fig. 1e). In addition, potential recombination bias from PCR, cloning and template switching in pooled lentiviral production can be filtered out after sequencing[33–35].

### Identification of essential UCEs
UCEs are non-coding genetic sequences that are identical among different species[36–38]. Given such stringent conservation during evolution, it is expected that many UCEs should have biological functions and be pivotal in different species[39,40]. However, recent research shows that ultra-conserved enhancers do not require perfect sequence matches to maintain their functions, when 23 of such enhancer UCEs were studied in vivo in mouse models[41]. To study the function of UCEs in a high-throughput fashion, a dual-CRISPR library was assembled on the basis of the published computation pipeline[42], which targets 4,047 UCEs in the human genome from UCNEbase[31] and 1,527 in vivo-validated conserved enhancers from VISTA Enhancer Browser[32]. In total, 63,879 dual-CRISPRs were designed, including 1,070 pairs of control guides from a previous study[29]. The dual-CRISPR library was packaged into lentivirus to infect K562 cells stably expressing Cas9 proteins (Supplementary Fig. 1a). Infected cells were first selected by puromycin and then kept in culture for 15 days to identify NCREs that would affect cell growth. Genomic DNA was isolated and PCR was performed to extract the dual CRISPRs. The abundance of different dual CRISPRs from cells on day 15 was compared to that from the initial population after puromycin selection (day 0) (Extended Data Fig. 1e). Two biological replicate experiments were performed to extract reliable hits. The replicates correlated relatively well with each other (calculated by Spearman correlation coefficients; day-0 replicates 1 and 2, 0.42; day-15 replicates 1 and 2, 0.38), indicating that the screening system is stable and reliable (Supplementary Fig. 1b). After filtering for low coverage, more than 90% of the target NCREs were matched with the paired

guide RNAs across all replicates (Supplementary Fig. 1c). The robust ranking algorithm model-based analysis of genome-wide CRISPR–Cas9 knockout (MAGeCK) was used to identify potential NCREs that affect cell growth on the basis of the screening data[43]. There were 346 UCEs and other potential NCREs depleted in the cell population that further grew for 15 days, compared with the initial population, suggesting that these UCEs are potentially essential NCREs in K562 cells (Fig. 1a and Supplementary Table 1). Unexpectedly, we also identified previously unannotated intergenic regions that affected the growth of K562 cells. Thus, the dual-CRISPR system was capable of interrogating both previously annotated and unannotated genomic regions in an unbiased fashion.

To validate the identified UCEs and the potential new NCRE, dual-CRISPR pairs targeting the potential essential NCREs were used to generate knockout (KO) clones in K562 cells (Fig. 1b–e, top). We first determined cell growth rates in these clones. When UCEs PBX3_Claudia (referred to as PBX3_Cl), FOXP1_Flora (referred to as FOXP1_Fl), PAX6_Tarzan (referred to as PAX6_Ta) and one potential NCRE (referred to as de_novo_1) were removed from K562 cells, respective clones grew significantly slower compared with the wild-type (WT) K562 cells, indicating that these UCEs regulate important genes or pathways in K562 cells (Fig. 1b–e, bottom, and Supplementary Fig. 1d). The cell growth phenotype was not due to the effects of the individual paired guide RNAs, as editing using a single-guide RNA from the pair did not result in cell growth defects but some growth advantage as shown from the Foxp1_Fl region (Supplementary Fig. 2a), which may have resulted from local editing effects of these single guides. We also compared the dual-CRISPR system with the dCas9–VP64 activating system (CRISPRa) and dCas9–KRAB repression system (CRISPRi). We designed three guide RNAs, which were used for both CRISPRa and CRISPRi systems, to tile each NCRE hit. Similar cell growth defects were observed when these regions were targeted by the CRISPRa system, which further confirmed the key function of these NCREs in cell growth (Supplementary Fig. 2b). However, CRISPRi targeting did not show any cell growth defects (Supplementary Fig. 2c), which may be due to effector-range differences between CRISPRa and CRISPRi technologies[44]. These data also indicate that the dual-CRISPR screening system could serve as an alternative method to overcome the limitations of CRISPRa and CRISPRi screening systems in studying the functions of NCREs.

We then surveyed the epigenetic modifications surrounding these NCREs. No clear combinations of epigenetic signatures were found to predict the function of these regions (Supplementary Fig. 3a), except that the de_novo_1 region sits next to a CTCF binding site (within 1 kb of the centre of the CTCF binding site) which is not directly involved in chromosome looping based on chromatin interaction analysis with paired-end tag sequencing (ChIA-PET) data (Supplementary Fig. 3b). When all identified essential UCEs and NCREs in K562 cells were compared to the rest of tested ones, enrichment of H3K27me3 and accessible chromatin regions as measured by ATAC-seq was observed (Supplementary Fig. 3c). As these UCEs are potential regulatory elements, we then tested their transcriptional regulatory activities using luciferase assays. Genomic fragments containing the UCEs were cloned into two different luciferase reporter systems for the detection of enhancer and silencer activities. Interestingly, no significant enhancer activity was observed from these regions using a commonly used minimal-promoter-driven luciferase reporter (Fig. 1f). However, PBX3_Cl and de_novo_1 showed silencer activity monitored by a PGK-promoter-driven luciferase reporter (Fig. 1g).

NCREs usually function in a tissue-specific manner. To test the functions of UCEs in a different cell type of origin, another NCRE essentiality screen was performed in 293T cells using the same dual-CRISPR library (Supplementary Fig. 4a and Table 2). As expected, only less than 10% of the essential UCEs and NCREs were shared between 293T cells and K562 cells (Supplementary Fig. 4b). We also identified tissue-specific TF motifs enriched in 293T cells (VENTX, C11orf9 and FOXO1 TF motifs) and K562 cells (ZNF187 and KLF12 TF motifs) ($P < 10^{-5}$, calculated using a hypergeometric test to determine significant TFs represented in the tissue-specific essential UCEs and NCREs compared to all tested ones), which probably reflects the tissue-specific expression or usage of different transcription factors. However, when genes that fall within 1 Mb surrounding the potential essential UCEs and NCREs were compared, more than 30% of the genes were shared

**Fig. 1 | Identification of essential UCEs. a**, MAGeCK algorithm was used to identify significant hits depleted from the cells cultured for an additional 15 days compared with the initial population. The Manhattan plot shows the distribution of all the target regions. Significant hits are above the dashed line, indicating the cut-off of MAGeCK RRA score of 0.01. Different colours represent different chromosomes. The UCEs selected for downstream analyses are indicated. The NCRE de_novo_1 is an intergenic fragment included in the library design. **b–e**, Knockout essential UCEs and NCREs from K562 cells using the dual-CRISPR system. Top: K562 cells were transfected with the respective guide RNA pairs that target PBX3_Cl (**b**), FOXP1_Fl (**c**), PAX6_Ta (**d**) and de_novo_1 (**e**). Single-cell clones with the respective NCRE deletion were selected. The blue and red arrowheads indicate the intact genomic regions and the NCRE deletions, respectively. Bottom: cell proliferation assay was performed by mixing the respective KO cell lines with cells expressing GFP at a 1:1 ratio. The changes in GFP percentage were monitored at indicated time points by fluorescence-activated cell sorting (FACS). Cells with dual-CRISPR guide RNAs targeting GFP sequences served as negative controls (Ctrl_KO). The y axis represents the relative ratio of the GFP negative cells to the positive cells. The ratio of cells in the initial mixture was set as 100% (n = 3 independent biological samples; values are shown as mean ± s.d.; PBX3_Cl_KO#1 ****P < 0.0001, PBX3_Cl_KO#2 ***P = 0.0003, FOXP1_Fl_KO#1 ****P < 0.0001, FOXP1_Fl_KO#2 ***P = 0.0004, PAX6_Ta_KO#1 ****P < 0.0001, PAX6_Ta_KO#2 **P = 0.0026, de_novo_1_KO#1 **P = 0.0016, de_novo_1_KO#2 ***P = 0.0003, calculated using two-way analysis of variance (ANOVA)). **f**, Enhancer activities were measured by luciferase assay. The respective NCREs were cloned by PCR into the enhancer reporter plasmid pGL4.23 with a minimal promoter. The empty pGL4.23 plasmid was used as the control for the baseline luciferase activities. The y axis represents the relative unit of luciferase activity compared to that of pGL4.23 empty plasmid (n = 3 independent biological samples; bars show mean ± s.d.; de_novo_1 NS P = 0.1867, FOXP1_Fl NS P = 0.7006, PBX3_Cl **P = 0.0065, PAX6_Ta NS P = 0.7339 calculated using two-tailed unpaired t-test). NS, not siginificant. **g**, Silencer activities were measured by luciferase assay. The respective NCREs were cloned by PCR into the silencer reporter plasmid pGL4.53 with a PGK promoter. The empty pGL4.53 plasmid was used as the control for the baseline luciferase activities. The y axis represents the relative unit of luciferase activity compared to that of pGL4.53 empty plasmid (n = 3 independent biological samples; bars show mean ± s.d.; de_novo_1 ***P = 0.0001, FOXP1_Fl NS P = 0.1718, PBX3_Cl ***P = 0.0007, PAX6_Ta NS P = 0.4754, calculated using two-tailed unpaired t-test). **h**, TADs identified by Hi-C surrounding the NCRE de_novo_1 are shown and the location of de_novo_1 is indicated by the vertical blue bar. Horizontal yellow and blue bars indicate distinct TADs. Transcription of nearby genes of the different KO clones was quantified by qPCR (n = 3 independent biological samples; bars show mean ± s.d.; UHRF2:Ctrl_KO NS P = 0.7381, UHRF2:de_novo_KO#1 NS P = 0.0786, UHRF2:de_novo_KO#2 NS P = 0.3125; KDM4C:Ctrl_KO NS P = 0.0777, KDM4C:de_novo_KO#1 NS P = 0.5003, KDM4C:de_novo_KO#2 NS P = 0.8219; PTPRD:Ctrl_KO NS P = 0.3794, PTPRD:de_novo_KO#1, **P = 0.0063, PTPRD:de_novo_KO#2, **P = 0.0091, calculated using two-tailed unpaired t-test). DNase I hypersensitive site (DHS). **i**, The expression level of gene PTPRD was measured by qPCR for K562 cells and K562 cells with CRISPR/dCas9-SAM activation systems targeting the PTPRD gene (PTPRD SAM) (n = 3 independent biological samples; bars show mean ± s.d.; PTPRD SAM1 **P = 0.0057, PTPRD SAM2 *P = 0.0135, calculated using two-tailed unpaired t-test). **j**, Cell proliferation assay for K562 WT and K562 cells with CRISPR/dCas9-SAM activation systems targeting the PTPRD gene (PTPRD SAM). The y axis represents the relative ratio of the GFP negative cells to the positive cells. The ratio of cells in the initial mixture was set as 100% (n = 3 independent biological samples; values are shown as mean ± s.d.; PTPRD SAM1 ***P = 0.0002, PTPRD SAM2 ****P < 0.0001, calculated using two-way ANOVA).

between the two cell lines (Supplementary Fig. 4c), suggesting that although a different set of UCEs and NCREs may function in these two cell lines, there might still be some overlapping activities shared among these different UCEs and NCREs that regulate a common set of genes that affect cell growth (Supplementary Fig. 4d,e).

## UCEs regulate a cascade of gene pathways affecting cell growth

NCREs may regulate proximal and distal genes, especially genes that are within the same topologically associating domains (TAD)[4,45]. To find out which genes might be affected by the deletion of these identified

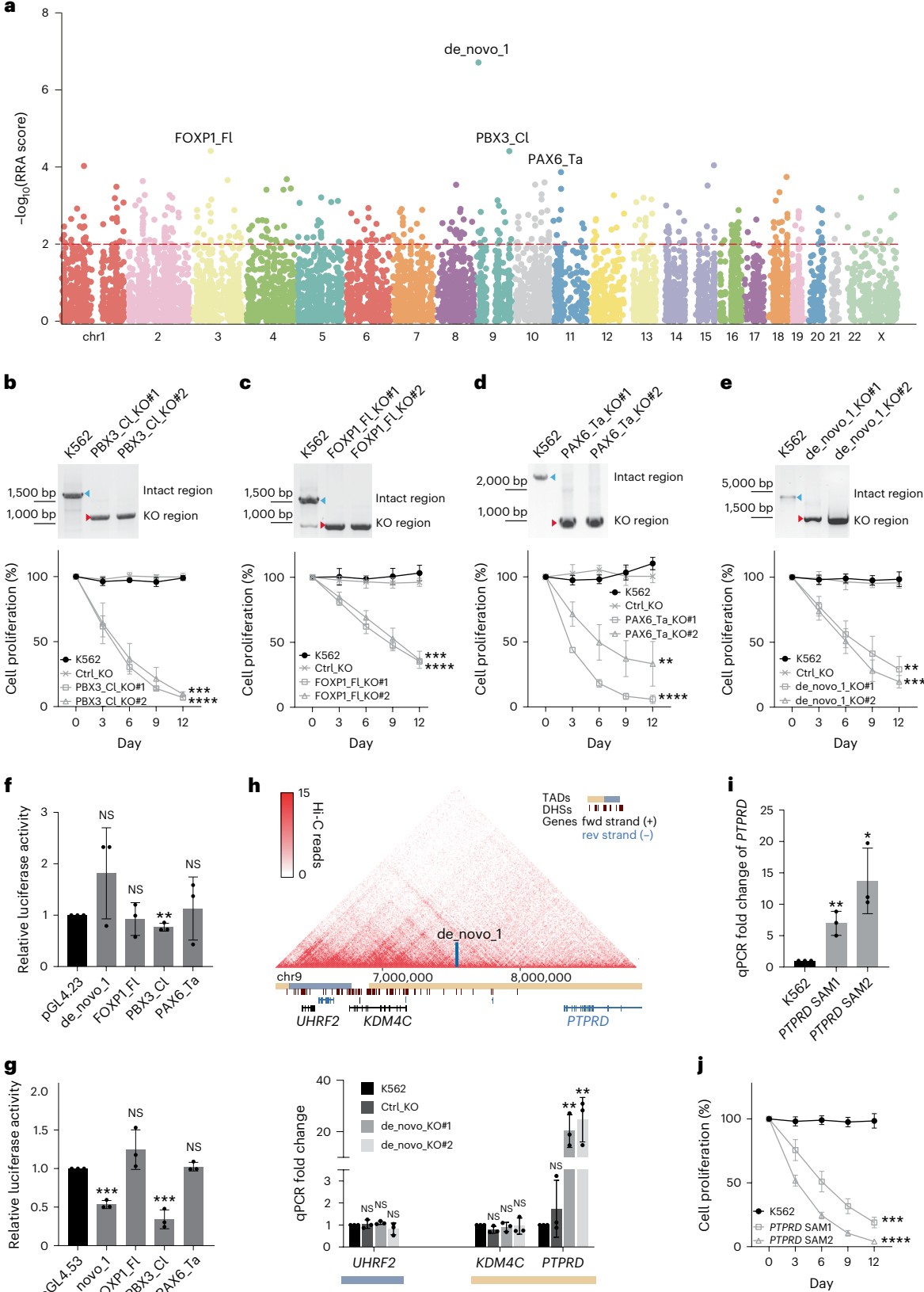

NCREs and potentially lead to the growth disadvantage, we integrated Hi-C data to identify genes that share the same TAD with the respective NCREs[46]. Transcriptional changes of the genes in the same TAD with the NCRE and in a close-by TAD were monitored between WT and NCRE-knockout K562 cells by quantitative PCR (qPCR). Transcription of *PTPRD* and *RCN1* genes, which are within the same TAD of the tested NCREs, was significantly upregulated in the knockout clones of de_novo_1 and PAX6_Ta, respectively (Fig. 1h and Extended Data Fig. 2a). In addition, these results also indicate that some of these NCREs are potential silencers, as corroborated by the luciferase assay from de_novo_1 (Fig. 1g). To survey the regulatory effects of UCE on global transcription, RNA-seq was performed on K562_PAX6_Ta KO clones. Transcriptional changes of genes that are 5 Mb surrounding the PAX6_Ta regions were compared. *RCN1* gene was significantly upregulated (Extended Data Fig. 2b, $P_{adj} = 8.39 \times 10^{-18}$, calculated using Wald test and adjusted using the Benjamini–Hochberg (BH) approach), confirming the qPCR results (Extended Data Fig. 2a). Furthermore, *CD59*, *EHF*, *ABTB2*, *CD44* and *PRR5L* genes were also significantly upregulated. Among these genes, *EHF* is a transcription factor, which may amplify the effects of PAX6_Ta regulation. *PRR5L* interacts with the mTORC2 complex and its upregulation would promote apoptosis[47], which may also contribute to the growth-delay phenotype. It should be noted that some of these deregulated genes may be indirectly regulated by PAX6_Ta. When cell cycle and apoptosis analyses were performed, these NCRE KO clones showed a slight but not significant increase in G1 cells and a mild increase in apoptotic cells within the de_novo_1 KO clones, the latter of which was significant, indicating that these clones were not in major crisis but had growth disadvantages (Supplementary Fig. 5).

To link the cell growth phenotype to the genes regulated by the identified regions, we further studied the *PTPRD* gene that was upregulated by the NCRE de_novo_1 removal. The *PTPRD* gene encodes a transmembrane receptor protein, tyrosine phosphatase, with tumour suppressor functions[48]. The CRISPR activation system was used to upregulate the *PTPRD* gene directly in K562 cells (Fig. 1i), which mimics the effects of NCRE de_novo_1 removal. A similar growth disadvantage was observed in cells with the direct upregulation of the *PTPRD* gene (Fig. 1j). The *PTPRD* gene knockout within the de_novo_1 KO clone rescued the impaired cell growth phenotype significantly (Extended Data Fig. 2c,d), suggesting that NCRE de_novo_1 may impair cell growth by regulating the *PTPRD* gene.

## UCEs function in drug resistance

Mutations or genetic changes in the NCREs could contribute to different diseases[49–52]. However, it is unknown whether NCREs are directly involved in drug responses. To test this, K562 cells infected with the dual-CRISPR library targeting UCEs were exposed to the tyrosine kinase inhibitor imatinib that targets the BCR-ABL fusion kinase in K562 cells and other related chronic myeloid leukaemias. After 15 days, surviving cells were collected and changes in the abundance of dual-CRISPRs were analysed.

After comparing the initial screening cell population, cells growing for 15 days without any drug treatment and cells growing under imatinib for 15 days, 81 NCREs possibly involved in resistance to imatinib treatment were enriched (Fig. 2a and Supplementary Table 3)[53]. First, individual K562 cell clones with UCEs ZNF503_Ophelia (ZNF503_Op) and QKI_Jonathan (QKI_Jo) deleted were made (Fig. 2b,c). These cells became more resistant to imatinib treatment than the control cells (Fig. 2d), indicating that the dual-CRISPR screens identified potential NCREs that may play a role in imatinib resistance. Luciferase assays were then performed to identify whether these regions serve as enhancers or silencers. While only QKI_Jo showed weak enhancer activity (Fig. 2e), both ZNF503_Op and QKI_Jo showed significant silencer activities when using a PGK-promoter-driven luciferase reporter (Fig. 2f). Therefore, these two UCEs may exert their functions in drug resistance via their potential silencer activities. We then tested whether any genetic variants may affect the silencer activity and identified that one single nucleotide polymorphism rs571942374 within QKI_Jo altered the silencer activity significantly (Supplementary Fig. 6), suggesting that patients bearing this single nucleotide polymorphism may respond less favourably to imatinib treatment. To identify potential genes that these regions may regulate, transcription changes of the nearby genes were studied. In the ZNF503_Op KO clones, genes *SAMD8* and *ZNF503* were upregulated (Fig. 2g), while in the QKI_Jo KO clones, gene *PACRG* was upregulated (Fig. 2h). These data also corroborate that these two NCREs may function as silencers (Fig. 2f), which contribute to the survival advantages of these cells during imatinib treatment, directly or indirectly affecting downstream genes. For instance, gene *ZNF503* encodes a transcriptional repressor that may regulate genes that drive tumour cell proliferation[54,55].

## Identification of essential enhancers using optimized dual-CRISPR systems

Enhancer regions have been extensively studied and well defined during the past few decades. However, most annotations are based on

**Fig. 2 | Drug resistance regulated by UCEs. a**, UCE–drug interactions in K562 cells. MAGeCK MLE algorithm was used to identify UCEs and NCREs involved in imatinib resistance on the basis of dual-CRISPR screens. Three different cell populations were used: the day-0 population, the day-15 imatinib treatment population and the day-15 non-treatment control population. The beta score indicates the degree of selection upon UCE or NCRE removal relative to the day-0 initial population. The *y* axis represents the beta scores of the day-15 imatinib treatment. The *x* axis shows beta scores of the day-15 non-treatment condition. The horizontal and vertical dashed lines indicate the mean ± 1 s.d. of the day-15 imatinib treatment and the day-15 non-treatment control beta score, respectively. The diagonal dashed line indicates the mean ± 1 s.d. of the differential beta scores, which can be calculated by subtracting the day-15 non-treatment control from the day-15 imatinib treatment beta score. The orange group shows UCEs or NCREs conferring imatinib resistance upon removal; the purple group shows UCEs or NCREs sensitizing cells to imatinib treatment upon removal. Selected UCEs for downstream analyses are marked. **b,c**, Knockout ZNF503_Op (**b**) and QKI_Jo (**c**) from K562 cells using the dual-CRISPR system. K562 cells were transfected with the respective guide RNA pairs that target the indicated UCEs. Single-cell clones with the respective UCE deletions were selected. The blue and red arrowheads indicate the intact genomic regions and the UCE deletions, respectively. **d**, Drug resistance conferred by ZNF503_Op and QKI_Jo knockouts. The ZNF503_Op and QKI_Jo KO cells were treated with 0.4 μM imatinib for 3 days. Cell viability was measured using CellTiter-Blue (*n* = 3 independent biological samples; bars show mean ± s.d.; ZNF503_Op_KO_#1 *P* = 0.0366, ZNF503_Op_KO_#2 *P* = 0.0415 and QKI_Jo_KO_#1 *P* = 0.0108, QKI_Jo_KO_#2 *P* = 0.0102, calculated using two-tailed unpaired *t*-test). **e**, Enhancer activities were measured by luciferase assay. The respective UCEs were cloned by PCR into the enhancer reporter plasmid pGL4.23 with a minimal promoter. The empty pGL4.23 plasmid was used as the control for the baseline luciferase activities. The *y* axis represents the relative unit of luciferase activity compared to that of pGL4.23 empty plasmids (*n* = 3 independent biological samples; bars show mean ± s.d.; ZNF503_Op [NS]*P* = 0.9705, QKI_Jo **P* = 0.0029, calculated using two-tailed unpaired *t*-test). **f**, Silencer activities were measured by luciferase assay. The respective UCEs were cloned by PCR into the silencer reporter plasmid pGL4.53 with a PGK promoter. The empty pGL4.53 plasmid was used as the control for the baseline luciferase activities. The *y* axis represents the relative unit of luciferase activity compared to that of pGL4.53 empty plasmids (*n* = 3 independent biological samples; bars show mean ± s.d.; ZNF503_Op ****P* = 0.0002 and QKI_Jo **P* = 0.0029, calculated using two-tailed unpaired *t*-test). **g,h**, Potential genes regulated by the UCEs. TADs identified by Hi-C surrounding ZNF503_Op (**g**) and QKI_Jo (**h**) are shown and the locations are indicated by vertical blue bars. Horizontal yellow and blue bars indicate distinct TADs. Transcriptions of nearby genes of the different KO clones were quantified by qPCR (*n* = 3 independent biological samples; bars show mean ± s.d.; ZNF503_Op_KO_#1: *SAMD8* **P* = 0.0049, *VDAC2* **P* = 0.007, *ZNF503* ****P* = 0.0008; ZNF503_Op_KO_#2: SAMD8 *****P* < 0.0001, *VDAC2* [NS]*P* = 0.1242, *ZNF503* ****P* = 0.0003. QKI_Jo_KO_#1: *PACRG* **P* = 0.0111, *QKI* **P* = 0.0011; QKI_Jo_KO_#2: *PACRG* **P* = 0.0126, *QKI* [NS]*P* = 0.7127, calculated using two-tailed unpaired *t*-test).

the combinations of key epigenetic modifications computationally or using ectopic enhancer reporter assays. Using the single-guide RNA CRISPR system, enhancers could be studied in their endogenous loci[25,27]. However, only limited regions could be studied due to technical issues or chosen biological readouts. There has not been a comprehensive study to examine all the potential enhancers within a defined cell line[56].

Using the dual-CRISPR system, we designed 254,203 pairs of guide RNAs targeting all 13,539 potential enhancers in K562 cells predicted by the ENCODE project[16]. Because for many enhancers it is not possible to identify properly paired guide RNAs with low off-target effects to remove the regions completely, a complementary strategy was used to design guide RNA pairs targeting the inside of the enhancer regions to aim to remove the core sequences in the centre (Fig. 3a). While

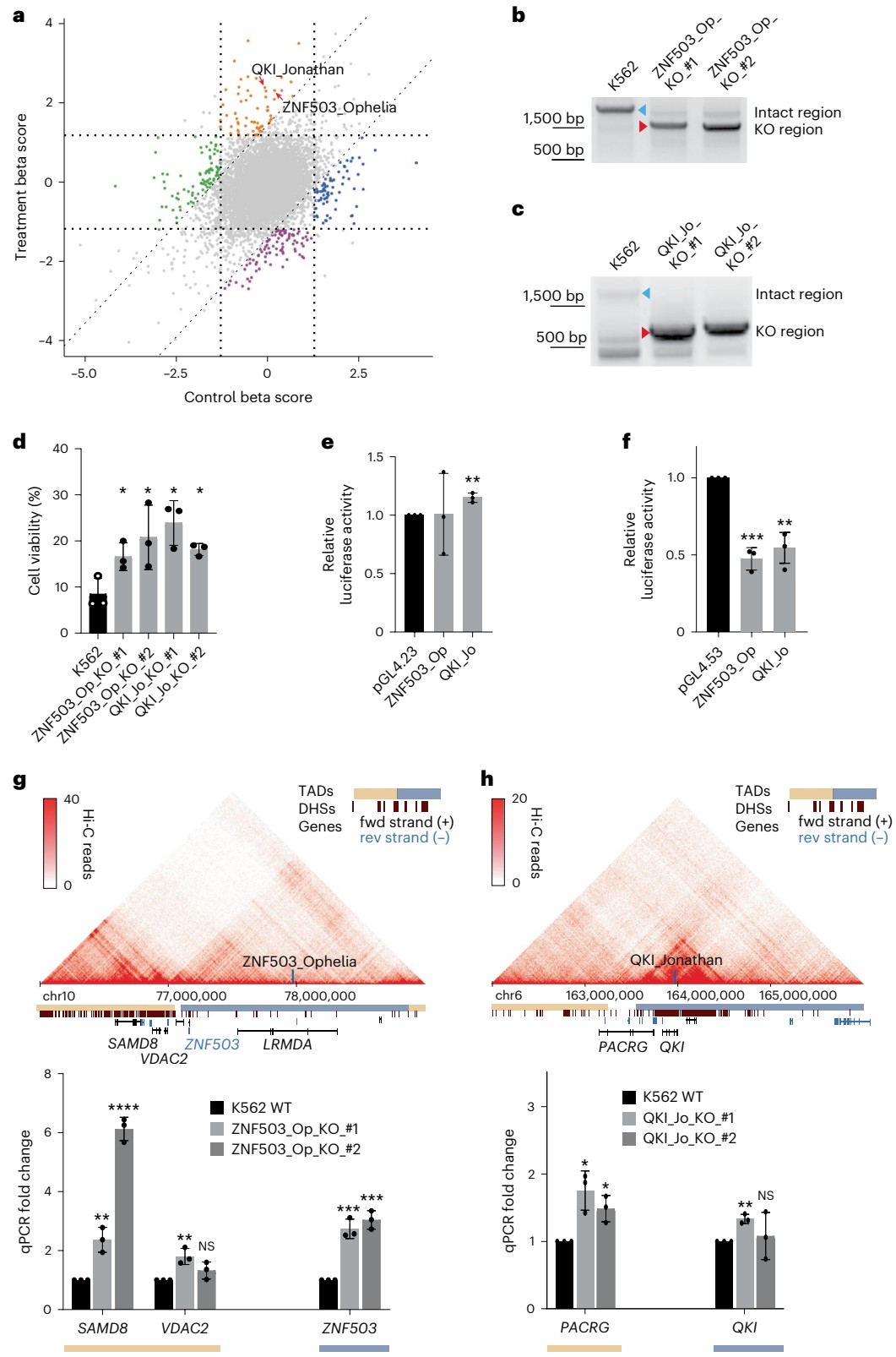

constructing these libraries, we further optimized the cloning procedures and final structure of the dual-CRISPR library system (Fig. 3b), which we named dual-CRISPR-2.0. The main improvement is that the distance between the two scaffolds was increased to 200 bp for optimal next-generation sequencing (NGS) efficiency (Fig. 3b). Using these genome-wide dual-CRISPR screening libraries, all potential enhancers in K562 cells were assayed and 1,005 enhancers were found to affect cell growth (Fig. 3c and Supplementary Table 4). We first validated the top hits by deleting these potential essential enhancer regions and observed decreased cell growth (Fig. 3d and Extended Data Fig. 3a–e), indicating that these potential enhancer regions may regulate cell growth in K562 cells. Luciferase assays were then performed to test the enhancer activities of these regions, where 4 out 6 regions led to a strong and significant luciferase gene upregulation (Fig. 3e). These regions shared a typical enhancer signature in K562 cells (Extended Data Fig. 3f–k). We then tested the potential silencer activity of these regions. Interestingly 2 out of the 6 regions also showed a significant silencer/repressor activity (Fig. 3f); at the same time, these 2 regions also exerted significant enhancer activity (Fig. 3e). These seemingly contradictory results in fact corroborate other recent studies where many NCREs were shown to have both enhancer and silencer/repressor activities in different biological contexts and when regulating different promoters[7,18,57]. In these studies, the promoters used for the enhancer and silencer reporter systems were different, similar to the luciferase assays we performed. Furthermore, another two NCRE regions did not show either enhancer or silencer activity, despite the clear growth disadvantage observed when these two NCREs were removed (Fig. 3d and Extended Data Fig. 3d). One possibility is that these NCREs do not regulate the specific promoters used in the luciferase assays.

To test how these essential enhancers regulate cell growth, we further analysed the transcription factor binding enrichment in all the essential enhancer regions and identified ZNF263, PATZ1 and KLF4 among the top enriched motifs (Fig. 3g, $E$-value $< 10^{-5}$ calculated using MEME comparing TF motifs enriched within essential enhancers using non-essential enhancers as background). It is possible that these transcription factors or their close family members that share similar binding motifs are responsible for the function of these NCREs. KLF4 is one of the key pluripotency transcription factors[58], and both PATZ1

and ZNF263 are also suggested to play a role in regulating cell proliferation[59,60]. Furthermore, all these three transcription factors have transcription repressor activity, or harbour both activator and repressor activities[59,61–63], again suggesting a complex transcriptional regulation via these essential NCREs. Therefore, studying the functions of NCREs in their endogenous loci using the dual-CRISPR system also provides complementary data compared to other widely used experimental and computational methods to define the functions of NCREs. NCREs often exert their functions by regulating different genes. When the genes potentially regulated by the essential enhancers were grouped by Activity-by-Contact (ABC) Model prediction and the functions of the genes were considered[64,65], the essential enhancer-regulated genes showed significantly lower fitness scores compared with the genes regulated by the rest of the non-essential enhancers (Fig. 3h, $P = 0.0052$ calculated using one-sided Wilcoxon rank-sum test). These data suggest that the essential enhancers may regulate key cellular genes in controlling cell growth. To gain more insights into the mechanisms of the growth defects associated with the deletion of these regions, we zoomed in on the potential genes that may be regulated by some of these NCREs. For instance, enhancer E22:23590 is located within the *BCR* region, which forms chromosomal translocation with the *Abl* gene present in many patients with chronic myelogenous leukaemia[66,67] and the K562 cell line. Although no clear regulatory functions were observed for this potential enhancer in K562, two nearby genes *Rab36* and *BCR-ABL* were differentially deregulated (Fig. 3i). Gene *Rab36* was downregulated. In contrast, *BCR-ABL* fusion gene was upregulated, suggesting a complex function of this NCRE. Downregulation of *Rab36* was shown to inhibit cell growth[68]. Also, it has been shown that upregulation of *BCR-ABL* stimulated the TGF-β pathway causing cell growth arrest[69]. Therefore, the deregulation of these genes may collectively result in the growth disadvantage induced by removing enhancer E22:23590.

## Inferring functional enhancer clusters containing redundant units

Multiple NCREs, especially enhancers, are often present in close vicinity surrounding their target genes. These enhancers often play redundant roles in regulating the same gene or genes[70,71], which complicates the assignment of the biological functions of such enhancers (Fig. 4a). So

**Fig. 3 | Identification of essential enhancers in K562 cells. a**, Outline of the design of dual-CRISPR libraries targeting all potential enhancers in K562 cells. Enhancers were predicted by ENCODE mainly on the basis of the combination of H3K4me1 and H3K27ac and other markers. Two strategies were used to design paired guide RNAs to target all the potential enhancers in K562 cells. Created with BioRender.com. **b**, Outline of the optimization of the dual-CRISPR system. To have an optimal NGS sequencing efficiency, the distance between the two scaffolds was increased from 50 bp to 200 bp by nonsense sequences. Created with BioRender.com. **c**, Essential enhancers in K562 cells were identified using dual-CRISPR screens. MAGeCK algorithm was used to identify significant hits depleted from cells cultured for an additional 15 days compared to the initial population. The Manhattan plot shows the distribution of all the target regions. Significant hits are above the dashed line indicating the MAGeCK RRA cut-off score of 0.01. Different colours represent different chromosomes. The essential enhancers selected for downstream analyses are indicated. **d**, Cell proliferation assay was performed by mixing the enhancer E22:23590 KO cell lines with cells expressing GFP. The changes in GFP percentage were monitored at indicated time points by FACS. Cells with dual-CRISPR guide RNAs targeting GFP sequences served as negative controls (Ctrl_KO). The $y$ axis represents the relative ratio of the GFP-negative cells to the positive cells. The ratio of cells in the initial mixture was set as 100% ($n = 3$ independent biological samples; values are shown as mean ± s.d.; ****$P < 0.0001$, calculated using two-way ANOVA). **e**, Enhancer activities were measured by luciferase assay. The essential K562 putative enhancers were cloned by PCR into the enhancer reporter plasmid pGL4.23 with a minimal promoter. The empty pGL4.23 plasmid was used as the control for the baseline luciferase activities. The $y$ axis represents the relative unit of luciferase activity compared to that of pGL4.23 empty plasmid ($n = 3$ independent biological samples; bars show mean ± s.d.;

E6:52372 ****$P < 0.0001$, E7:135735 $^{NS}P = 0.2465$, E8:124178 ***$P = 0.0002$, E12:123591 **$P = 0.0010$, E14:71791 ****$P < 0.0001$, E22:23590 $^{NS}P = 0.3073$, calculated using two-tailed unpaired $t$-test). **f**, Silencer activities were measured by luciferase assay. The essential K562 putative enhancers were cloned by PCR into the silencer reporter plasmid pGL4.53 with a PGK promoter. The empty pGL4.53 plasmid was used as the control for the baseline luciferase activities. The $y$ axis represents the relative unit of luciferase activity compared to that of pGL4.53 empty plasmid ($n = 3$ independent biological samples; bars show mean ± s.d.; E6:52372 $^{NS}P = 0.4655$, E7:135735 $^{NS}P = 0.4104$, E8:124178 ***$P = 0.0003$, E12:123591 $^{NS}P = 0.2297$, E14:71791 ****$P < 0.0001$, E22:23590 $^{NS}P = 0.2432$, calculated using two-tailed unpaired $t$-test). **g**, Motif enrichment analysis in essential K562 putative enhancers. The top 3 significantly enriched TF motifs are shown, ZNF263 $E$-value $= 0.00000146$, PATZ1 $E$-value $= 0.0000037$ and KLF4 $E$-value $= 0.00000452$, calculated using one-tailed Fisher's exact test. The $E$-value is the $P_{adj}$ multiplied by the number of motifs tested. **h**, Copy number bias-corrected essentiality scores of genes ($n = 877$) regulated by essential K562 enhancers compared to that of genes ($n = 10,197$) regulated by non-essential K562 enhancers. The $y$ axis represents the cell fitness scores of genes from the GeCKOv2 library loss-of-function screens in K562 cells. **$P = 0.0052$, calculated using one-sided Wilcoxon rank-sum test. **i**, TADs identified by Hi-C surrounding the E22:23590 region are shown and the location of E22:23590 is indicated by the vertical blue bar. Horizontal yellow and blue bars indicate distinct TADs. Transcription of nearby genes of the different KO clones was quantified by qPCR ($n = 3$ independent biological samples; bars show mean ± s.d.; *RAB36*: E22:23590_KO#1 (KO#1) and E22:23590_KO#2 (KO#2) ****$P < 0.0001$, *BCR-ABL*: E22:23590_KO#1 ***$P = 0.0009$, E22:23590_KO#2 **$P = 0.0021$, calculated using two-tailed unpaired $t$-test).

far, most other groups and we have focused on identifying single NCREs or enhancers with strong phenotypes, such as on cell growth. It has been challenging to study clusters of enhancers in a high-throughput and genome-wide fashion. We hypothesized that for a cluster of enhancers with redundant activity, some of them would individually show some but not significant screening enrichment scores. However, when these enhancers are considered as a cluster, their combined biological effects would stand out. On the basis of this, a new analysis was performed on

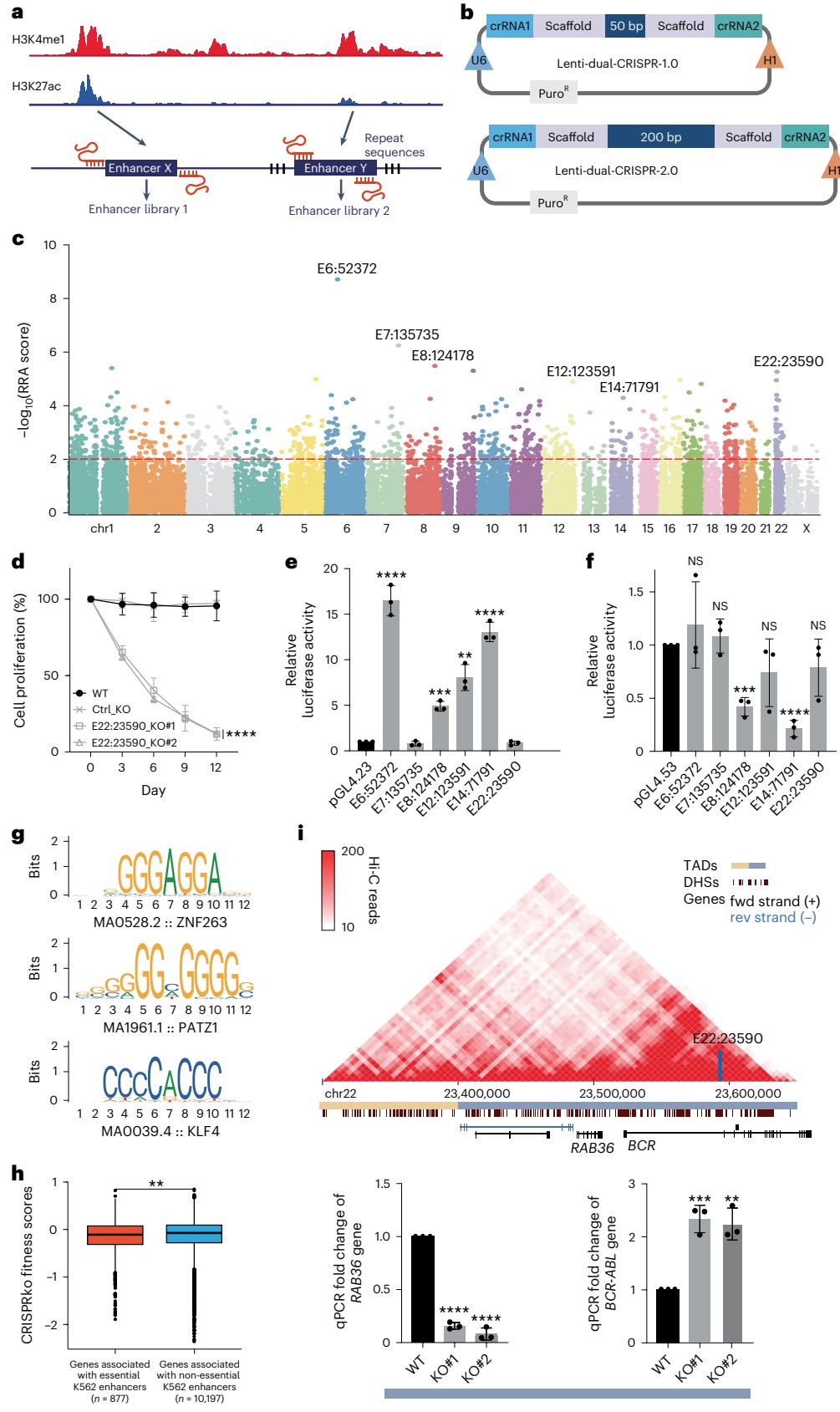

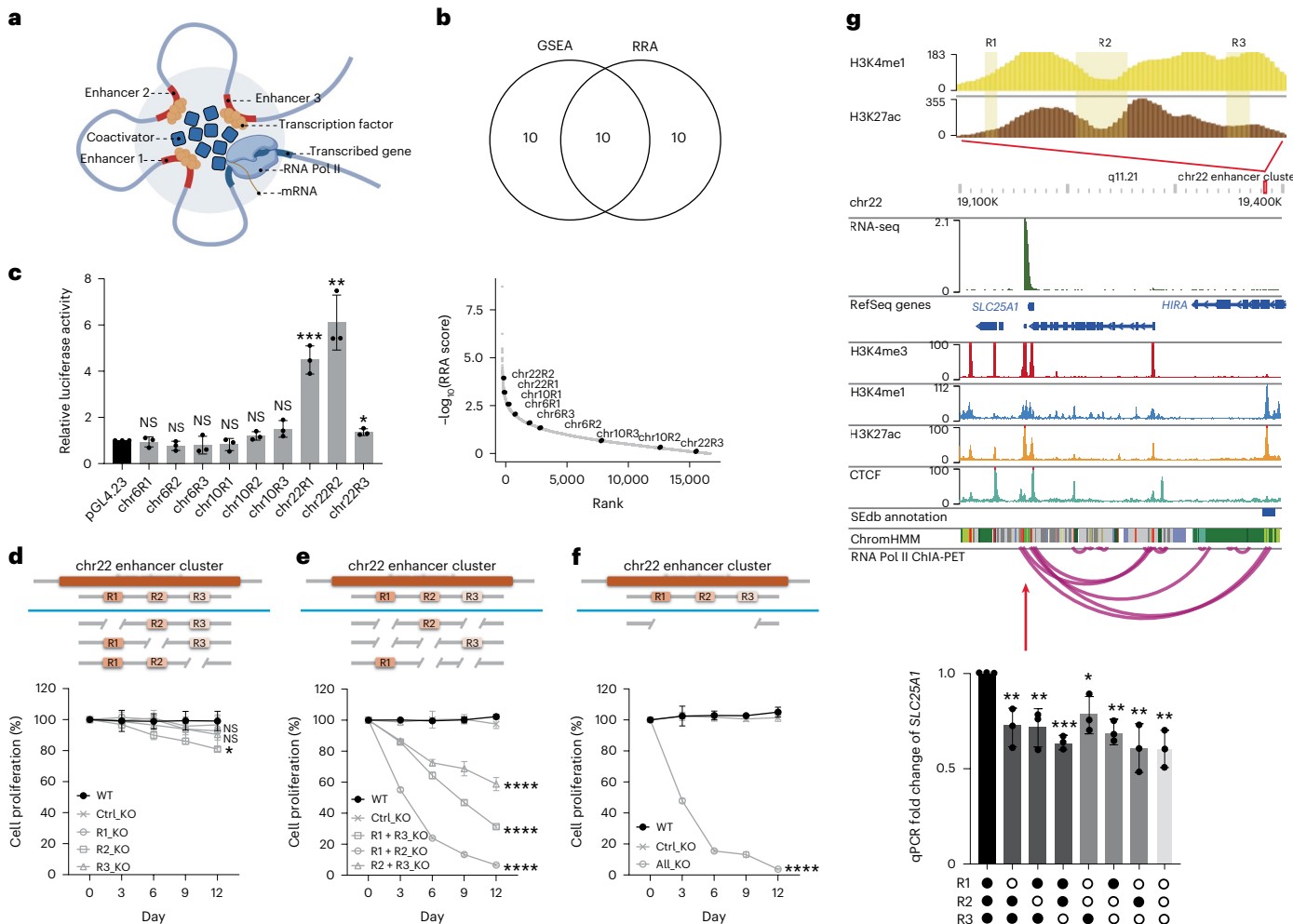

**Fig. 4 | Identifying essential cluster of enhancers with redundant functions.**
**a**, Illustration of a cluster of enhancers regulating gene transcription. Created with BioRender.com. **b**, Top: Venn diagram indicating the overlapping clusters identified by GSEA and RRA. Bottom: plot showing the ranking of the individual enhancers from the top 3 clusters in the original screening analysis. The y axis represents the MAGeCK RRA score. The x axis represents the ranking of the individual enhancers based on the RRA score. R1, R2 and R3 represent the three individual potential enhancers on the respective chromosomes (chr) targeted by the dual-CRISPR libraries **c**, Enhancer activities were measured by luciferase assay. Individual putative enhancer elements from the top 3 enhancer clusters were cloned by PCR into the enhancer reporter plasmid pGL4.23 with a minimal promoter. The empty pGL4.23 plasmid was used as the control for the baseline luciferase activities. The y axis represents the relative unit of luciferase activity compared to that of pGL4.23 empty plasmid (n = 3 independent biological samples; bars show mean ± s.d.; chr6R1 $^{NS}P$ = 0.6800, chr6R2 $^{NS}P$ = 0.1147, chr6R3 $^{NS}P$ = 0.4200, chr10R1 $^{NS}P$ = 0.3176, chr10R2 $^{NS}P$ = 0.1312, chr10R3 $^{NS}P$ = 0.0721, chr22R1 $^{***}P$ = 0.0006, chr22R2 $^{**}P$ = 0.0017, chr22R3 $^*P$ = 0.0122, calculated using two-tailed unpaired t-test). **d**–**f**, Top: illustration of one-enhancer (**d**),

two-enhancer (**e**) and three-enhancer (**f**) removal from the cluster. Bottom: cell proliferation assay was performed by mixing the KO cell lines with cells expressing GFP at a 1:1 ratio. The changes in GFP percentage were monitored at indicated time points by FACS. Cells with dual-CRISPR guide RNAs targeting GFP sequences served as negative controls (Ctrl_KO). The y axis represents the relative ratio of the GFP-negative cells to the positive cells. The ratio of cells in the initial mixture was set as 100%. R1, R2 and R3 represent the three individual potential enhancers on chr22 targeted by the dual-CRISPR libraries (n = 3 independent biological samples; values are shown as mean ± s.d.; R1_KO $^{NS}P$ = 0.7047, R3_KO $^{NS}P$ = 0.3443, R2_KO $^*P$ = 0.0348 and R1 + R3_KO, R1 + R2_KO, R2 + R3_KO, All_KO $^{****}P$ < 0.0001, calculated using two-way ANOVA). **g**, Top: epigenetic signatures surrounding the chr22 enhancer cluster. The RNA Pol II ChIA-PET loops are shown. The red arrow indicates the location of gene *SLC25A1*. Bottom: transcription of *SLC25A1* in different enhancer KO clones was quantified by qPCR (n = 3 independent biological samples; values are shown as mean ± s.d. for each bar; R1_KO $^{**}P$ = 0.0066, R2_KO $^{**}P$ = 0.0075, R3_KO $^{***}P$ = 0.0001, R1 + R2_KO $^*P$ = 0.0195, R2 + R3_KO $^{**}P$ = 0.0015, R1 + R3_KO $^{**}P$ = 0.0062 and All_KO $^{**}P$ = 0.0025, calculated using two-tailed unpaired t-test).

the enhancer essentiality screen (Fig. 3), where the targeted enhancers were further grouped according to previously defined clusters on the basis of a distance metric[72]. To reliably capture the essential enhancer clusters, two computational models, gene set enrichment analysis (GSEA) and MAGeCK robust rank aggregation (RRA), were applied[43,73], and P values obtained by the two methods were corrected by the BH procedure for false discovery rate (FDR) control (Methods). The shared top clusters computed by both methods were then used for the downstream study (Venn diagram in Fig. 4b; see also Supplementary Table 5). The individual enhancers from the top 3 enhancer clusters were further

studied (Fig. 4b, RRA score distribution of these individual regions from the original essentiality screen). Luciferase assays were performed to determine the enhancer activity of these individual regions, and only enhancers from chromosome 6 showed strong enhancer activity using the pGL4.23 reporter (Fig. 4c).

To test the functions of these individual enhancers and the clusters in their endogenous loci: first, only one enhancer was removed from the clusters containing three enhancers using dual-CRISPR targeting. Only moderate growth disadvantage was observed (Fig. 4d and Supplementary Fig. 7a,d). Then, two enhancers were removed, which resulted

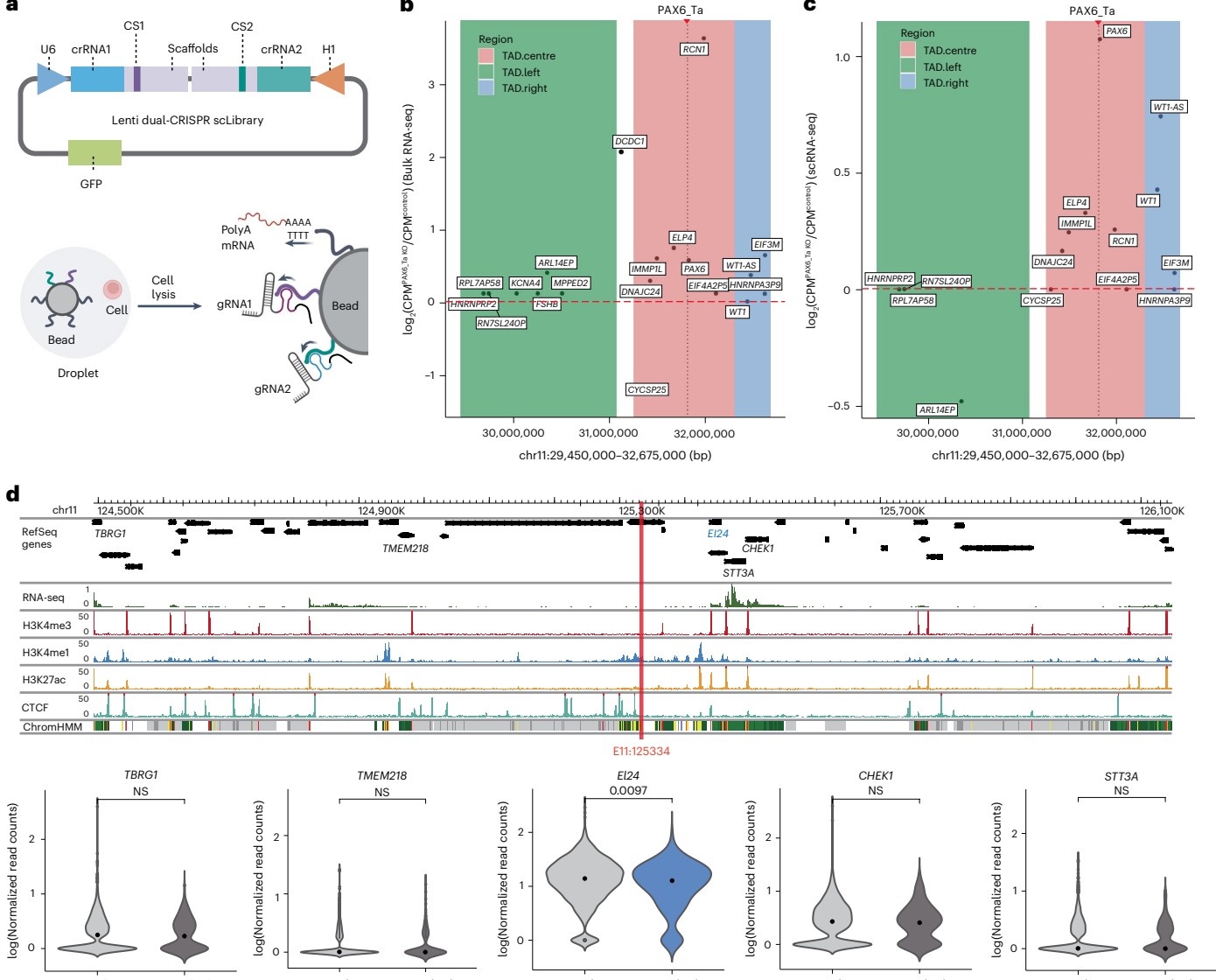

**Fig. 5 | Single-cell RNA-seq coupled with the dual-CRISPR screening system.**
**a**, Illustration of adapting the dual-CRISPR screening system with scRNA-seq.
Top: two distinct capture sequences (CS1 and CS2) were inserted into the stem
loops of the guide RNA scaffolds. Bottom: the guide RNAs could then be captured
together with the mRNA within individual single cells. Created with BioRender.
com. **b**, Transcriptional changes of genes surrounding UCE PAX6_Ta measured by
bulk RNA-seq. The *y* axis shows the log fold change of genes between the PAX6_Ta
KO clone and control K562 cells. The *x* axis indicates the genomic coordinate.
The dashed vertical line indicates the location of PAX6_Ta. Distinct colours
represent different TADs. **c**, Transcriptional changes of genes surrounding

PAX6_Ta measured by scRNA-seq. The *y* axis shows the log fold change of the
captured genes between single cells with the guide RNA pair targeting PAX6_Ta
and control K562 single cells. The *x* axis indicates the genomic coordinate. The
dashed vertical line indicates the location of PAX6_Ta. Distinct colours represent
different TADs. **d**, Top: epigenetic signatures surrounding enhancer E11:125334
as indicated by the red vertical line. Bottom: differential gene expression testing
results. Violin plots show the normalized expression levels of candidate genes
in perturbed (188 cells) and control (4,282 cells) groups. The gene *EI24* was
significantly downregulated (*P* = 0.0097, calculated using a MAST-fitted model).

in three possible combinations of two-enhancer deletion clones. In
general, the two-enhancer deletion clones had stronger growth sup-
pression compared with the deletion of any single enhancer (Fig. 4d,e
and Supplementary Fig. 7a,b,d,e), suggesting that some redundant
functions are shared among these enhancer archipelagos as described
before[53]. Indeed, the enhancer RNA (eRNA) expression (measurement
of enhancer activity[74]) of the nearby enhancers did not change or even
increased when only one enhancer was deleted, while eRNA expres-
sion of all the enhancers decreased when two enhancers of the chr22
enhancer cluster were removed (Supplementary Fig. 7g–i). When all
three enhancers were removed, proliferation was strongly impaired
(Fig. 4f and Supplementary Fig. 7c,f). For the chr22 enhancer cluster,

the cell growth defect was probably caused by regulating the *SLC25A1*
gene (Fig. 4g). These data indicate that the dual-CRISPR screening
system is also capable of capturing clusters of NCREs with redundant
biological functions in a genome-wide and systematic study.

## Studying the functions of NCREs using the dual-CRISPR system at the single-cell level

Coupling the NCREs with their respective regulated genes has
been challenging. Assays such as ChIA-PET or Hi-C that probe the
three-dimensional chromatin interactions could provide indica-
tions for genome-wide gene–NCRE physical interactions, although
no direct measurement of the transcriptional regulation could be

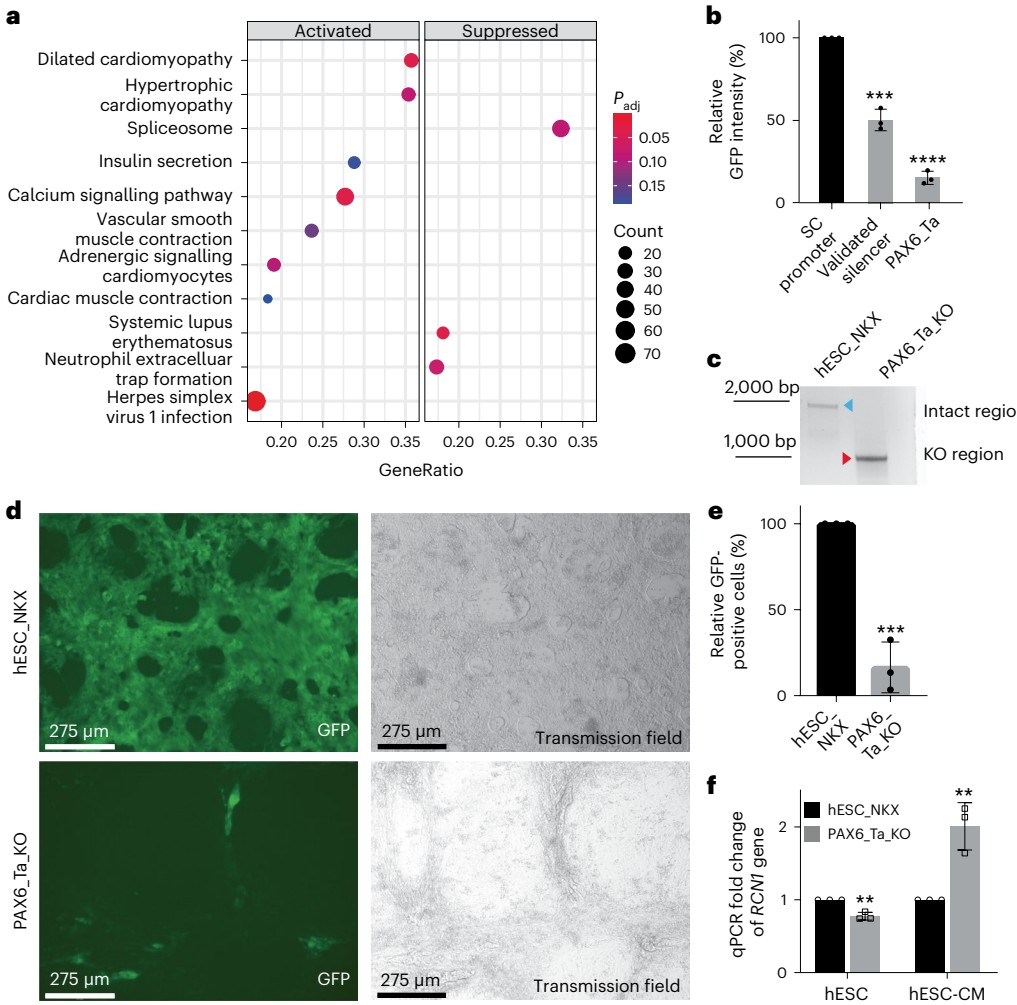

**Fig. 6 | UCE PAX6_Ta regulates cardiomyocyte differentiation in hESCs.**
**a**, KEGG pathway enrichment analysis of all genes differentially regulated in UCE PAX6_Ta KO cells. The *y* axis represents the significantly enriched KEGG pathways, and the *x* axis represents their corresponding gene ratios, that is, the ratio between the number of unique genes in a specific pathway and the number of unique genes mapped to all of these pathways. Left: significantly activated pathways. Right: significantly suppressed pathways. The dot size denotes the number of genes mapped in a pathway, and the colour indicates the significance level of the enrichment. **b**, Silencer activities were measured by the repressive ability on the SC promoter driving the expression of GFP. A previously published silencer served as the positive control (validated silencer). The empty vector (SC promoter) was used as the control to normalize the respective repressive activities (*n* = 3 independent biological samples; values are shown as mean ± s.d. for each bar; validated silencer ***P = 0.0002 and PAX6_Ta ****P < 0.0001, calculated using two-tailed unpaired *t*-test). **c**, Knockout PAX6_Ta in *NKX2-5^eGFP/w* hESCs. PCR results show the removal of PAX6_Ta in *NKX2-5^eGFP/w*

hESC PAX6_Ta KO bulk cells (PAX6_Ta_KO) compared to the *NKX2-5^eGFP/w* hESCs (hESC_NKX). The blue and red arrowheads indicate the intact genomic regions and the NCRE deletions, respectively. **d**, The removal of PAX6_Ta caused a defect in cardiomyocyte differentiation, as measured by NKX2-5/GFP expression. Top: hESC_NKX after differentiation. Bottom: PAX6_Ta_KO after differentiation. Scale bar, 275 μm. **e**, Quantification of cardiomyocyte differentiation as shown in **d** using FACS. The cardiomyocyte differentiation efficiency measured by GFP in hESC_NKX was set to 100% (*n* = 3 independent biological samples; values are shown as mean ± s.d.; PAX6_Ta_KO ***P = 0.0006, calculated using two-tailed unpaired *t*-test). **f**, The transcription of gene *RCN1* was quantified by qPCR and compared between hESC_NKX and PAX6_Ta_KO cells, before cardiomyocyte differentiation (hESC) and after cardiomyocyte differentiation (hESC-CM) (*n* = 3 independent biological samples; values are shown as mean ± s.d.; hESC **P = 0.001789 and hESC-CM **P = 0.005731, calculated using two-tailed unpaired *t*-test).

made. Single-cell (sc) RNA-seq combined with CRISPR perturbation (Perturb-seq) has been used to couple genetic perturbation, either targeting genes or enhancers, with broader transcriptome changes in single cells[75,76]. However, these studies still relied on single-guide-RNA-mediated perturbation, and previous knowledge of the potential function of the NCREs is required to choose the right dCas9-repressor system. Removing a larger fragment of the genome while monitoring the transcriptome changes could provide a useful tool to study both the coding and non-coding part (irrespective of previous knowledge of the NCRE) of the genome at the single-cell level. To explore this option, the dual-CRISPR screening system was

modified to capture the transcriptome changes of single cells upon removal of NCREs. Two distinct capture sequences were added to the dual scaffolds of guide RNAs[77], allowing direct capture of the two guide RNAs and mRNAs within a single cell (Fig. 5a). As a pilot test, 82 pairs of guide RNAs targeting 42 different NCREs from the top list of the genome-wide dual-CRISPR screens were selected and cloned into the sc-dual-CRISPR system that contains GFP as a marker. K562/Cas9 cells were infected with lentivirus containing the sc-dual-CRISPR library at a low multiplicity of infection (MOI) of 0.2 to make sure each cell only contained one pair of guide RNAs. Then, GFP-positive cells were sorted and processed for scRNA-seq (Supplementary Fig. 8a–g).

Two single-cell sequencing libraries, either aiming to target up to 10,000 cells per chip or 30,000 cells per chip, were made. After perturbation index assignment and filtering multiplet cell data, 1,199 and 3,271 usable single cells were collected from the two batches, respectively. Therefore, it is possible to retrieve more usable single cells per perturbation with a higher number of input cells (Supplementary Fig. 8h,i), when pairs of guide RNAs were used as additional cell barcodes. The two experiments were combined with a mean of 110 cells per NCRE target for downstream analysis (Supplementary Table 6). A median of 4,454 genes per cell and a median of 21,446 unique molecular identifiers of mRNA molecules per cell were observed (Supplementary Fig. 8j). We observed a similar trend of gene deregulation around the TADs of PAX6_Ta between bulk RNA-seq data (Fig. 5b) and scRNA-seq data (Fig. 5c; 127 cells) when the PAX6_Ta region was removed. To identify significantly deregulated genes targeted by the tested NCREs around their vicinity, differential expression analyses were performed using MAST[78], a method tailored to fit a two-part generalized linear model for zero-inflated and bimodal-distributed single-cell gene expression data. Due to the detection limit on lowly expressed transcripts[75], only genes with acceptable mean normalized expression levels and detected in a sufficient percentage of cells were used for differential expression test using MAST (Methods). Transcription changes in the positive control genes (*RPL18A*, *RPL13*, *RPL21* and *RPL8*), where the dual CRISPRs targeted the promoters of these genes, confirmed the feasibility of the sc-dual-CRISPR system to capture the transcription changes of the NCRE-target genes (Supplementary Fig. 8k). There were 22 significant NCRE–gene pairs identified using this method (Fig. 5d, and Supplementary Fig. 8l–n and Table 7). For example, a potential enhancer E11:125334 that was identified to play a role in imatinib resistance was found to downregulate gene *El24* upon dual-CRISPR editing (Fig. 5d). Gene *El24* is indeed associated with resistance to many chemotherapeutic drugs including imatinib[79,80], suggesting that the enhancer E11:125334 may regulate the *El24* gene to exert imatinib resistance. In addition, a potential enhancer E16:30551, which was identified to affect cell growth, was found to downregulate genes *PPP4C* and *BOLA2B* and simultaneously upregulate gene *ZNF689* (Supplementary Fig. 8l). In this case, the *PPP4C* gene encodes the protein phosphatase 4 catalytic subunit, and *PPP4C*-deficient thymocytes showed decreased proliferation and enhanced apoptosis in vivo[81]. The gene *BOLA2B* was reported to associate with human hepatocellular carcinoma progression and the *BOLA2B*-knockout mouse model showed a slow tumourigenicity[82]. Deregulation of these genes together upon E16:30551 deletion may lead to cell growth disadvantage. These data show that it is possible to combine dual-CRISPR-mediated NCRE deletion with scRNA-seq to identify potential genes regulated by NCREs.

## UCE in the PAX6 region affects hESC cardiomyocyte differentiation

Assigning specific biological functions to NCREs is still challenging. UCEs are especially interesting as these regions are conserved among different species, and for a long time, these regions have been speculated to play fundamental functions in evolution. However, recent research indicated that mice with a few UCEs removed showed no abnormalities, and sequence conservation did not play an important role in the enhancer function within UCEs in mice[41,83]. Yet, only a small percentage of the UCEs have been studied, and only the ones with enhancer function were considered. It is possible that other molecular functions may be associated with these UCEs in different developmental stages. It is therefore important to first define the potential functions of UCEs, which could be facilitated by systematically studying the functions of UCEs in different biological contexts using the dual-CRISPR system.

Global transcriptome analyses showed that in K562_PAX6_Ta KO clones, several heart-related pathways were deregulated (Fig. 6a), suggesting that this UCE may play a role in cardiomyocyte function. Although no enhancer or silencer activity of PAX6_Ta was shown in

canonical luciferase assays in K562 cells (Fig. 1f,g), both qPCR and RNA-seq data measuring the surrounding gene expression of this UCE suggest that it might be a silencer (Extended Data Fig. 2). As NCREs often function in a tissue- and promoter-specific manner, we then constructed a silencer reporter driven by a supercore (SC) promoter for expressing GFP[84], which measures silencer activity on a promoter different from the PGK promoter used in previous luciferase assays. A reduction in GFP fluorescence was observed, similar to that in a previously published silencer[4], indicating that UCE PAX6_Ta may be a potential silencer in this context in K562 cells (Fig. 6b and Supplementary Fig. 9a).

To further investigate the potential function of PAX6_Ta in cardiac development, *NKX2-5eGFP/w* hESCs were transfected with dual-CRISPR–Cas9 targeting the PAX6_Ta region. Efficient genome editing and complete knockout were observed in the PAX6_Ta_KO_bulk hESCs (Fig. 6c and Supplementary Fig. 9b). Then, both the parental *NKX2-5eGFP/w* hESCs and the PAX6_Ta_KO_bulk hESCs were subjected to cardiomyocyte differentiation. The parental *NKX2-5eGFP/w* hESCs became GFP+ upon expression of the cardiac marker gene *NKX2-5* once they committed into cardiomyocytes successfully[85]. In contrast, a significant reduction of GFP+ cells was observed in PAX6_Ta_KO_bulk hESCs that underwent cardiac differentiation (Fig. 6d,e). Furthermore, the small percentage of GFP+ cardiomyocytes formed from the PAX6_Ta_KO_bulk hESCs showed irregular beat patterns compared with *NKX2-5eGFP/w* hESCs (Supplementary Videos 1–4). Such irregular beat patterns were not observed in AAVS1_KO_bulk hESCs (Supplementary Fig. 9c–e), which served as the dual-CRISPR–Cas9-editing control. When the transcription changes of the genes surrounding UCE PAX6_Ta were measured, genes *RCN1* and *PAX6* were significantly upregulated in PAX6_Ta_KO_bulk hESCs compared with the parental cells after cardiomyocyte differentiation (hESC-CM) (Fig. 6f and Supplementary Fig. 9f,g). In contrast, these two genes were moderately but significantly downregulated before differentiation (Fig. 6f and Supplementary Fig. 9f,g). Furthermore, a series of genes contributing to cardiomyocyte functionality were also monitored and only *CACNG8* was deregulated (Supplementary Fig. 9f,g). RCN1 is a CREC family member calcium-binding protein[86]. As calcium homoeostasis is essential in cardiomyocytes[87], deregulation of *RCN1* may affect cardiomyocyte differentiation. PAX6 is a transcription factor that is key for neuronal differentiation[88] and may cause unwanted gene expression during cardiomyocyte differentiation. It is possible that the PAX6_Ta region keeps these genes in check during cardiomyocyte differentiation and the loss of PAX6_Ta caused the deregulation of these genes, leading to failure during cardiomyocyte differentiation. These data suggest that PAX6_Ta may be an important NCRE in cardiomyocyte differentiation via the regulation of surrounding genes and possibly other downstream genes that are key to heart development and physiology.

## Discussion

NCREs are essential in regulating the transcription of genes and coordinating genomic information to form complex organisms. NCREs may function in activating, repressing or insulating transcription activities. In this study, we developed an adaptable dual-CRISPR system that could be used to study NCREs irrespective of their biological functions in a genome-wide fashion. The integrated dual-CRISPR libraries could be amplified and sequenced using routine methods, without the need for custom sequencing primers or barcodes to infer pairs of guide RNAs. We constructed several dual-CRISPR libraries targeting 4,047 UCEs, 1,527 in vivo-validated conserved enhancers and all potential 13,539 enhancers predicted in K562 cells. Using these libraries, the functions of NCREs in cell survival and drug responses were studied in K562 cells and 293T cells.

Our results show that many UCEs might play important roles in cell survival or resistance to imatinib treatment. In previous in vivo studies using mouse models, genetic editing of a few selected UCEs did not show strong phenotypes, although UCEs are expected to be

functionally essential due to their high evolutionary conservation. It is possible that some UCEs may only function in defined tissues or developmental stages. Results from our research and future investigations may help to narrow down the potential tissues for in vivo studies to better understand these evolutionary puzzles. Intriguingly, many of the UCEs showed silencer activities based on luciferase assays or CRISPR editing (Figs. 1, 2 and 6). Compared with enhancers or insulators, silencers are less well-studied and their roles in different biological pathways are still to be identified[7]. Although silencers are over-represented in the top hits, it could be biased from the screening readouts which looked for growth disadvantages. The real representation of silencers and enhancers in UCEs may benefit from future similar studies focusing on distinct biological pathways.

Many NCREs are expected to regulate multiple genes, which together may contribute to the phenotypes when NCREs were manipulated in their endogenous loci. Our results from studying all potential enhancers predicted from K562 cells showed that many enhancers may show silencer activities or no activities in luciferase assays (Fig. 3e,f), although biological functions were observed and nearby genes were deregulated when these NCREs were removed from their endogenous loci (Fig. 3d,i). This indicates that studying these NCREs in their endogenous loci is needed to define their potential functions and regulated pathways, where the dual-CRISPR screening system and its single-cell application extension are useful. Furthermore, results showed that in their endogenous loci, NCREs may have multiple regulatory functions, that is, acting as both enhancers and silencers (Fig. 3i), which has previously been observed and proposed on the basis of reporter assays[7,18]. Therefore, this dual-CRISPR system could be used to study NCREs with complex regulatory functions or even without previous defined functions (Fig. 1e,h), which would not be possible using the CRISPR–dCas9 activation/repression systems. It has been shown that many NCREs function in clusters and may have redundant roles. Results showed that the dual-CRISPR system was able to identify not only individual functional NCREs but also NCRE clusters in biological contexts (Fig. 4).

Unlike targeting genes, where genome-wide one-fit-for-all CRISPR libraries could be used for different cell types of the same genome, NCREs often function in a tissue-specific manner, which requires a versatile CRISPR system that could be tailored to target different NCREs with potential distinct functions. However, similar to other CRISPR systems, the dual-CRISPR system also relies on high editing efficiency and low off-target effects of the guide RNAs, especially as the dual-CRISPR system requires two functional guide RNAs to work at the same time. Therefore, it is recommended to design more distinct guide RNA pairs, which would increase the chance of multiple guide RNA pairs deleting the same region and also allow for more reliable statistical analyses[89]. Furthermore, recombination among guide RNA pairs is associated with the systematic problems of cloning and virus packaging[34,35]. These wrongly paired reads need to be filtered before the final analyses. As the dual-CRISPR system aims to delete the entire NCREs, it is not able to pinpoint the exact functional units within the NCRE, compared with other complementary CRISPRi systems, which tile the defined regions using an array of single-guide RNAs[90,91]. This dual-CRISPR screening system has multiple advantages over existing similar systems, and we expect that this system will have broad applications in studying the functions of NCREs and other non-coding parts of the genome. Our results also show that NCREs might play important roles in drug resistance, and we identified a critical UCE that regulates cardiomyocyte differentiation.

## Methods

### Cell culture

K562 cells were cultured in RPMI 1640 + L-glutamine (Gibco), 10% fetal bovine serum (Biowest) and 1% penicillin-streptomycin (Gibco). 293T cells were cultured in DMEM (Gibco), 10% fetal bovine serum and 1% penicillin-streptomycin. Cell density and culture conditions were maintained according to the ENCODE Cell Culture Guidelines. *NKX2-5*^eGFP/w^ hESCs were cultured in StemFlex medium (ThermoFisher A3349401) on Biolaminin (LN521-02)-coated six-well plates and passaged using TrypLE Select (ThermoFisher 12563011). K562/Cas9 and 293T/Cas9 cells were generated using lentivirus made from Lenti-Cas9-Blast (Addgene 52962).

### Dual-CRISPR plasmid construction

The lentiviral dual-CRISPR plasmid used for the screen was made on the basis of the lentiGuide-Puro (Addgene 52963). The human U6 and H1 promoters were cloned to replace the U6 promoter of the lentiGuide-Puro plasmid. The plasmid is referred to as Lenti-dual-CRISPR-U6-H1 (pBP43). The dual scaffolds were cloned to the pUC19 backbone. The plasmid is referred to as pUC19-dual-scaffold-1.0 (pBP44) or pUC19-dual-scafffold-2.0 (pBP49).

The dual-CRISPR plasmid containing Cas9 that was used to generate knockout clones for screen validations was made on the basis of the pSpCas9(BB)-2A-Puro (PX459) V2.0 (Addgene 62988). The human U6 and H1 promoters were cloned to replace the U6 promoter in the PX459V2, and the 3XFLAG tag on the Cas9 protein was replaced with the HA tag. The plasmid is referred to as Dual-CRISPR-Cas9-U6-H1 (pBP48). For cloning the paired guide RNAs into this plasmid, the guide RNAs were designed and cloned as follows:

U6 side: GTGGAAAGGACGAAACACCGN$_{20}$ (guide RNA target sequence) GTTTTAGAGCTAGAAATAGC

H1 side: TATGAGACCACTCTTTCCCGN$_{20}$ (guide RNA target sequence) GTTTTAGAGCTAGAAATAGC

Dual-CRISPR-Cas9-U6-H1 (pBP48) was digested with BbsI, and the dual scaffold was isolated from pUC19-dual-scaffold-2.0 (pBP49) digested with BbsI. Then, these four fragments were assembled using NEBuilder HiFi according to manufacturer protocol.

### Dual-CRISPR library design

The paired guide RNA sequences of the dual-CRISPR library targeting the UCEs and some validated enhancers were from the published computation pipeline CRISPETa (referred to as UCE library)[42]. The predicted K562 enhancers based on ENCODE ChIP-seq data using a machine learning model were selected as targets for the dual-CRISPR screen[16]. The software package CRISPRseek was used to search for potential protospacer sequences, with PAM NGG pattern as the potential CRISPR targeting regions[92]. Guide RNAs with high predicted cleavage efficiency and specificity were chosen with the following parameter cut-offs: guide RNAefficacy > 0.15, top5OfftargetTotalScore < 47 and top10OfftargetTotalScore < 50. These scores are based on experimentally derived off-target scoring schemes[93] to rank the off-target specificity of guide RNA design. The total scores of the top 5 and top 10 off-target regions were calculated, as these are the most likely off-target sites to be cleaved. Lower scores mean lower predicted off-target editing potential. Guide RNAs were further filtered to avoid overlapping with exons and repetitive regions.

First, the immediate upstream and downstream flanking regions (200 bp in size) of the enhancers were selected for guide RNA design. Single-guide RNAs from the two flanking regions were paired, resulting in up to 25 pairs of guide RNAs per targeting enhancer region (referred to as enhancer library 1 in Fig. 3a targeting 3,995 enhancers). Due to genomic repeats and other constraints, not all enhancers could be targeted by the previous design strategy. To target the rest of the predicted enhancers in K562 cells, pairs of guide RNAs were selected within the enhancer regions (5′ proximal and 3′ proximal to the enhancers), with around 14 pairs of guide RNAs per enhancer (referred to as enhancer library 2 in Fig. 3a targeting 13,020 enhancers). In this way, all ENCODE-predicted enhancers in K562 cells were targeted.

The oligo pools were then designed and ordered from CustomArray/GenScript. For UCE library and enhancer library 1, pairs of guide

RNAs were designed to be compatible with Lenti-dual-CRISPR-1.0 as follows:

ATCTTGTGGAAAGGACGAAACACCG-[guide RNA1, 20 nt]-gttttgagacgggatccCGTCTCAAAAC [reverse complement of guide RNA2, 20 nt]-CGGGAAAGAGTGGTCTCATACAGAACTTAT. For enhancer library 2, pairs of guide RNAs were designed to be compatible with Lenti-dual-CRISPR-2.0 as follows:

ATCTTGTGGAAAGGACGAAACACCG-[guide RNA1, 20 nt]-gtttttagagctaGAAAtagcaagttGAGACG-[barcode, 10 nt]-CGTCTCAACTTGCTATTTCTAGCTCTAAAAC-[reverse complement of guide RNA2, 20 nt]-CGGGAAAGAGTGGTCTCATACAGAACTTAT.

Each dual-CRISPR library also contains control paired guide RNAs from a previous study, which target promoters, exons and introns of 17 ribosomal genes and 3 cancer-related genes (*FOXA1*, *HOXB13* and *EZH2*), non-targeting guide RNAs and guide RNAs targeting the adeno-associated virus integration site 1 (AAVS1) loci[29].

### Single-cell dual-CRISPR library design

For single-cell dual-CRISPR library design, 82 paired guide RNA sequences were selected from the pooled dual-CRISPR screen experiments to target 42 significant NCREs affecting cell growth and imatinib resistance, with 1–3 unique pairs of guide RNAs chosen for each region. Ten extra pairs of guide RNAs were selected to target the AAVS1 region as negative controls. The oligo pool was then ordered and assembled into the single-cell dual-CRISPR library.

### Construction of the pooled and single-cell dual-CRISPR libraries

The synthesized oligo pools were first amplified by PCR using the following primers:

Forward primer, ATCTTGTGGAAAGGACGAAA; reverse primer, ATAAGTTCTGTATGAGACCA.

For Lenti-dual-CRISPR-1.0 libraries, PCR procedures using NEB-Next High-Fidelity 2X PCR master mix were 98 °C for 30 s, 18 cycles of 98 °C for 10 s, 68 °C for 30 s and 72 °C for 30 s, and 72 °C for 5 min. For each reaction, 80 ng of the oligo pool was used for a 100 μl PCR reaction, and 20 reactions per library were pooled. The pooled PCR products were further size selected and gel purified using a QIAGEN MinElute column. In the first step of library construction, the amplified oligo libraries were assembled into the digested Lenti-dual-CRISPR-U6-H1 plasmids using Gibson assembly. The assembly mix was made using 200 ng of digested dual-CRISPR-U6-H1 plasmids, 30 ng of insert DNA (at molar ratio 1:10) and 10 μl of 2× Gibson assembly master mix for a final volume of 20 μl. The assembly mix was incubated at 50 °C for 60 min, and ten reactions in total were pooled for each library. The pooled reaction products were purified by ethanol precipitation and resuspended in 10 μl water, from which 2 μl of the products was electroporated into 25 μl of Endura electrocompetent cells (Endura 60242-2). In total, five electroporation reactions were pooled and grown in 5 ml SOC recovery medium for 2 h. Then 5 μl from the 5 ml SOC recovery medium was used to perform serial dilution plating to determine the transformation efficiency, and the rest was further cultured in 1,000 ml lysogeny broth medium with 100 μg ml⁻¹ carbenicillin overnight. The plasmids containing the oligos were extracted using the Qiagen Maxiprep kit and further digested with BsmBI to open the plasmids at the restriction enzyme sites placed on oligo inserts. To assemble the final Lenti-dual-CRISPR-1.0 libraries, the BsmBI-digested plasmids containing the oligos were ligated with the dual-scaffold fragments (digested and isolated from pUC19-dual-scaffold-1.0) using T7 DNA ligase (NEB M0318). The ligation mix was made using 20 μg of digested plasmids, 1.2 μg of the dual-scaffold fragment, 200 μl of 2× T7 buffer and 40 μl T7 DNA ligase for a final volume of 400 μl. The ligation mix was incubated at 16 °C overnight. The ligation reaction products were purified by ethanol precipitation, resuspended in 210 μl of water and treated with Plasmid-Safe (Epicentre, E8101K). The products were

then purified using a QIAGEN MinElute column and eluted in 10 μl of water, which was electroporated into 50 μl of Endura electrocompetent cells and grown in 2 ml SOC recovery medium for 2 h. Then, 2 μl of the 2 ml SOC recovery medium was used to perform serial dilution plating to determine the transformation efficiency, and the rest was further cultured in 500 ml lysogeny broth medium with 100 μg ml⁻¹ carbenicillin. The final Lenti-dual-CRISPR-1.0 libraries were extracted using the Qiagen Maxiprep kit.

For Lenti-dual-CRISPR-2.0 libraries, the procedures to clone the oligo pools into the Lenti-dual-CRISPR-U6-H1 plasmids were similar. Gibson assembly was used to clone the dual-scaffold-2.0 (with 200 bp sequences inserted between the two scaffold sequences) to make the final Lenti-dual-CRISPR-2.0 libraries.

### Lentivirus production

For each library, the 293T cells were grown in five T175 flasks at 50% confluency before transfection. For each flask of 293T cells grown in 25 ml of fresh medium, 15 μg of library plasmids, 7 μg of psPAX2, 3.5 μg of pCMV-VSV-G and 76.5 μl of X-tremeGENE9 DNA transfection reagent were mixed in 1 ml of serum-free medium and used for transfection. Fresh medium was added the day following transfection. Media supernatant containing virus particles was collected on the second and third days after transfection, pooled, filtered and further concentrated using Lenti-X according to manufacturer protocol. The virus titre was then determined by making serial ($10^{-3}$ to $10^{-10}$) dilutions of 4 μl of frozen virus supernatant in media containing 8 μg ml⁻¹ of polybrene to infect 293T cells. Two days after infection, cells were selected with 2 μg ml⁻¹ puromycin for an additional 7 days. The virus titre was then calculated on the basis of the surviving colonies and the related dilution.

### Pooled dual-CRISPR screen

K562/Cas9 cells were infected with the respective virus libraries at an MOI of 0.2 by spin infection. For spin infection, $3 \times 10^6$ cells in each well of a 12-well plate were infected with the virus in 1 ml of medium containing 8 μg ml⁻¹ of polybrene. In total, four plates were used for each library infection to infect a total of $1.5 \times 10^8$ cells, which would result in ~300× to ~500× coverage of the dual-CRISPR libraries. Two days after infection, cells were selected using 2 mg ml⁻¹ of puromycin for another 6 days. Dead cells were then removed with Histopaque-1077 (Sigma) by centrifuging cells at 400 *g* for 30 min at room temperature. For each biological replicate experiment, the lentivirus was produced again and the infection was repeated.

After puromycin selection, an aliquot of $10^8$ K562/Cas9 cells infected with the respective dual-CRISPR libraries was frozen as the control population (day 0). From the same cell population, $10^8$ K562 cells were further cultured for another 15 days to identify essential NCREs that affect cell growth; and another $10^8$ cells were cultured under 0.1 μM imatinib treatment for 15 days. Dead cells were removed and drugs were refreshed during the subculture of imatinib-treated cells. On day 15, $10^8$ cells for each culture were collected and frozen for the next step. For 293T cells, procedures were similar, except that the dead cells were removed by refreshing the medium.

### Library sequencing

Genomic DNA was isolated using the QIAamp DNA Blood Maxi kit (QIAGEN). Dual-CRISPR paired guide RNAs integrated into the chromosomes were then PCR amplified using LongAmp *Taq* DNA Polymerase (NEB M0323) using primers annealing to U6 and H1 promoters (Supplementary Table 8). Stagger sequences were introduced to the PCR primers to increase the diversity of the next-generation sequencing libraries, when the flowcell only contains the dual-CRISPR sequencing libraries[94]. The stagger primers are not necessary if libraries are pooled with other sequencing samples with diverse sequences, and a one-step PCR using primers containing Illumina sequencing adaptors will be sufficient. For each 100 μl PCR reaction, 10 μg of genomic DNA, 50 μl

of 2× LongAmp master mix, 2 µl of 25 µM U6_stagger primers and 2 µl of 25 µM H1_stagger primers were used. In total, seven PCR reactions with different pairs of stagger primers were used and pooled, assaying 70 µg of genomic DNA. PCR procedures were 94 °C for 30 s, 25 cycles of 94 °C for 15 s, 60 °C for 15 s and 65 °C for 60 s, and 65 °C for 10 min. These fragments were then cleaned up and gel purified using MinElute PCR purification kit. Then, the Illumina TruSeq adaptors were ligated and sequencing libraries were prepared according to the ENCODE protocol and sequenced on the Illumina HiSeq4000 platform. It is recommended to quantify the Illumina sequencing libraries separately using qPCR before pooling with other samples.

### Pooled dual-CRISPR screening analysis
Cutadapt 3.4 was used to extract the unique 20 nt protospacer sequences from each pair of guide RNA sequences by locating the U6/H1 promoter sequences from the 5′ end and scaffold sequence from the 3′ end of the 20 nt protospacer sequence (U6 promoter, ATATATCTTGTGGAAAGGACGAAA; H1 promoter, ATAAGTTCT-GTATGAGACCACTCTT)[95]. The read pairs that did not contain the correct promoter and scaffold sequences were not considered. To ensure that functional CRISPR guide RNA sequences were counted, the protospacer sequence and the additional 20 bp (for both Lenti-dual-CRISPR-1.0 and Lenti-dual-CRISPR-2.0) into the CRISPR scaffold sequences were kept during the trimming of the sequence reads (tracrRNA identifying sequence, AAGTTAAAAT). Trimmed reads were then mapped to the indexed paired protospacer references generated by Bowtie2 on the basis of our initial library designs, and only aligned reads with mapping quality scores over 23 were used for downstream analyses[96].

MAGeCK RRA was used to identify the significant hits depleted after 15 days of culture compared with the day-0 initial cell population, with cut-off of RRA score < 0.01 (ref. 43). MAGeCK maximum likelihood estimation (MLE) was performed to identify NCREs that confer imatinib resistance by comparing the day-15 imatinib-treated cell population, day-15 non-treatment culture cell population and day-0 initial cell population[97]. The NCREs whose loss confers imatinib resistance were identified as regions that were positively selected (that is, NCREs with beta scores > mean + 2 × s.d.) in the day-15 imatinib-treated cell populations but weakly selected in the day-15 non-treatment culture populations. Gene pathway over-representation analyses were based on Gene Ontology term and Kyoto Encyclopedia of Genes and Genomes (KEGG) pathway definitions with the cut-off of FDR < 0.25. Genes potentially affected by NCREs were identified as the ones located in the range of ±1 Mb of the NCRE regions.

### Single-cell RNA-seq combined with the dual-CRISPR screen
The single-cell dual-CRISPR library was introduced into K562/Cas9 cells via spin infection at a final MOI of 0.2. After 1 week, cells that incorporated the single-cell dual-CRISPR library were sorted on the basis of GFP expression using FACS sort. To prepare for single-cell processing, $0.5 × 10^6$ cells at a concentration of 1,500 cells per ml in 0.04% BSA-PBS were used for single-cell RNA and guide RNA capture according to the 10x Genomics protocol. The 10x Genomics Chromium platform was used to generate the single-cell libraries, which were sequenced on the Illumina NovaSeq 6000 platform.

### Single-cell dual-CRISPR screening analysis
The Cell Ranger 6.0.1 pipeline was used to process Chromium single-cell data to align reads and generate feature-barcode matrices for the mRNAs and capture guide RNAs. Seurat 4.0.2 was used to process the single-cell RNA-seq data[98]. The gene expression matrix was normalized using the 'LogNormalize' method with a scale factor of 10,000 and log transformed. Cell-cycle-related scores were regressed during data scaling. Differential expression tests and P value calculations were performed using the MAST-fitted model on the basis of the whole

transcriptome data in all single cells, except for genes with mean normalized expression <0.01 to filter out low-expression genes[78]. To control for confounding factors in the differential expression testing between perturbed and control cell groups, we included logarithms of the total number of expressed genes per cell and the total number of guide RNAs detected per cell as covariates in the MAST regression model to overcome test miscalibration.

TADs around the candidate NCREs were used to narrow down the potential genes deregulated by CRISPR editing. The K562 TAD dataset was downloaded from 3D Genome Browser and the median window size is ~1 Mbp. The neighbouring gene coordinates were extracted using biomaRt[99]. The gene expression changes were quantified as the $\log_2$(fold change) of the mean of normalized gene expression from the perturbed cell population divided by the mean of normalized gene expression from the negative control cell population.

### RNA-seq sample preparation and analysis
RNA was isolated using ISOLATE II RNA mini kit (BIO-52073), and sequencing libraries were made with the Invitrogen Collibri Stranded RNA Library Preparation kit (A39003024) according to manufacturer protocol. The whole transcriptome sequencing was done on the Illumina NovaSeq 6000 platform.

The Snakemake pipeline (https://github.com/snakemake-workflows/rna-seq-star-deseq2) was used to process the bulk RNA-seq sample. Briefly, Cutadapt v.3.4 was used to trim adaptors from reads, and STAR was used to align the spliced transcripts to the reference genome (GRCh37 assembly release 75) and quantify the read counts per gene[100]. DESeq2 was used to perform differential expression analysis[101]. P values were corrected by the BH procedure for FDR control.

The KEGG pathway enrichment analysis was based on the GSEA method with FDR < 0.25 as the cut-off as recommended by GSEA, which integrates the expression level of individual genes and aggregates gene expression in the pathway analysis to manifest the phenotypic differences, that is, to show whether the pathway is activated or suppressed[73].

### Enhancer cluster analysis
Individual enhancers from the dual-CRISPR libraries (enhancer libraries 1 and 2) were first clustered on the basis of the super-enhancer annotations in K562 cells[72]. Two computational methods, GSEA and MAGeCK RRA, were used to compute the depletion scores of enhancer clusters associated with cell growth[43,73]. P values obtained by the two methods were corrected using the BH procedure for FDR control. Among the top 20 clusters identified from both methods, 10 clusters were shared, which were considered for further validation. The illustration of the enhancer cluster was created with Biorender.com.

### Luciferase assay
Candidate NCREs were amplified with primers containing a homologous arm from the genomic DNA of K562 cells. These fragments were then inserted in front of the promoters of the luciferase plasmids pGL4.23 (Promega, with some modification on the cloning sites, detecting the enhancer activity) and pGL4.53 (Promega, detecting the silencer activity) by using NEBuilder HiFi. Cells were then co-transfected with the pRL-CMV Renilla reporter plasmid and the pGL4.53 or pGL4.23 plasmid with the NCRE sequences inserted. The luciferase assay was performed using the Dual-Luciferase Reporter Assay System from Promega according to manufacturer protocol. Original luciferase plasmid without any insertion was used as the control. All luciferase assays were from three independent transfections performed on different days. All tested regions and associated primers are listed in Supplementary Table 8.

### Dual-CRISPR–Cas9-mediated NCRE knockout
Paired guide RNAs targeting the 5′ and 3′ ends of the NCREs were selected from the screening libraries. All selected guide designs falling in intron

regions were at least 10 bp away from adjacent splicing sites. The guide RNA sequences were cloned into the dual-CRISPR-Cas9-U6-H1 (pBP48) plasmid containing the guide RNA scaffold and Cas9 sequence. K562 cells were transfected with the plasmids containing the respective pairs of guide RNAs and then selected for successful transfection using puromycin. K562 cells transfected with the dual-CRISPR plasmids containing a pair of guide RNAs targeting the GFP sequence were used as the CRISPR-editing control (control_KO). Single clones of cells were picked and verified using PCR and Sanger sequencing.

### Quantitative PCR

The expression of eRNA was first confirmed on the basis of total RNA-seq and GRO-seq data[74]. For both gene and eRNA expression, RNA was extracted using ISOLATE II RNA mini kit, including DNase I digestion (Bioline BIO-52073). The complementary DNA was synthesized using SuperScript IV VILO master mix (Invitrogen 11756050). Real-time PCR was performed using the SensiFAST SYBR No-ROX kit (Bioline BIO-98020) on the Biorad CFX Opus 384 real-time PCR system. The expression of the housekeeping gene *GAPDH* was used as the control. For all qPCR experiments, three biological replicates were performed and *P* values were calculated using an unpaired two-tailed Student's *t*-test.

### Cell proliferation assay

Cell proliferation assay was performed by mixing the NCRE_KO cell lines with cells expressing GFP at a 1:1 ratio. The changes in GFP percentage were monitored at indicated time points by FACS. Ctrl_KO cells were used as the negative control. To test the imatinib response, NCRE_KO clones and K562 cells were seeded into a 96-well plate and treated with 0.4 μM imatinib (Selleckchem STI571) for 3 days. Cell viability was then measured using the CellTiter-Blue viability assay (Promega G8082). Relative survival data were normalized to their respective untreated NCRE_KO clones and K562 cells.

### CRISPR–Cas9-mediated NCRE knockout in hESCs

The crRNAs targeting the 5' and 3' ends of UCE PAX6_Tarzan or the selected AAVS1 region were ordered from Integrated DNA Technologies (IDT). To form the respective crRNA:tracrRNA duplex, 2.2 μl of crRNA and 2.2 μl of tracrRNA (IDT) were mixed in 0.6 μl nuclease-free duplex buffer. The mix was heated at 95 °C for 5 min, then cooled to room temperature for 10 min. Then, 0.5 μl of crRNA:tracrRNA duplex, 0.24 μl of Cas9 Nuclease V3 (IDT) and 0.76 μl of buffer R (Neon Transfection System) were mixed and incubated for 20 min at room temperature to form the CRISPR–Cas9 complex. Electroporation was performed by mixing 0.25 million *NKX2-5*<sup>eGFP/w</sup> hESCs in 22 μl buffer R with 2 μl of dual-CRISPR–Cas9 complex according to the Neon nucleofector protocol. Cells were then transferred to a laminin-coated 12-well plate for culturing. Genomic DNA was isolated 4 days later and the deletion of PAX6_Tarzan was verified using PCR and Sanger sequencing. Primer and guide RNA sequences are listed in Supplementary Table 8. The maintenance of *NKX2-5*<sup>eGFP/w</sup> hESCs and cardiomyocyte differentiation were performed as described previously[102].

### Reporting summary

Further information on research design is available in the Nature Portfolio Reporting Summary linked to this article.

### Data availability

Pooled screen and scRNA-seq data are available at GEO under accession code GSE254241. RNA-seq data are available at GEO under accession code GSE247234. The raw and analysed datasets generated during the study are available for research purposes from the corresponding authors on reasonable request. We plan to make the reagents widely available to the academic community through Addgene. Source data are provided with this paper.

### Code availability

The code for processing the pooled dual-CRISPR screening data is available on GitHub at https://github.com/PangLab/DualCRISPR_pooled_screen_snakemake_pipeline (ref. 103).

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

## Acknowledgements

This work was supported by the Gisela Thier Fellowship from Leiden University Medical Center, the KWF Young Investigator Grant 11707 from the Dutch Cancer Society, and the ERC Starting Grant 950655-Silencer from the European Research Council (all awarded to B.P.). Y.L., S.S. and L.Z. were supported by CSC scholarships. X.L. was supported by the departmental start-up funding from Case Western Reserve University. M.P.S. was supported by NIH Grant 5RM1HG00773509 from the Centers of Excellence in Genomic Science (CEGS). R.P.D. was supported by Novo Nordisk Foundation grant NNF21CC0073729 from the Novo Nordisk Foundation Center for Stem Cell Medicine (reNEW). J.J.G. was supported by NIH grant K08CA245024. We thank M. Mol and M. S. González for providing technical support for the hESC cardiomyocyte differentiations. Parts of figures were prepared using BioRender.com.

## Author contributions

B.P. conceptualized the project. B.P. and Y.L. developed the methodology. Y.L., B.P., A.A.-H., L.Z., M.K., J.J.S. and S.S. conducted investigations. X.L., M.T., Y.L. and N.X. developed software and conducted formal analysis. B.P. and Y.L. wrote the original draft. B.P., M.T., N.X., A.A.-H., S.S., X.L., R.P.D., H.M., J.J.G. and M.P.S. reviewed and edited the manuscript. Y.A., S.L.K., R.P.D., H.M., J.J.G. and M.P.S. acquired resources. B.P., X.L. and M.P.S. supervised the project. B.P. acquired funding.

## Competing interests

M.P.S. is a founder and member of the science advisory boards of Personalis, SensOmics, Qbio, January, Mirvie and Filtricine, and a member of the science advisory boards of Genapsys and Epinomics. B.P. and M.P.S. have filed a provisional patent application on this work (US36/586269). The other authors declare no competing interests.

## Additional information

**Extended data** is available for this paper at https://doi.org/10.1038/s41551-024-01204-8.

**Correspondence and requests for materials** should be addressed to Michael P. Snyder, Xiao Li or Baoxu Pang.

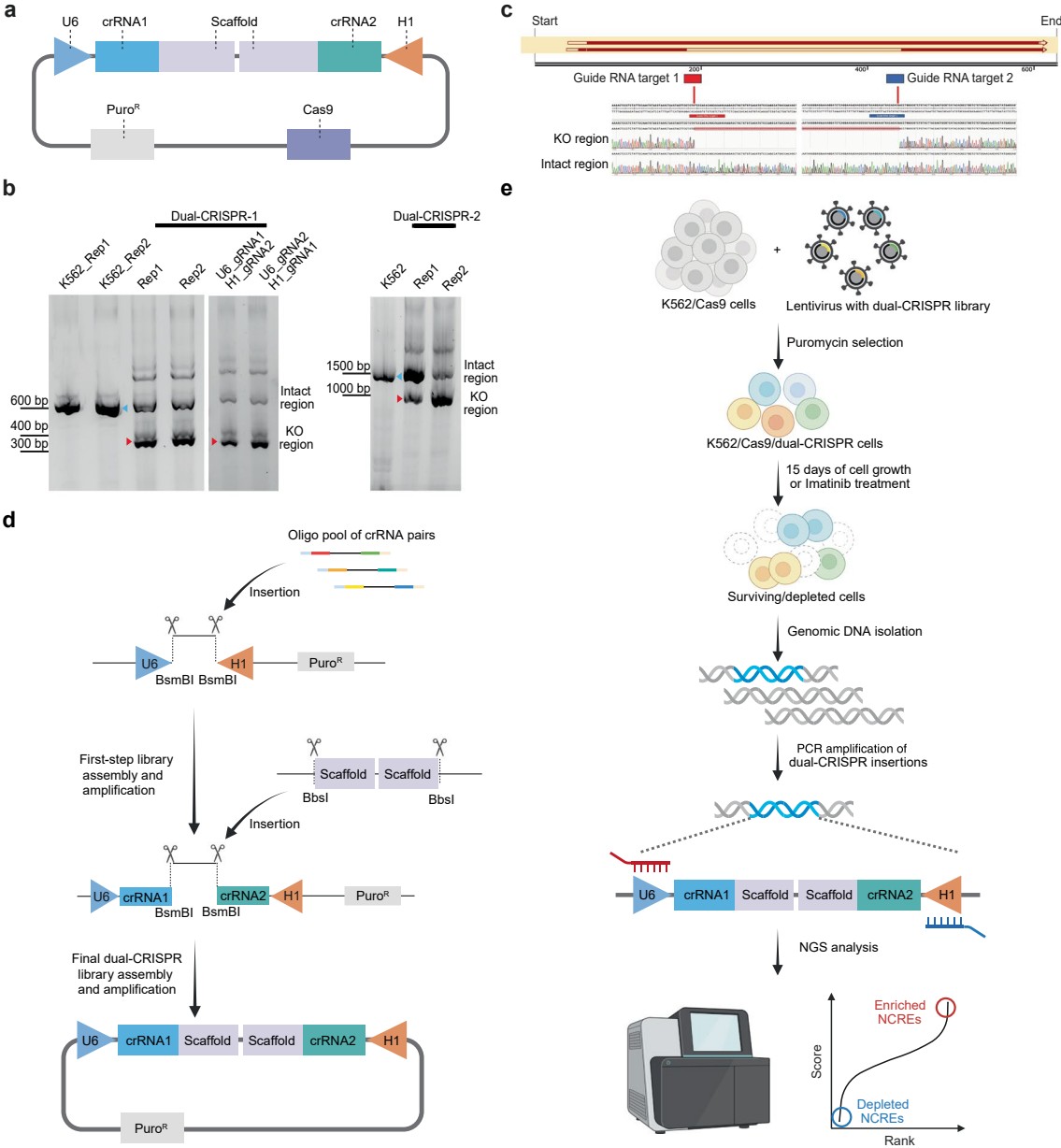

**Extended Data Fig. 1 | Design and testing of the dual-CRISPR screening system. a**, The expression of the two guide RNAs was driven by the human U6 and H1 promoters, respectively. The transcription driven by the U6 and H1 promoters is in a convergent direction, with transcription termination sequences at the end of each guide RNA scaffold sequence. The plasmid also expresses the *S.pyogenes* Cas9 (SpCas9) nuclease and the puromycin resistance gene to help select transfected cells. Created with BioRender.com. **b**, K562 cells were transiently transfected with the designed plasmids containing two guide RNAs flanking the testing sequences. In the left panel, the PCR amplicon of 600 bp in size (intact region) indicates the original DNA sequence in the genome, and the genomic sequence knockout (KO) by the two guide RNAs is shown as the PCR amplicon of 350 bp in size (KO region). The differences in transcription and editing efficiency driven by U6 and H1 promoters were tested by swapping the respective gRNAs, indicated as U6_gRNA1/H1_gRNA2 and U6_gRNA2/H1_gRNA1. Similar editing efficiency was observed driven by the two different promoters. In the right panel, the PCR amplicon of 1500 bp in size (intact region) indicates the original DNA

sequence upstream of the *Top2a* gene, and the genomic sequence knockout by the two guide RNAs is shown as the PCR amplicon of 1000 bp in size (KO region). **c**, Sanger sequencing results of the PCR fragments in (**b**, dual-CRISPR-1) show complete knockout targeted by the designed two guide RNAs. **d**, Cloning strategy of the dual-CRISPR screening libraries. To make the dual-CRISPR libraries, oligo pools that contain dual crRNAs and additional restriction enzyme recognition sites were cloned between the two promoters. Then the plasmids containing the crRNAs were amplified and digested again to insert the dual-tracrRNA scaffolds to form the complete dual-CRISPR plasmid library. Created with BioRender.com. **e**, Cells containing dual-CRISPR libraries can be used to study different biological pathways. Genomic DNA from different cell populations is then isolated, and the dual-CRISPR libraries can be amplified by direct PCR reactions, ready for next-generation sequencing (NGS). The changes in abundance of dual CRISPRs are calculated by different algorithms to identify potential NCRE hits. Created with BioRender.com.

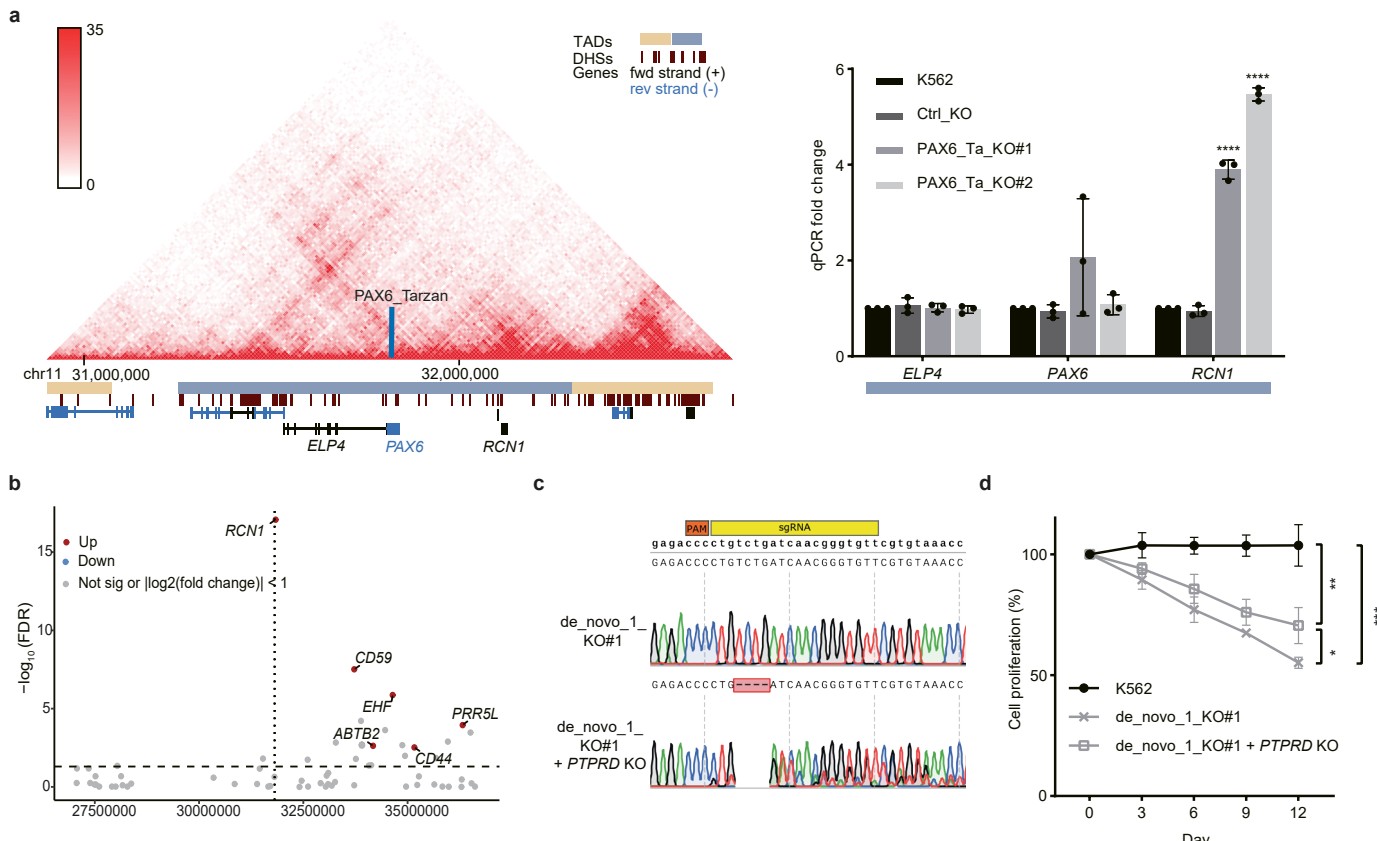

**Extended Data Fig. 2 | NCREs regulate their surrounding genes to exert their functions. a**, TADs identified by Hi-C surrounding PAX6_Ta are shown, and the location of PAX6_Ta is indicated by the vertical blue bar. Horizontal yellow and blue bars indicate distinct TADs. Transcription of three nearby genes within the same TAD as PAX6_Ta was quantified by qPCR ($n$ = 3 biological independent samples; bars show mean value ± s.d.; ****$P$ < 0.0001, calculated using unpaired t-test). **b**, RNA-seq was performed to identify differentially expressed genes around NCRE PAX6_Ta from PAX6_Ta_KO cells versus K562 WT cells. The x axis represents the coordinates of genes surrounding PAX6_Ta ranging from - 5 Mb to + 5 Mb. The y axis shows the -log10(FDR) of nearby genes by DESeq2. The dashed horizontal line indicates the FDR cutoff of 0.05, and the vertical line indicates the location of PAX6_Ta. Each dot represents one gene, the red dots represent significantly up-regulated genes, blue dots represent significantly down-regulated genes, and the gray ones are either non-significant genes or genes with

|log2FC| <1. **c**, CRISPR knockout was used to silence the *PTPRD* gene in de_novo_1_KO#1 cells. Sanger sequencing of gRNA targeting region within the exon region of *PTPRD* gene (chr9:8314246-8733946, hg19) showed a 4-nucleotide-deletion leading to a frameshift mutation, compared to the original de_novo_1_KO#1. **d**, Growth effect of *PTPRD* gene knockout in de_novo_1 KO clone (de_novo_1_KO#1 + *PTPRD*_KO) compared to the de_novo_1 KO clone (de_novo_1_KO#1). Cell proliferation assay was performed by mixing the KO cell lines with cells expressing GFP at a 1:1 ratio. The changes in GFP percentage were monitored at indicated time points by FACS. K562 cells served as control. The y axis represents the relative ratio of the GFP negative cells to the positive cells. The ratio of cells in the initial mixture was set as 100%. ($n$ = 3 independent biological samples; values are shown as the mean ± s.d.; *$P$ = 0.0472, **$P$ = 0.0061, ***$P$ = 0.0008, calculated using two-way ANOVA).

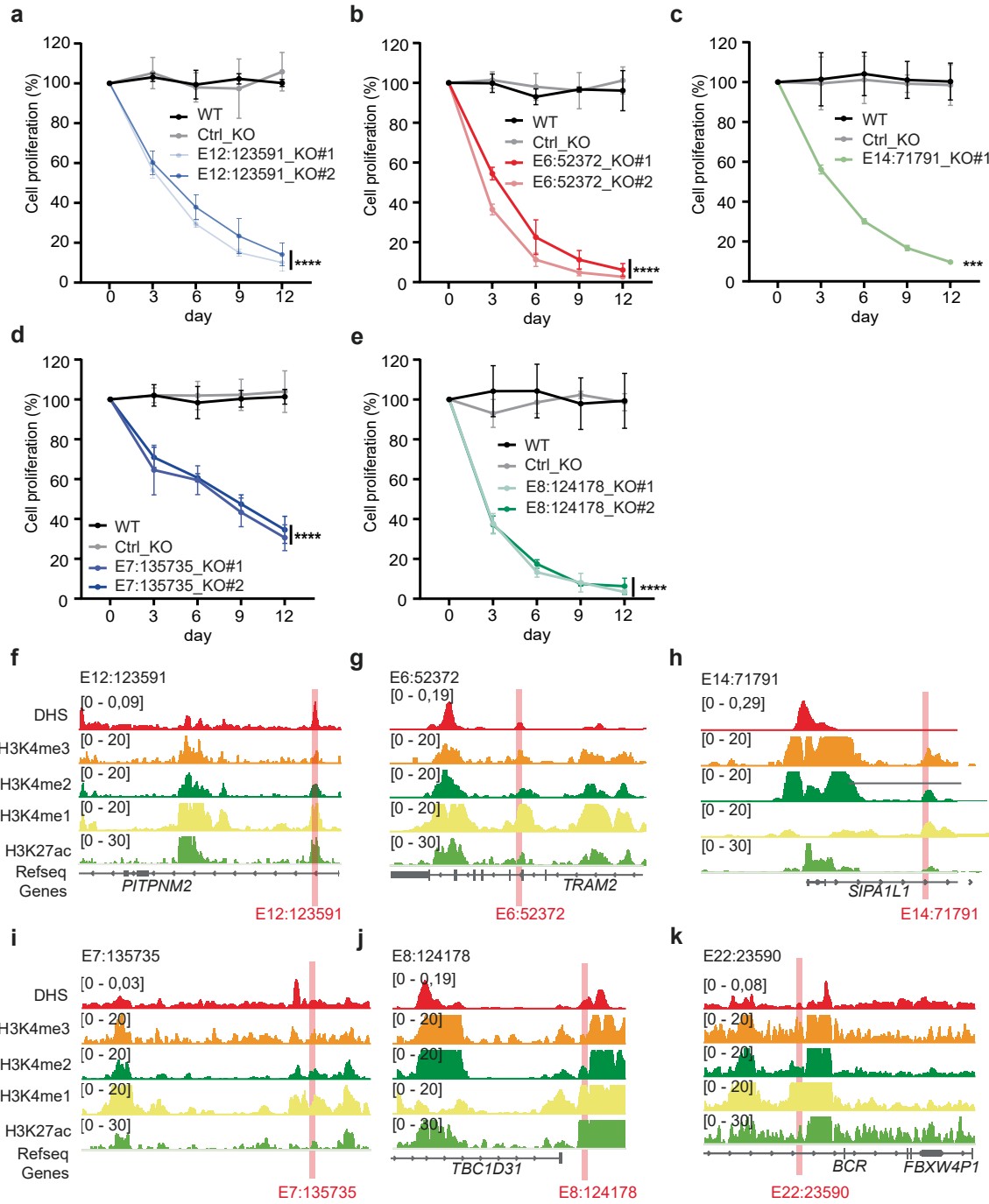

**Extended Data Fig. 3 | Cell growth effects and epigenetic patterns of essential enhancers in K562 cells. a-e**, The removal of essential enhancers E12:123591 (**a**), E6:52372 (**b**), E14:71791 (**c**), E7:135735 (**d**), E8:124178 (**e**) caused cell growth disadvantage in K562 cells. Cell proliferation assay was performed by mixing the KO cell lines with cells expressing GFP at a 1:1 ratio. The changes in GFP percentage were monitored at indicated time points by FACS. Cells with dual-CRISPR guide RNAs targeting GFP sequences served as negative controls (Ctrl_KO). The y axis represents the relative ratio of the GFP negative cells to the positive cells. The ratio of cells in the initial mixture was set as 100% ($n$ = 3 biological independent samples; values are shown as the mean ± s.d.; ***$P$ = 0.0002, ****$P$ < 0.0001, calculated using two-way ANOVA). **f-k**, Epigenetic signatures surrounding the identified essential enhancers in K562 cells. E12:123591 (**f**), E6:52372 (**g**), E14:71791 (**h**), E7:135735 (**i**), E8:124178 (**j**) and E22:23590 (**k**).

# Reporting Summary

## Statistics

For all statistical analyses, confirm that the following items are present in the figure legend, table legend, main text, or Methods section.

| n/a | Confirmed | |
|---|---|---|
| ☐ | ☒ | The exact sample size (*n*) for each experimental group/condition, given as a discrete number and unit of measurement |
| ☐ | ☒ | A statement on whether measurements were taken from distinct samples or whether the same sample was measured repeatedly |
| ☐ | ☒ | The statistical test(s) used AND whether they are one- or two-sided<br>*Only common tests should be described solely by name; describe more complex techniques in the Methods section.* |
| ☐ | ☒ | A description of all covariates tested |
| ☐ | ☒ | A description of any assumptions or corrections, such as tests of normality and adjustment for multiple comparisons |
| ☐ | ☒ | A full description of the statistical parameters including central tendency (e.g. means) or other basic estimates (e.g. regression coefficient) AND variation (e.g. standard deviation) or associated estimates of uncertainty (e.g. confidence intervals) |
| ☐ | ☒ | For null hypothesis testing, the test statistic (e.g. *F*, *t*, *r*) with confidence intervals, effect sizes, degrees of freedom and *P* value noted<br>*Give P values as exact values whenever suitable.* |
| ☒ | ☐ | For Bayesian analysis, information on the choice of priors and Markov chain Monte Carlo settings |
| ☒ | ☐ | For hierarchical and complex designs, identification of the appropriate level for tests and full reporting of outcomes |
| ☐ | ☒ | Estimates of effect sizes (e.g. Cohen's *d*, Pearson's *r*), indicating how they were calculated |

*Our web collection on statistics for biologists contains articles on many of the points above.*

## Software and code

Policy information about availability of computer code

| | |
|---|---|
| Data collection | Sequencing was performed via the Illumina Hiseq4000 platform. Gels and blots were imaged by using the Bio-rad ChemiDoc Imaging System. |
| Data analysis | The R package CRISPRseek was used to search for potential protospacers sequences with PAM NGG pattern as the potential CRISPR targeting regions.<br><br>Cutadapt 3.4 was used to extract the unique 20nt protospacer sequences from each pair of guide RNA sequences by locating the U6/H1 promoter sequences from the 5' end and scaffold sequence from the 3' end of the 20nt protospacer sequence (U6 promoter, ATATATCTTGTGGAAAGGACGAAA; H1 promoter, ATAAGTTCTGTATGAGACCACTCTT). The trimmed reads were then mapped to the indexed paired protospacers references generated by Bowtie2 based on the library designs, and only aligned reads with mapping quality (MAPQ) score over 23 were used for downstream analyses.<br><br>MAGeCK RRA was used to identify the significant hits depleted after 15-day culture compared with the day 0 initial cell population, with the cutoff of RRA score < 0.01.<br><br>MAGeCK MLE was performed to identify NCREs that confer imatinib resistance, by comparing the 15-day imatinib-treated cell population, 15-day culture cell population and day 0 initial cell population. The NCREs whose loss confers Imatinib resistance were identified as regions that were positively selected (i.e., NCREs with beta scores > mean + 2×s.d.) in the 15-day imatinib-treated populations but are weakly selected in the 15-day culture populations.<br><br>The snakemake pipeline (https://github.com/snakemake-workflows/rna-seq-star-deseq2) was used to process the bulk RNA-seq sample. The gene pathway enrichment analyses based on GO term and KEGG pathway definitions were conducted by clusterProfiler with the cutoff of |

FDR < 0.25.
3D Genome Browser (www.3dgenome.org) and WashU Epigenome Browser (https://epigenomegateway.wustl.edu/browser/) were used for data visualization.

Cell Ranger 6.0.1 pipeline was used to process Chromium single-cell data to align reads, generate feature-barcode matrices for the mRNAs, and capture guide RNAs. Seurat 4.0.2 was used to process the single-cell RNA-seq data.

The code for processing the pooled dual-CRISPR screen data is available on GitHub: https://github.com/PangLab/DualCRISPR_pooled_screen_snakemake_pipeline.

For manuscripts utilizing custom algorithms or software that are central to the research but not yet described in published literature, software must be made available to editors and reviewers. We strongly encourage code deposition in a community repository (e.g. GitHub). See the Nature Portfolio guidelines for submitting code & software for further information.

# Data

Policy information about availability of data

All manuscripts must include a data availability statement. This statement should provide the following information, where applicable:
- Accession codes, unique identifiers, or web links for publicly available datasets
- A description of any restrictions on data availability
- For clinical datasets or third party data, please ensure that the statement adheres to our policy

Pooled screen and scRNA-seq sequencing data are available at GEO under accession code GSE254241. RNA-seq data are available at GEO under accession code GSE247234. The raw and analysed datasets generated during the study are available for research purposes from the corresponding authors on reasonable request. We plan to make the reagents widely available to the academic community through Addgene.

# Research involving human participants, their data, or biological material

Policy information about studies with human participants or human data. See also policy information about sex, gender (identity/presentation), and sexual orientation and race, ethnicity and racism.

| | |
|---|---|
| Reporting on sex and gender | The study did not involve human participants. |
| Reporting on race, ethnicity, or other socially relevant groupings | – |
| Population characteristics | – |
| Recruitment | – |
| Ethics oversight | – |

Note that full information on the approval of the study protocol must also be provided in the manuscript.

# Field-specific reporting

Please select the one below that is the best fit for your research. If you are not sure, read the appropriate sections before making your selection.

☒ Life sciences    ☐ Behavioural & social sciences    ☐ Ecological, evolutionary & environmental sciences

For a reference copy of the document with all sections, see nature.com/documents/nr-reporting-summary-flat.pdf

# Life sciences study design

All studies must disclose on these points even when the disclosure is negative.

| | |
|---|---|
| Sample size | For dual-CRISPR-sequencing experiments, K562/Cas9 cells were infected with the respective virus libraries at a multiplicity of infection 0.2 by spin-infection. For spin-infection, 3 million cells in each well of a 12-well plate were infected in 1 ml of medium containing 8 µg/ml of polybrene. In total, four plates were used for each infection to infect a total of 150 million cells, which would result in ~300× to ~500× coverage of the dual-CRISPR libraries.<br><br>For single-cell RNA-sequencing experiments, cells were loaded to recover a median coverage of 500 cells per guide. To prepare for single-cell processing, 500.000 cells at a concentration of 1,500 cells /ml in 0.04% BSA-PBS were used for single-cell RNA and guide RNA capture according to the 10x Genomics protocol. |
| Data exclusions | Pooled dual-CRISPR screen: puromycin selected cells were used for pooled dual CRISPR screen. Plasmid DNA did not contain the correct promoter and scaffold sequences ligations were not considered in pooled screen analysis.<br><br>Single-cell dual-CRISPR screen: GFP-expressing cell were sorted and loaded for direct gRNA capture and scRNA-seq assay. Cells with low UMI |

counts were removed from single cell CRISPR analysis in cellranger pipeline. Only single cells with captured correct guide-RNA pairs were kept for downstream analysis (4,470 cells).

| | |
|---|---|
| Replication | All PCR assays, Western blots, luciferase assays and qPCR assays were performed at least 2–3 times using biological replicates performed at different days. Similar observation was made for each replicate. |
| Randomization | Randomization was not relevant to the study, as it was based on comparing defined CRISPR-Cas9 edited cells under distinct time points or treatment conditions. |
| Blinding | Blinding was not relevant to the study, as it was based on objective quantitative methods. |

# Reporting for specific materials, systems and methods

We require information from authors about some types of materials, experimental systems and methods used in many studies. Here, indicate whether each material, system or method listed is relevant to your study. If you are not sure if a list item applies to your research, read the appropriate section before selecting a response.

## Materials & experimental systems

| n/a | Involved in the study |
|---|---|
| ☐ | ☒ Antibodies |
| ☐ | ☒ Eukaryotic cell lines |
| ☒ | ☐ Palaeontology and archaeology |
| ☒ | ☐ Animals and other organisms |
| ☒ | ☐ Clinical data |
| ☒ | ☐ Dual use research of concern |
| ☒ | ☐ Plants |

## Methods

| n/a | Involved in the study |
|---|---|
| ☒ | ☐ ChIP-seq |
| ☐ | ☒ Flow cytometry |
| ☒ | ☐ MRI-based neuroimaging |

## Antibodies

| | |
|---|---|
| Antibodies used | Actin antibody # MA5-15452, Thermo Fisher Scientific (1:5000 dilution); Monoclonal ANTI-FLAG M2 antibody # F1804, Sigma (1:1000 dilution) |
| Validation | Actin antibody # MA5-15452<br>Species Reactivity: Hamster, Human, Mouse, Non-human primate, Rat<br>Specificity: MA5-15452 targets beta-Actin in FACS, IF, and WB applications and shows reactivity with Hamster, Human, mouse, Non-human primate, and Rat samples. The MA5-15452 immunogen is synthetic peptide corresponding to amino-terminal residues of human beta-Actin, conjugated to KLH.MA5-15452 detects beta-Actin which has a predicted molecular weight of approximately 42kDa.<br>Monoclonal ANTI-FLAG M2 antibody # F1804<br>Monoclonal ANTI-FLAG M2 antibody produced in mouse, 1 mg/mL, clone M2, affinity isolated antibody, buffered aqueous solution (50% glycerol, 10 mM sodium phosphate, and 150 mM NaCl, pH 7.4). Detects a single band of protein on a Western Blot from mammalian crude cell lysates. |

## Eukaryotic cell lines

Policy information about cell lines and Sex and Gender in Research

| | |
|---|---|
| Cell line source(s) | K562 and 293T cells were all from the cell-line collections of the ENCODE project. hESC NKX2-5eGFP/w cells were from the authors of Nat Methods 8, 1037–1040 (2011). |
| Authentication | Cell lines from ENCODE were purchased from ATCC. ATCC performs routine cell-line authentication, and ENCODE maintains stocks of previously authenticated cell lines. hESC NKX2-5eGFP/w were authenticated by the providers. |
| Mycoplasma contamination | All cell lines tested negative for mycoplasma contamination. |
| Commonly misidentified lines (See ICLAC register) | No commonly misidentified cells lines were used. |

# Flow Cytometry

## Plots

Confirm that:

☒ The axis labels state the marker and fluorochrome used (e.g. CD4-FITC).

☒ The axis scales are clearly visible. Include numbers along axes only for bottom left plot of group (a 'group' is an analysis of identical markers).

☒ All plots are contour plots with outliers or pseudocolor plots.

☒ A numerical value for number of cells or percentage (with statistics) is provided.

## Methodology

| | |
|---|---|
| Sample preparation | K562 cells were analysed by flow cytometry or purified by fluorescence-activated cell sorting after lentiviral transduction and growth in standard culture conditions. GFP expression was used as a marker of successful transduction. |
| Instrument | Flow-cytometry data were collected using one LSR II flow cytometers, and cell sorting was performed on a FACSAria2 (BD Biosciences). |
| Software | Flow-cytometry data were collected using BD FACSDiva software (versions 8.0 and 8.0.1). FACSDiva was also used for cell sorting. And FlowJo_V10 was used for analysis. |
| Cell population abundance | – |
| Gating strategy | SSC/FSC gates were first applied to determine cells. Cells were then gated to separate GFP+ from GFP- cells. |

☒ Tick this box to confirm that a figure exemplifying the gating strategy is provided in the Supplementary Information.

