## [Peer Review File · Nature Biomedical Engineering]

Genome-wide Cas9-mediated screening of essential non-coding regulatory elements via libraries of paired single-guide RNAs

Corresponding author: Baoxu Pang

Editorial note

This document includes relevant written communications between the manuscript's corresponding author and the editor and reviewers of the manuscript during peer review. It includes decision letters relaying any editorial points and peer-review reports, and the authors' replies to these (under 'Rebuttal' headings). The editorial decisions are signed by the manuscript's handling editor, yet the editorial team and ultimately the journal's Chief Editor share responsibility for all decisions.

Any relevant documents attached to the decision letters are referred to as **Appendix #**, and can be found appended to this document. Any information deemed confidential has been redacted or removed. Earlier versions of the manuscript are not published, yet the originally submitted version may be available as a preprint. Because of editorial edits and changes during peer review, the published title of the paper and the title mentioned in below correspondence may differ.

Correspondence

Mon 23 Oct 2023

Decision on Article NBME-23-2310A

Dear Prof Pang,

Thank you again for submitting to *Nature Biomedical Engineering* your manuscript, "The genome-wide dual-CRISPR screening identifies essential non-coding regulatory elements". The manuscript has been seen by three experts, whose reports you will find at the end of this message. You will see that the reviewers appreciate the work, and that they raise a number of technical criticisms that we hope you will be able to address. In particular, we would expect that a revised version of the manuscript provides:

- * Extended validation data of the efficacy and specificity of the dual-CRISPR system, as per the relevant comments from all reviewers.
- * Clear discussion of the limitations of the dual-CRISPR system.
- * Thorough and clear reporting of the statistics and methodology.

Also, please do ensure that the data from the screens are deposited on GEO on submission of the revised manuscript. This will be a requirement for publication.

When you are ready to resubmit your manuscript, please upload the revised files, a point-by-point rebuttal to the comments from all reviewers, the reporting summary, and a cover letter that explains the main improvements included in the revision and responds to any points highlighted in this decision.

Please follow the following recommendations:

- * Clearly highlight any amendments to the text and figures to help the reviewers and editors find and understand the changes (yet keep in mind that excessive marking can hinder readability).
- * If you and your co-authors disagree with a criticism, provide the arguments to the reviewer (optionally, indicate therelevant points in the cover letter).

* If a criticism or suggestion is not addressed, please indicate so in the rebuttal to the reviewer comments and explain the reason(s).

* Consider including responses to any criticisms raised by more than one reviewer at the beginning of the rebuttal, in a section addressed to all reviewers.

* The rebuttal should include the reviewer comments in point-by-point format (please note that we provide all reviewers will the reports as they appear at the end of this message).

* Provide the rebuttal to the reviewer comments and the cover letter as separate files.

We hope that you will be able to resubmit the manuscript within 16 weeks from the receipt of this message. If this is the case, you will be protected against potential scooping. Otherwise, we will be happy to consider a revised manuscript as long as the significance of the work is not compromised by work published elsewhere or accepted for publication at *Nature Biomedical Engineering*.

We hope that you will find the referee reports helpful when revising the work, which we look forward to receive. Please do not hesitate to contact me should you have any questions.

Best wishes,

Pep

Pep Pàmies
Chief Editor, Nature Biomedical Engineering

Reviewer #1 (Report for the authors (Required)):

NCREs, including UCEs, play pivotal roles in various biological pathways and can be genomic medicine targets. However, existing methods do not measure the functions of NCREs in their natural environment. Addressing this gap, Li and his colleagues have developed a dual-CRISPR system capable of deleting thousands of NCREs systematically and genome-wide without additional barcoding. The system targets various UCEs and enhancers, exploring their biological functions and impact on cell survival and drug response in specific cells. The study identified many UCEs with silencer activities and enhancers with dual functions. Specifically, a UCE region named PAX6_Tarzan was found to play a crucial role in cardiomyocyte differentiation from human embryonic stem cells. The research provides a versatile tool to explore the function of NCREs and other non-coding genome parts, potentially contributing to our understanding of human diseases and developing new drug targets. Below, I've shared my concerns and specific critiques of the study.

Minor:

1. The study introduces a dual-CRISPR system wherein two RNA polymerase III promoters, U6 and H1, facilitate the transcription of two CRISPR guide RNAs as depicted in Supplementary Fig. 1a. The author asserts that having two distinct promoter sequences minimizes the likelihood of recombination events in plasmids or infected cells. However, the supporting evidence for this claim remains undisclosed.
2. Figure 1b demonstrates impaired cell growth resulting from the knockout of the UCEs for Foxp1, PBX3, PAX6, and de_novo_1. However, the manuscript does not elucidate whether this impairment is attributable to cell cycle inhibition or apoptosis, a clarification crucial for interpreting the experimental results.
3. Figures 1h, 2h, and 2i show that the knockout of UCE activates some genes within the same TADs. However, there is no mention of the impact on genes in neighboring TADs. Given the similarity to experiments in Supplementary Figure 6b, a comprehensive analysis is warranted.
4. In Figure 3, dual-CRISPR-2.0 is utilized to assess the roles of enhancers in K562 cells, revealing 1,005 enhancers linked to cell growth. Subsequent Luciferase assays identified four out of six regions as potential enhancers. Despite this, the term "silencers" describes two of these regions, where "repressors" might convey a more accurate depiction of the phenomena observed.

Major:

1. Supplementary Figure 2a presents three k562 clones, yet it remains unclear which clone was chosen for subsequent experiments. Additionally, the potential impact of Cas9 expression on the overall screening efficiency needs to be addressed.
2. Three methods are employed to address off-target effects of gRNAs in Supplementary Figure 3. However, since the UCE for Foxp1, PBX3, and PAX6 are closed regions that DHS and histone markers can't detect in Supplementary Figures 3a and 3c, the efficiency of the CRISPR in these regions remains unexplained. To rigorously rule out off-target effects, especially for the regions depicted in Supplementary Figure 4, incorporating genome-wide off-target detection methods like DISCOVER-Seq+ is recommended.
3. Supplementary Figure 4 raises the intriguing observation that traditional histone markers are absent in these UCE regions. How about the other UCEs detected in this manuscript? Queries also remain regarding the interaction of these UCEs with neighboring genes and regulatory elements, with a suggestion for more accurate methods like 4C-seq to validate interaction findings.
4. Despite UCEs being identified as lacking cell-type-specific features, the manuscript notes that most NCREs function in a tissue-specific manner. How does this happen? Although partially addressed this point by comparing 293T and K562 cells in Supplementary Figure 5, is it because the regions that could be deleted in 293T cells can't be deleted in K562 cells? Since 10% of them could be detected in both compartments, what is the difference between these 10% UCE and the rest of UCE?
5. In Fig. 1h and Supplementary Fig. 6a, observations indicate the upregulation of PTPRD and RCN1 genes in specific knockout clones. Queries arise on the potential interaction between these UCEs and gene promoters and whether CRISPRa targeting de_novo_1 and PAX6_Ta could also activate PTPRD and RCN1 genes?
6. Figure 2 showcases the pivotal role of UCE in drug resistance, focusing solely on K562 cells. Expanding the study to incorporate additional cell lines could unearth shared UCEs, potentially serving as attractive drug targets for clinical studies.
7. In Figure 4d, the choice of gRNAs targeting GFP as controls is questioned because deleting DNA fragments could induce DNA damage responses. Moreover, the potential impact of deleting one or two enhancers on the activities of the remaining enhancers and the feasibility of using CRISPRi to recapitulate cell growth inhibition effects merit consideration.
8. Figure 5 endeavors to expand dual-CRISPR application to study NCRE functions at the single-cell level, identifying 22 significant NCRE-gene pairs. However, questions persist regarding the efficiency of this system compared to Perturb-seq and its capability to discern cell-type-specific NCRE-gene pairs.
9. Figure 6a shows a KEGG pathway enrichment analysis for genes differentially regulated in PAX6_Ta KO cells within +/-5Mbp around PAX6_Ta. The study of genes beyond this range and their association with heart-related pathways remains unexplored. Additionally, the choice of FDR < 0.25 and its significance are queried, with a recommendation for at least three biological repeats for Figures 6e and 6f.

Reviewer #2 (Report for the authors (Required)):

Non-coding regulatory elements (NCREs) including non-coding RNAs, promoters, enhancers, silencers and insulators have their unique roles in cellular physiology, which were increasingly studied in the past decades. However, their characterizations and functions in regulating gene expressions are far from complete dissected. This manuscript by Li et al. developed a straightforward dual-CRISPR screening system capable of deleting thousands of NCREs genome-wide to study their functions in cell growth and drug response under distinct biological contexts.

Basically, this study is timely and novel, hopefully revealing an amount of unappreciated NCREs, with the help of the advancing CRISPR/Cas9 technology. In terms of this, they discovered and examined several NCREs, such as PBX3_CI, FOXP1_FI, PAX6_Ta, QKI-Jo, ZNF503_Op, as well as an intergenic NCRE, de_novo_1, which is excited. The discovered NCREs were experimentally examined to play roles in cell proliferation or drug resistance. Furthermore, by performing several assays such as Dual-Luciferase assays, their potential functions acting as enhancers or silencers were assessed. They also figured out that PAX6_Ta, by acting as a silencer, regulated the cardiomyocytes differentiation from hPSCs.

While putting these in the first place is due to the development of dual-CRISPR system for high-throughput analysis, which extends from the technology of dual-sgRNA deletion previously used in the authors' lab. Nonetheless, this manuscript has put the field of NCREs towards a new milestone. However, despite the successful application of this dual-CRISPR system, several concerns, especially in experimental designs, may dampen the integrity or confidence of this study.

1. In this manuscript, dual-sgRNAs were designed to delete a fragment of genomic DNAs containing NCREs or their core

regions. Also, the authors established several KO cell lines that were used in Figure 1, 2 and 6, which are homozygously deleted. It appears that both sgRNAs were high efficient in DNA cleavages. Yet, it also raised the concern that how exactly are the on-target efficiencies of designed sgRNAs in the library? It is quite important, considering that there are also great probabilities that only single sgRNAs work or work separately, failing to delete a desired DNA fragment that might dwarf the NCRE coverage of this system.

2.Line 251-253: How did the authors choose the genes in the same TAD for qPCR analysis (Figure 1h; Figure 2h)? What is the selection criteria (distance from NCREs)? For example, to analyze the genes with de_novo_1, why the authors selected PTPRD, but not KDM4C?

3.Line 272-276: The authors concluded that NCRE de_novo_1 may impair cell growth by regulating the PTPRD gene. However, to clarify this, it is suggested to downregulate PTPRD by CRISPRi in de_novo_1 KO cells, but not WT cells, which can demonstrate that PTPRD downregulation may rescue the effects of de_novo_1 KO.

4.Line 300-303: The authors identified one SNP rs571942374 altered the silencer activity and may render the patients bearing this SNP less favorably to imatinib treatment. However, the effect of rs571942374 is actually not quite clear, with only a minor significance (Figure 2g). Other assays may be required to confirm this.

5.Line 533-542: The authors figured out that PAX6_Ta KO impaired the cardiomyocyte differentiation, and detected that RCN1 was significantly upregulated, which was also seen in K562 cells. The concern is: (a) what is the function of RCN1? (2) Did RCN1 play a role in cardiomyocyte differentiation? The authors should explain and discuss in term of these. In supplementary figure 10g, we can see a dramatic increase of PAX6, with a minor increase of RCN1. However, PAX6 is a typical neurepithelium marker. How would PAX6 upregulation, or put it further, an NCRE in a PAX6 locus, modulates the cardiomyocyte formation. The authors may clarify it.

6.In the Discussion section, the authors described the strength of the dual-CRISPR system, however, the limitations of the system (or this study) shall also be listed.

Other minor points:

- Figure 2d: the y axis shall be 'Cell viability', if I am not wrong.
- Line 513: please clarify what is the super-core promoter, and what is exactly the selected validated silencer.
- Line 635: Shall be pBP43, but not Dual-CRISPR-Cas9-U6-H1?
- Line 641: the abbreviation of ultraconserved elements (UCEs) has already been listed in Line 114.

Reviewer #3 (Report for the authors (Required)):

In this paper, Li, Tan, Akkari-Henić et al. present a dual-CRISPR screening system and test it on K562 and 293T cell lines. They focus on the identification of essential non-coding regulatory elements, including Ultra-Conserved Elements (UCEs) and previously characterized/annotated regions from VISTA and ENCODE. Intriguing aspects of the study include the identification of silencers and functional NCRE clusters. The adaptability of the proposed system for single-cell readout is also noteworthy. While the paper is interesting, it has several weak points, such as potential confounders related to data quality, proposed validation strategies, and a lack of comparison with previous studies that have probed the function of enhancers using different CRISPR-based systems.

Major points

1. There are several instances where specific thresholds were used to process the data based on different computational tools. The rationale for setting these thresholds is unclear, as is the impact of changing them on the study's conclusions. For example:

- "Differential expression tests and p-value calculations were performed by the MAST-fitted model based on whole transcriptome data in all single cells except for genes with mean normalized expression < 0.0165. "
- "... top50OfftargetTotalScore < 47 and top100OfftargetTotalScore < 50"

2. "No raw data of the CRISPR screenings have been deposited. These raw data will be provided upon request." Given the widespread reproducibility crisis, it is paramount that the authors deposit all the data for all the screens on GEO. They can use an embargo date and password-protected access if there are concerns about sharing data pre-publication.

3. "To test this plasmid system, two guide RNAs targeting the 5 and 3 prime ends of one DNA fragment in the genome were inserted. After transfecting the cells, two functional guide RNAs were expressed and able to successfully delete the targeted region from the human genome " How general is the efficacy of this system? How many regions were tested? Given the intrinsic variability of editing, proving that the system works on a single DNA fragment may not be sufficient.

4. "...all potential single guide RNAs that target both ends of each NCRE were designed. " It is not clear what boundaries were considered in designing all the possible sgRNAs.

5. “..two distinct promoter sequences should reduce the chance of potential recombination events in plasmids or infected cells.” How was this formally tested? What was the average recombination rate?

6. “The replicates correlated well with each other, indicating the screen system is stable and reliable “ Please report the correlation value in the main text. Additionally, upon examining Supplementary Figure 2b, it appears that the correlation is quite poor. Furthermore, "14 days rep1" seems to correlate better with "control rep1" rather than with "14 days rep2," suggesting potential batch effects or other technical issues.

7. “The cell growth phenotype was not due to the off-target effects of the individual paired guide RNAs, as editing using a single guide from the pair did not result in cell growth defects (Supplementary Fig. 3a).”
Supplementary Figure 3a shows an increase in proliferation for both individual guides, partially contradicting the authors' interpretation that these guides should not affect proliferation. It is not clear how the authors exclude off-target effects based on this experiment. This discrepancy should be explicitly reported and discussed.

8. “ However, CRISPRi targeting did not show any cell growth defects (Supplementary Fig. 3c), which may result from effector-range differences between CRISPRa and CRISPRi technologies.

“This dual CRISPR screening system has multiple advantages over existing similar systems, and we expect that this system will have broad applications in studying the functions of NCREs and other non-coding parts of the genome.”

The authors should first validate that CRISPRi works as expected on previously characterized enhancers in K562 by performing qPCR of genes that have already been associated with perturbation of these enhancers by CRISPRi. Excellent candidates are the MYC enhancers, which have been previously associated with cell proliferation by Fulco et al (<https://pubmed.ncbi.nlm.nih.gov/27708057/>). This validation/comparison is crucial to ensure the system is working as expected before making claims about advantages of the new proposed system. In addition is important to discuss and compare also with additional screens in K562: <https://www.ncbi.nlm.nih.gov/pmc/articles/PMC6886585/> and <https://pubmed.ncbi.nlm.nih.gov/34326544/>.

9. “It has been challenging to dissect the roles of enhancers in their endogenous loci in a comprehensive manner, and therefore no related genome-wide systematic study has been done yet. ”

This statement is incorrect. Previous studies have used CRISPRi to study enhancers at scale (~6K regions), such as the one found here: <https://www.ncbi.nlm.nih.gov/pmc/articles/PMC6690346>. It is crucial to compare the results presented in this paper with those from prior studies. A formal comparison is also needed for the section "Studying the functions of NCREs using the dual-CRISPR system at the single-cell level." From the data presented, it is not clear that the system described in this paper is superior to that of Gasperini et al. It would be beneficial to test the same validated elements in this study with the dual-CRISPR system to assess the sensitivity and specificity of the proposed assay.

10. When making statements about significance, it is important to the p-value, and the statistical test used in parentheses. For example, “RCN1 gene was significantly upregulated (p=?, test?) ”

11. What are the sequence features of the discovered silencers? It would be interesting to scan these regions for potential transcription factor (TF) binding sites for repressors.

12. “While constructing these libraries, we further optimized the cloning procedures and final structure of the dual CRISPR library system (Fig. 3b), which we named dual-CRISPR-2.0. The main improvement is that the distance between the two scaffolds was increased to 200 bp for optimal NGS sequencing efficiency “

From the text, it is not clear what the problem was with the previous system or how the new system is an improvement. It is important to discuss this and present some data to support this point.

13. “To test how these essential enhancers regulate cell growth, we further analyzed the transcription factor binding enrichment in all the essential enhancer regions and identified ZNF263, PATZ1 and KLF4 among the top enriched motifs”

It may be worth assessing the expression levels of these proposed factors. For example, KLF4 doesn't seem to be highly expressed in K562 cells, so other transcription factors from the same family, such as Klf1, which is highly expressed, may be the actual factors involved.

14. “The NCREs whose loss confers Imatinib resistance were identified as regions that were positively selected (i.e., NCREs with beta scores > mean + 2x.s.d.) in the 15-day imatinib-treated populations but are weakly selected in the 15-day culture populations.” It is also important to consider and report the p-values associated with the proposed threshold. Additionally, how many of these candidate regions are statistically significant?

Minor

1. Line 182: "After filtering low coverage" could be better phrased as "After filtering for low coverage."

2. Line 274 - "de_nono_1" -> de_novo_1

Sun 03 Mar 2024

Decision on Article NBME-23-2310B

Dear Prof Pang,

Thank you for your revised manuscript, "The genome-wide dual-CRISPR screening identifies essential non-coding regulatory elements". Having consulted with the original reviewers (whose comments you will find at the end of this message), I am pleased to write that we shall be happy to publish the manuscript in *Nature Biomedical Engineering*, provided that the points specified in the attached instructions file are addressed.

When you are ready to submit the final version of your manuscript, please upload the files specified in the instructions file.

We encourage authors to take up transparent peer review. If you are eligible and opt in to transparent peer review, we will publish, as a single supplementary file, all the reviewer comments for all the versions of the manuscript, your rebuttal letters, and the editorial decision letters. **If you opt in to transparent peer review, in the attached file please tick the box 'I wish to participate in transparent peer review'; if you prefer not to, please tick 'I do NOT wish to participate in transparent peer review'**. In the interest of confidentiality, we allow redactions to the rebuttal letters and to the reviewer comments. If you are concerned about the release of confidential data, please indicate what specific information you would like to have removed; we cannot incorporate redactions for any other reasons. More information on transparent peer review is available.

Best wishes,

Pep

Pep Pàmies
Chief Editor, Nature Biomedical Engineering

P.S. Nature Portfolio journals encourage authors to share their step-by-step experimental protocols on a protocol-sharing platform of their choice. Nature Portfolio's Protocol Exchange is a free-to-use and open resource for protocols; protocols deposited in Protocol Exchange are citable and can be linked from the published article. More details can be found at www.nature.com/protocolexchange/about.

Reviewer #1 (Report for the authors (Required)):

The researchers have devised a streamlined dual-CRISPR screening approach that allows genome-wide deletion of thousands of non-coding regulatory elements (NCREs) to explore their roles in various biological settings within mammalian cells. This research indicates that many NCREs are functionally significant, impacting human health and disease, and could potentially be targets for drug development. The dual CRISPR system offers several improvements over existing methods and is anticipated to be widely applicable in the functional study of NCREs and other non-genomic coding regions. Notably, the findings also suggest that NCREs may be pivotal in drug resistance, with identifying an essential ultra-conserved element (UCE) involved in cardiomyocyte differentiation presenting opportunities for clinical translation. All my questions have been thoroughly addressed; I endorse the paper and believe it will meet the high scientific standards of Nature Biomedical Engineering.

Reviewer #2 (Report for the authors (Required)):

The authors have addressed my concerns. Hope this developed system will have broad applications in studying the functions of NCREs and other non-coding parts of the genome.

Reviewer #3 (Report for the authors (Required)):

The authors have sufficiently addressed all the concerns raised.

Nature Biomedical Engineering is a Transformative Journal. Authors may publish their research with us through the traditional subscription access route, or make their paper immediately open access through payment of an article-processing charge. More information about publication options is available.

You may need to take specific actions to comply with funder and institutional open-access mandates. If the work described in the accepted manuscript is supported by a funder that requires immediate open access (as outlined, for example, by Plan S) and your manuscript was originally submitted on or after January 1st 2021, then you will need to select the gold OA route. Authors selecting subscription publication will need to accept our standard licensing terms (including our self-archiving policies), and these will supersede any other terms that the author or any third party may assert apply to any version of the manuscript.

Rebuttal 1

Reviewer #1 (Report for the authors (Required)):

NCREs, including UCEs, play pivotal roles in various biological pathways and can be genomic medicine targets. However, existing methods do not measure the functions of NCREs in their natural environment. Addressing this gap, Li and his colleagues have developed a dual-CRISPR system capable of deleting thousands of NCREs systematically and genome-wide without additional barcoding. The system targets various UCEs and enhancers, exploring their biological functions and impact on cell survival and drug response in specific cells. The study identified many UCEs with silencer activities and enhancers with dual functions. Specifically, a UCE region named PAX6_Tarzan was found to play a crucial role in cardiomyocyte differentiation from human embryonic stem cells. The research provides a versatile tool to explore the function of NCREs and other non-coding genome parts, potentially contributing to our understanding of human diseases and developing new drug targets. Below, I've shared my concerns and specific critiques of the study.

We appreciate that the referee finds our research interesting. As you pointed out, the dual-CRISPR system is versatile. The design and construction of the dual-CRISPR libraries are straightforward. The screening and downstream analysis pipelines are compatible with other CRISPR screenings targeting genes using single RNAs, which makes it easier for research groups with CRISPR screening experiences to use. We hope to provide a useful and complementary tool next to the existing ones, such as CRISPRi systems, to advance our understanding of the non-coding human genome. Based on your suggestions, we performed additional extensive experiments and analyses, such as dual-CRISPR editing efficiency testing, clonal CRISPR off-target detection, Discover-Seq+, cell growth assays on about 30 newly generated CRISPR clones with proper biological replicates, phenotypic assays of dual-CRISPR clones, and other computational analyses. The data and related conclusions are listed in more detail below. These results support that the dual-CRISPR system is efficient and reliable for studying NCREs in their endogenous environment. Based on your constructive suggestions, the revised manuscript is substantially improved, and we hope you are also convinced that the dual-CRISPR system is a useful tool to serve the broad research community in studying the functions of NCREs.

Minor:

1. The study introduces a dual-CRISPR system wherein two RNA polymerase III promoters, U6 and H1, facilitate the transcription of two CRISPR guide RNAs as depicted in Supplementary Fig. 1a. The author asserts that having two distinct promoter sequences minimizes the likelihood of recombination events in plasmids or infected cells. However, the supporting evidence for this claim remains undisclosed.

We thank the referee for pointing this out. The statement you referred to was based on the literature suggesting reverse transcriptase-mediated recombination can lead to the uncoupling of molecular tags and barcodes, depending on the length between the homologous sequence, for instance, up to 28% for the distance of 720bp[1]. A similar

system with two identical human U6 promoters was also made during the initial design of the dual-CRISPR system. However, no PCR products could be made when amplifying the dual-crRNA sequences with the same primer-targeting U6 sequence, and no subsequent dual-CRISPR libraries could be made. Therefore, we could not provide direct data supporting this statement, so we have removed it from the manuscript.

Instead, we try to state that direct sequencing of the two guide RNAs would remove other potential biases that have been reported. For instance, the swapping of sequences due to homologous recombination during lentivirus packaging, which could be up to ~50%[1-3]; or due to cloning, which could be ~20%[3].

We revised the related statement in the revised manuscript as follows: “In addition, potential recombination bias from PCR, cloning, and template switching in pooled lentiviral production can be filtered out after sequencing” in lines 163-164.

2. Figure 1b demonstrates impaired cell growth resulting from the knockout of the UCEs for Foxp1, PBX3, PAX6, and de_novo_1. However, the manuscript does not elucidate whether this impairment is attributable to cell cycle inhibition or apoptosis, a clarification crucial for interpreting the experimental results.

Thank the referee for this suggestion. We performed both cell cycle and apoptosis analyses on the NCRE KO clones, and observed a slight but not significant increase in G1 cells among all these NCRE KO clones and a mild increase in apoptotic cells within the PAX6_Ta and de_novo_1 KO clones, the latter of which was significant (Supplementary Figure 7). In addition, the “negative regulation of cell cycle G1/S phase transition” pathway was also enriched with an adjusted p-value of 0.04 in the PAX6_Ta clone from gene ontology analysis implemented by the GSEA method method (Supplementary Table 10). All these differences were often mild compared to the K562 cells treated with Imatinib, which served as positive controls, indicating that these NCRE KO clones were not in major crisis but had growth disadvantages.

We have added these data as a new Supplementary Figure 7 with description in the revised manuscript in lines 281-285.

3. Figures 1h, 2h, and 2i show that the knockout of UCE activates some genes within the same TADs. However, there is no mention of the impact on genes in neighboring TADs. Given the similarity to experiments in Supplementary Figure 6b, a comprehensive analysis is warranted.

Thanks for pointing this out. In the figures, we did not make it very clear that the genes we have tested are from different TADs. We have added additional bars with respective colors to indicate the different TADs. In general, we have tested most of the genes from the different TADs. The genes we did not show are usually with very low expression levels, which could not be detected reliably by qPCR experiments.

4. In Figure 3, dual-CRISPR-2.0 is utilized to assess the roles of enhancers in K562 cells,

revealing 1,005 enhancers linked to cell growth. Subsequent Luciferase assays identified four out of six regions as potential enhancers. Despite this, the term “silencers” describes two of these regions, where “repressors” might convey a more accurate depiction of the phenomena observed.

We thank the referee for this suggestion. We also included “repressor” to describe these regions in the revised manuscript.

Major:

1. Supplementary Figure 2a presents three k562 clones, yet it remains unclear which clone was chosen for subsequent experiments. Additionally, the potential impact of Cas9 expression on the overall screening efficiency needs to be addressed.

Thank the referee for pointing this out. As indicated in the legend, clone #10 was selected for the subsequent screening. The rationale was that this clone has the highest expression level, which we did not explain and has now been added to the figure legend. We routinely test the expression level of Cas9 of this clone before and after the screening or before starting new rounds of screening, and we did not observe the loss of Cas9 expression. Therefore, the potential impact of Cas9 expression on the overall screening efficiency would be minimal.

2. Three methods are employed to address off-target effects of gRNAs in Supplementary Figure 3. However, since the UCE for Foxp1, PBX3, and PAX6 are closed regions that DHS and histone markers can't detect in Supplementary Figures 3a and 3c, the efficiency of the CRISPR in these regions remains unexplained. To rigorously rule out off-target effects, especially for the regions depicted in Supplementary Figure 4, incorporating genome-wide off-target detection methods like DISCOVER-Seq+ is recommended.

The referee pointed out important issues related to CRISPR editing, including both on-target and off-target effects. We checked the editing efficiency of these respective UCEs in K562 cells and observed, on average, 60%-80% deletion of the respective UCEs in bulk cells (Figure 1 for the referee).

To rule out off-target effects in the clones we tested, we performed Sanger sequencing on the possible off-target regions of the two guide RNAs targeting the respective UCEs. As these are individual clones, potential off-target editing will be present in all the cells from the same clone and will be detected directly by Sanger sequencing. We did not observe off-target editing from these clones (Supplementary Table 11). Therefore, these data indicate that the phenotype we observed from these clones was not due to the potential off-target effects of the individual guide RNAs.

Your suggestion to use DISCOVER-Seq+ to recover all potential off-target regions may indeed provide more insights into the off-target effects. We thought this method might provide an overall estimation of the off-target effects of the screening libraries. DISCOVER-Seq and DISCOVER-Seq+ were only tested to detect the off-target regions

from one guide RNA during transient transfection, so that it would provide detectable signals at both on-target and off-target sites from the whole cell population used for ChIP-seq experiments. As no experiments have been done to detect the overall off-target sites from a CRISPR library using DISCOVER-Seq+, we were excited to test our UCE dual-CRISPR screening library. We transiently transfected the UCE dual-CRISPR library into K562 cells expressing Cas9. As the library contains many different guide RNAs, and to increase the chance of recovering signals from each individual guide RNA, we transfected 80 ug of the dual-CRISPR library into 40×10^6 cells by electroporation and used all these cells for MRE11 ChIP-seq experiments, based on the protocol of DISCOVER-Seq+. However, we were not able to detect any significant on-target or off-target sites. When we compared our control sample with the published one from DISCOVER-Seq+ that would measure some baseline DNA damage events, we observed a good correlation, indicating the ChIP-seq experiments were successful (Figure 2 for the referee). We reasoned that transiently transfecting too many different guide RNAs into cells would lead to the chopping of the chromosomes, stress response, and cell death that were observed before, therefore obscuring the DNA damage response that could be measured by MRE11 ChIP-seq.

3. Supplementary Figure 4 raises the intriguing observation that traditional histone markers are absent in these UCE regions. How about the other UCEs detected in this manuscript? Queries also remain regarding the interaction of these UCEs with neighboring genes and regulatory elements, with a suggestion for more accurate methods like 4C-seq to validate interaction findings.

As the referee suggested, we compared the possible epigenetic markers of essential UCEs identified from K562 cells with the rest of the UCEs, and found H3K27me3 histone modification and accessible chromatin regions identified by ATAC-seq were more enriched in the essential UCEs. This observation is also consistent with our observation that some of the essential UCEs are silencers (Figure 1g), which might be in the repressed chromatin regions with repressor transcription factors binding (accessible chromatin regions within the repressed chromatin regions)[4]. Thank you for this constructive suggestion. We have added this observation to Supplementary Figure 4c and updated it in the revised manuscript.

As also recommended by the referee, we used virtual 4C to check possible 3D interactions within these regions. Similar to the observation based on CTCF ChIA-PET, we did not observe very clear distal interactions, except for some interactions close to PBX3_claudia UCE (Figure 3 for the referee).

4. Despite UCEs being identified as lacking cell-type-specific features, the manuscript notes that most NCREs function in a tissue-specific manner. How does this happen? Although partially addressed this point by comparing 293T and K562 cells in Supplementary Figure 5, is it because the regions that could be deleted in 293T cells can't be deleted in K562 cells? Since 10% of them could be detected in both compartments, what is the difference between these 10% UCE and the rest of UCE?

We thank the referee for pointing these out. It has been observed that most of the non-coding regulatory regions function in a tissue-specific manner, which regulates the limited number of around 20,000 genes in time and space to form more complex tissues. As many UCEs function as regulatory elements, it is reasonable that they may also function in a tissue-specific manner, as we observed from our screenings. However, how exactly NCREs function in a tissue-specific manner is still an exciting research question. It is possible that both the epigenetic environment of the NCREs and the abundance of transcription factors change during tissue differentiation, which results in tissue-specific regulation. We hope to contribute more knowledge related to your question in the future.

To answer your concerns that the unequal dual-CRISPR editing efficiency at the same regions of different cell types led to the differences, we transfected the same dual-CRISPRs targeting the respective regions of K562 cells into the 293T cells. We observed similar deletion efficiency between the two cell lines. Therefore, the tissue-specificity of UCEs we observed should reflect the real biology rather than the result of technical issues (Figure 1b for the referee).

We performed motif analysis to see if there are any differences between the 10% shared UCEs and the rest. As there are only 26 regions within the 10% shared UCEs, we could not identify meaningful motif enrichment. However, there are different sets of motifs enriched in the rest of UCE from K562 and 293T cells, for instance, ZNF187 and KLF12 in K562-specific essential UCEs while VENTX, C11orf9 and FOXO1 only enriched in 293T-specific essential UCEs, which probably reflects the tissue-specific expression or usage of different transcription factors. These motifs have been updated in the Supplementary Figure 5b.

5. In Fig. 1h and Supplementary Fig. 6a, observations indicate the upregulation of PTPRD and RCN1 genes in specific knockout clones. Queries arise on the potential interaction between these UCEs and gene promoters and whether CRISPRa targeting de_novo_1 and PAX6_Ta could also activate PTPRD and RCN1 genes?

Thank the referee for pointing this out. As indicated in Supplementary Figure 4 (and Figure 3 for the referee), no direct interactions between the UCEs and their respective genes were observed. But at the same time, after performing the experiments suggested by you, we observed that CRISPRa targeting de_novo_1 and PAX6_Ta could also activate PTPRD and RCN1 genes as shown in Figure 4 for the referee. Such regulations may via some other machinery, which merits future characterization.

6. Figure 2 showcases the pivotal role of UCE in drug resistance, focusing solely on K562 cells. Expanding the study to incorporate additional cell lines could unearth shared UCEs, potentially serving as attractive drug targets for clinical studies.

We are excited that the referee shared the same view in this respect. Indeed, we are currently expanding our research into the lines of research you suggested and have observed some encouraging early results, which still require more time for thorough

exploration. In the current manuscript, we want to show the potential of UCEs as markers for drug response or possible future drug targets, serving as a proof-of-concept.

7. In Figure 4d, the choice of gRNAs targeting GFP as controls is questioned because deleting DNA fragments could induce DNA damage responses. Moreover, the potential impact of deleting one or two enhancers on the activities of the remaining enhancers and the feasibility of using CRISPRi to recapitulate cell growth inhibition effects merit consideration.

Thanks for your construction suggestions. To show that deleting DNA fragments would not, in general, lead to a growth disadvantage, we deleted two control UCEs. As shown in Figure 5a for the referee, no growth disadvantages were observed, suggesting DNA damage response was not the reason for the phenotype.

As you suggested, we tested the CRISPRi to target individual enhancers or combined enhancers. As shown in Figure 5b-g for the referee, we did not observe a clear phenotype, which could be the same reason as we explained in Supplementary Figure 3c. As a positive control to show the CRISPRi system works as expected, we also tested a published CRISPRi targeting the Myc enhancer, where a significant reduction of Myc gene expression and cell growth reduction were observed (Figure 5h for the referee) [5].

In addition, following your suggestion, we tested the potential impact of deleting one or two enhancers on the activities of the remaining enhancers. We measured the respective enhancer eRNA expression by qPCR, which is one readout of enhancer activity[6]. We observed that deleting one enhancer with an enhancer cluster did not alter the activity of the nearby enhancers or sometimes even upregulated the activity of the nearby enhancers. This observation also corroborates the observation that deleting one enhancer from the cluster did not lead to a growth disadvantage, as shown in Figure 4d; as the nearby enhancers may take over the regulation. However, when two enhancers were deleted, activity from all the enhancers within the same cluster dropped, possibly due to the collapses of the transcriptional regulation complex at the enhancer cluster. These data suggest a very interesting hypothesis that individual enhancers within an enhancer cluster may contribute to the stability of the transcriptional regulation complex. We have included these data in Supplementary Figure 10g, h and i, and updated the interpretation and experiment details in the revised manuscript in lines 446-449.

8. Figure 5 endeavors to expand dual-CRISPR application to study NCRE functions at the single-cell level, identifying 22 significant NCRE-gene pairs. However, questions persist regarding the efficiency of this system compared to Perturb-seq and its capability to discern cell-type-specific NCRE-gene pairs.

Thank you for pointing this out. We aimed to present proof-of-concept experiments to show that it is possible to expand the dual-CRISPR system into the single-cell system. As suggested, we tried to compare with the most relevant dataset generated in K562 cells by Gasperini et al, which we previously referenced in the manuscript[7]. However, the

target regions selected in our studies are largely different from those tested by Perturb-seq. Therefore, we are not able to compare these two systematically.

Nonetheless, we identified one enhancer that was jointly assayed by our dual-CRISPR system and the Perturb-seq system (Figure 6a, b for the Referee). Deleting this enhancer via the dual-CRISPR led to the down-regulation of the target gene FAM83A by around 11.3%, whereas Gasperini et al achieved 35% and 14.7% reduction of FAM83A expression by two different CRISPRi/guide RNAs respectively. This suggests that the dual-CRISPR system may be as good as the Perturb-seq in identifying cell-type-specific NCRE-gene pairs, and could serve as an alternative method compared to the currently available CRISPR-based single-cell systems.

9. Figure 6a shows a KEGG pathway enrichment analysis for genes differentially regulated in PAX6_Ta KO cells within +/-5Mbp around PAX6_Ta. The study of genes beyond this range and their association with heart-related pathways remains unexplored. Additionally, the choice of FDR < 0.25 and its significance are queried, with a recommendation for at least three biological repeats for Figures 6e and 6f.

We thank the referee for noticing this. We performed both pathway analyses, as the referee suggested, and we mislabeled Figure 6a in the initial submission. Figure 6a actually showed the pathway analysis of all the genes differentially regulated in the PAX6_Ta cells. We have corrected this error in the revised manuscript, and sorry for this. The choice of FDR < 0.25 was recommended by GSEA to list significantly deregulated pathways[8]. Several heart-related pathways in this figure have the adjusted p-values smaller than 0.05. However, we agree that it is confusing to mention FDR < 0.25 in this figure legend and have moved this to the Methods part, where we could elaborate in more detail.

Thank you for pointing out that additional biological repeats are needed. These two specific experiments were done during COVID, when access to the lab was very limited in the Netherlands, especially in our medical center, where protecting weak patients was the priority. These ESC experiments required several weeks of continuous experiments. At the same time, many special reagents related to ESC experiments were back ordered for months. That is why these related figures lacked the additional replicates. We have repeated these experiments during the revision and provided updated figures with the proper number of biological replicates and statistical analyses for Figures 6e, 6f, and Supplementary Figures 12f and 12g.

Reviewer #2 (Report for the authors (Required)):

Non-coding regulatory elements (NCREs) including non-coding RNAs, promoters, enhancers, silencers and insulators have their unique roles in cellular physiology, which were increasingly studied in the past decades. However, their characterizations and functions in regulating gene expressions are far from complete dissected. This manuscript by Li et al. developed a straightforward dual-CRISPR screening system capable of deleting thousands of NCREs genome-wide to study their functions in cell growth and drug response under distinct biological contexts.

Basically, this study is timely and novel, hopefully revealing an amount of unappreciated NCREs, with the help of the advancing CRISPR/Cas9 technology. In terms of this, they discovered and examined several NCREs, such as PBX3_CI, FOXP1_FI, PAX6-Ta, QKI-Jo, ZNF503_Op, as well as an intergenic NCRE, de_novo_1, which is excited. The discovered NCREs were experimentally examined to play roles in cell proliferation or drug resistance. Furthermore, by performing several assays such as Dual-Luciferase assays, their potential functions acting as enhancers or silencers were assessed. They also figured out that PAX6-Ta, by acting as a silencer, regulated the cardiomyocytes differentiation from hPSCs.

While putting these in the first place is due to the development of dual-CRISPR system for high-throughput analysis, which extends from the technology of dual-sgRNA deletion previously used in the authors' lab. Nonetheless, this manuscript has put the field of NCREs towards a new milestone. However, despite the successful application of this dual-CRISPR system, several concerns, especially in experimental designs, may dampen the integrity or confidence of this study.

It is a great pleasure that the referee finds the dual-CRISPR system timely and novel. We share the same excitement in that the dual-CRISPR system may serve as a versatile tool to study different types of NCREs, such as non-coding RNAs, promoters, enhancers, silencers, insulators, or other complex elements. As this system is similar to commonly used CRISPR systems targeting genes, it may be picked up easily by research groups familiar with CRISPR screenings in terms of design, construction, application, and analysis pipeline. Thank you for your suggestions, which we addressed in detail below. In general, we clarified how we maximize the deletion efficiency and study and interpret the biology of the top hits properly. Most importantly, based on your concerns, we also discussed the system's limitations. The experiments performed and clarifications made according to your comments further support the efficacy of the dual-CRISPR system and substantially improve the manuscript. We hope you will find the revised manuscript providing confident research that may benefit a broader research community.

1. In this manuscript, dual-sgRNAs were designed to delete a fragment of genomic DNAs containing NCREs or their core regions. Also, the authors established several KO cell lines that were used in Figure 1, 2 and 6, which are homozygously deleted. It appears that both sgRNAs were high efficient in DNA cleavages. Yet, it also raised the concern that how exactly are the on-target efficiencies of designed sgRNAs in the library? It is

quite important, considering that there are also great probabilities that only single sgRNAs work or work separately, failing to delete a desired DNA fragment that might dwarf the NCRE coverage of this system.

We thank the referee for raising this point. In fact, during the initial design of the dual-CRISPR system, we also shared the same concerns as yours. In the design of the dual-CRISPR libraries targeting the potential enhancers in K562 cells, we applied a stringent filter only to consider guide RNAs with high editing efficiency. In addition, for each enhancer region, we also designed 14-25 pairs of guide RNAs to maximize the chance to have functional pairs for deletion; compared to that, for typical single-guide CRISPR screening libraries, there are usually 5-10 guides per gene. We have clarified this in the revised manuscript and also discussed the limitations as you suggested.

2.Line 251-253: How did the authors choose the genes in the same TAD for qPCR analysis (Figure 1h; Figure 2h)? What is the selection criteria (distance from NCREs)? For example, to analyze the genes with de_novo_1, why the authors selected PTPRD, but not KDM4C?

We thank the referee for pointing this out, which we should make clearer. As mentioned at the start of the paragraph: “NCREs may regulate proximal and distal genes, especially genes that are within the same topologically associating domains (TAD)”. Therefore, we tested genes based on whether or not they are present in the same TADs. We have revised the statement to “Transcription of PTPRD and RCN1 genes, which are within the same TAD of the tested separate NCRE, were significantly upregulated in the knockout clones of de_novo_1 and PAX6_Ta respectively”. Furthermore, we have added separate color bars, matching the respective TADs below the qPCR data to make the presentation clear in the revised figures. The reason the PTPRD gene was selected is that this gene was significantly upregulated upon de_novo_1 knockout. KDM4C did not change, which suggests it might not be directly regulated by the de_novo_1 region. These two genes are within the same TAD of the de_novo_1 region.

3.Line 272-276: The authors concluded that NCRE de_novo_1 may impair cell growth by regulating the PTPRD gene. However, to clarify this, it is suggested to downregulate PTPRD by CRISPRi in de_novo_1 KO cells, but not WT cells, which can demonstrate that PTPRD downregulation may rescue the effects of de_novo_1 KO.

We thank the referee for this constructive suggestion, which would strengthen our conclusion. As you suggested, we used CRISPR to knock out the PTPRD gene in the de_novo_1 KO cells (We tested CRISPRi but did not get good downregulation). The PTPRD gene knockout within the de_novo_1 KO cells rescued the cell growth phenotype of the original de_novo_1 KO clone significantly, although not to the same level as the WT cells. It is possible that multiple CRISPR editing made the cells unable to recover fully to the initial WT cell growth phenotype. We have included these data in the Supplementary Figure 6c and 6d in lines 293-295 in the revised manuscript.

4.Line 300-303: The authors identified one SNP rs571942374 altered the silencer activity

and may render the patients bearing this SNP less favorably to imatinib treatment. However, the effect of rs571942374 is actually not quite clear, with only a minor significance (Figure 2g). Other assays may be required to confirm this.

We thank the referee for the suggestion. Indeed, the effect of rs571942374 was not that clear. During the revision, we performed additional luciferase assays to check if such a difference was consistent, and we did still see only this SNP showed a significant effect (with 5 replicate experiments in total). As we use these data to suggest the possibility that SNPs may affect imatinib response via altering the NCRE activity, we have moved this dataset to the revised Supplementary Figure 8.

5.Line 533-542: The authors figured out that PAX6_Ta KO impaired the cardiomyocyte differentiation, and detected that RCN1 was significantly upregulated, which was also seen in K562 cells. The concern is: (a) what is the function of RCN1? (2) Did RCN1 play a role in cardiomyocyte differentiation? The authors should explain and discuss in term of these. In supplementary figure 10g, we can see a dramatic increase of PAX6, with a minor increase of RCN1. However, PAX6 is a typical neurepithelium marker. How would PAX6 upregulation, or put it further, an NCRE in a PAX6 locus, modulates the cardiomyocyte formation. The authors may clarify it.

We thank the referee for pointing out these, which we could have explained better in the manuscript. RCN1 is a CREC family member calcium-binding protein[9]. As calcium homeostasis is important in cardiomyocytes[10], deregulation of RCN1 may affect cardiomyocyte differentiation. As you pointed out, PAX6 is a transcription factor that is critical for neuronal differentiation[11], which should not be expressed during cardiomyocyte differentiation. It is possible that the PAX6_Ta region is supposed to repress the expression of these genes, at least during cardiomyocyte differentiation. The loss of PAX6_Ta caused the deregulation of these genes during cardiomyocyte differentiation. Therefore, they together may contribute to the defects we observed.

Based on your comments, we have included the possible explanation in the revised manuscript to clarify the relation of these genes with cardiomyocyte differentiation defects in lines 564-580.

We are sorry for the large variation of the PAX6 expression data. These specific experiments were done during COVID, when access to the lab was very limited in the Netherlands, especially in our medical center where protecting weak patients was the priority. ESC experiments usually require several weeks of continuous experiments. At the same time, many special reagents related to ESC experiments were back ordered for months. That is why these related figures lacked the additional replicate. Now, we have repeated these experiments during the revision and provided updated figures with the proper number of biological replicates and statistical analyses.

6.In the Discussion section, the authors described the strength of the dual-CRISPR system, however, the limitations of the system (or this study) shall also be listed.

Thank you for your suggestion. We have listed the possible limitations based on your comments in the revised manuscript in lines 634-643.

“However, similar to other CRISPR systems, the dual-CRISPR system also relies on high editing efficiency and low off-target effects of the guide RNAs, especially as the dual-CRISPR system requires two functional guide RNAs to work at the same time. Therefore it is recommended to design more distinct guide pairs, which would increase the chance that multiple guide pairs would delete the same region and also allow for more reliable statistical analyses. Furthermore, recombination among guide RNA pairs is associated with the systematic problems of cloning and virus packaging. These wrongly paired reads need to be filtered before the final analyses.”

Other minor points:

- Figure 2d: the y axis shall be ‘Cell viability’, if I am not wrong.

Thank you for your suggestion. It has been corrected to “Cell viability”.

- Line 513: please clarify what is the super-core promoter, and what is exactly the selected validated silencer.

Sorry for the confusion here. The super-core promoter is a published strong promoter that contains four core promoter motifs—the TATA box, initiator (Inr), motif ten element (MTE) and downstream promoter element (DPE)—in a single promoter, and is distinctly stronger than the CMV and AdML promoters[12]. The validated silencer was from the previous publication[4]. We have included the respective references in the revised manuscript.

- Line 635: Shall be pBP43, but not Dual-CRISPR-Cas9-U6-H1?

Thank the referee for pointing this out and sorry for the confusion. In this context, we were referring to the steps to clone the dual-CRISPR plasmid containing Cas9 used to generate knockout clones for screening validations. We have revised this in the method to make it clear in lines 671-688.

- Line 641: the abbreviation of ultraconserved elements (UCEs) has already been listed in Line 114.

Thank you for pointing this out. We have corrected this in the revised manuscript.

Reviewer #3 (Report for the authors (Required)):

In this paper, Li, Tan, Akkari-Henić et al. present a dual-CRISPR screening system and test it on K562 and 293T cell lines. They focus on the identification of essential non-coding regulatory elements, including Ultra-Conserved Elements (UCEs) and previously characterized/annotated regions from VISTA and ENCODE. Intriguing aspects of the study include the identification of silencers and functional NCRE clusters. The adaptability of the proposed system for single-cell readout is also noteworthy. While the paper is interesting, it has several weak points, such as potential confounders related to data quality, proposed validation strategies, and a lack of comparison with previous studies that have probed the function of enhancers using different CRISPR-based systems.

We really appreciate that the referee finds our research intriguing. As you pointed out, the dual-CRISPR system allows the identification of functional silencers and NCRE clusters, but also potentially other functional NCREs such as insulators, etc. The versatility and adaptability of the dual-CRISPR system may provide the research community with a complementary tool to further elucidate the biology of the non-coding genome. As you suggested, we clarified the details related to the dual-CRISPR system, reported the statistics not only in the legends but also in the text, provided the raw data and evidence to show the efficiency, reliability, and advantage of the system, further looked into the potential biology related to the identify NCREs, and properly interpreted the observation. Although, due to the differences in the regions of interest, we were not able to systematically compare the efficacy of the single-cell dual-CRISPR system with the existing CRISPRi Pertube-seq systems, we did observe that the dual-CRISPR system would capture the same NCRE-gene pair, indicating the single-cell dual-CRISPR system could be a good alternative option in some applications. Based on your comments, we also discussed the possible limitations of the dual-CRISPR system.

In general, your suggestions and proposed experiments substantially strengthened the revised manuscript. We hope you are convinced the dual-CRISPR system is a good complementary tool comparing to the existing CRISPRi systems, and would benefit the research community in studying NCREs.

Major points

1. There are several instances where specific thresholds were used to process the data based on different computational tools. The rationale for setting these thresholds is unclear, as is the impact of changing them on the study's conclusions. For example:

- “Differential expression tests and p-value calculations were performed by the MAST-fitted model based on whole transcriptome data in all single cells except for genes with mean normalized expression < 0.0165. ”

- “... top5OfftargetTotalScore < 47 and top10OfftargetTotalScore < 50”

We thank the referee for pointing it out. We applied the cut-off of 0.01 to filter out genes with low mean normalized expression levels because these genes will bias the statistical modeling results[13]. These low-abundant genes exhibit larger technical variation even after normalization, which is not real biological variation and will cause the model to overfit the scRNA-seq data. We have updated these descriptions in the revised manuscript in lines 898-899.

The “top5OfftargetTotalScore” and “top10OfftargetTotalScore” are the scores defined in the CRISPRseek package, which was based on experimentally-derived off-target scoring schemes[14], to rank the off-target specificity of guide RNA design. The total scores of the top 5 and top 10 off-target regions are calculated, respectively, which are the most likely off-target sites to be cleaved in a targeting experiment. The lower score means lower predicted off-target editing potential. We have updated these descriptions in the revised manuscript in lines 701-706.

2. "No raw data of the CRISPR screenings have been deposited. These raw data will be provided upon request." Given the widespread reproducibility crisis, it is paramount that the authors deposit all the data for all the screens on GEO. They can use an embargo date and password-protected access if there are concerns about sharing data pre-publication.

We thank the referee for the suggestion. We have deposited our screening data on GEO. Pooled screen and scRNA-seq sequencing data are available at GEO under accession number GSE254241 (reviewer token: ifghywsiznoxvix). RNA-seq data are available at GEO under accession GSE247234 (reviewer token: ulileesqfrwhvet).

The code for processing the pooled dual-CRISPR screen data is available on GitHub: https://github.com/PangLab/DualCRISPR_pooled_screen_snakemake_pipeline.

We have updated these in the revised manuscript in lines 1004-1012.

3. “To test this plasmid system, two guide RNAs targeting the 5 and 3 prime ends of one DNA fragment in the genome were inserted. After transfecting the cells, two functional guide RNAs were expressed and able to successfully delete the targeted region from the human genome ” How general is the efficacy of this system? How many regions were tested? Given the intrinsic variability of editing, proving that the system works on a single DNA fragment may not be sufficient.

We thank the referee for pointing these out. During the design of the system, we tested more conditions but only used one example to make the data presentation more condensed. For instance, because two different promoters were used to drive the transcription of the respective guide RNAs, we also swapped the promoter and guide RNA combinations of the same plasmid. We observed that U6 and H1 promoter activity would lead to similar editing efficiency. We have included these data in the Supplementary Figure 1b, together with another region we tested. In theory, the efficiency of the system relies on the on-target editing activity of the designed single guide RNAs.

Therefore, during the design, efficient guide RNAs with low off-target effects were selected. Based on others' and our own previous experiences, introducing two guide RNAs with good activity into one cell would efficiently delete NCREs[4]. Furthermore, we design 14-25 pairs of guide RNAs to target each region to increase the likelihood that sufficient pairs of guide RNAs will delete the NCREs. During our validation, we also observed efficient editing with the dual-CRISPR system (Figure 1 for the referee).

As you may also appreciate, the most challenging part of the system design was often the testing and identifying the optimal conditions for cloning and recovering the screening libraries. The final optimal procedures were described in detail in the Methods.

4. "...all potential single guide RNAs that target both ends of each NCRE were designed." It is not clear what boundaries were considered in designing all the possible sgRNAs.

Thank you for pointing this out. The boundaries considered in designing all the possible sgRNAs varied depending on the regions of interest.

In the Methods part, we described "the immediate upstream and downstream flanking regions (200 bp in size) of the enhancers were selected for guide RNA design. Single gRNAs from the two flanking regions were paired, resulting in up to 25 pairs of gRNAs per targeting enhancer region (referred to as enhancer library 1 in Fig. 5a targeting 3,995 enhancers). Due to genomic repeats and other constraints, not all enhancers could be targeted by the previous design strategy. To target the rest of the predicted enhancers in K562 cells, pairs of guide RNAs were selected within the enhancer regions (5' proximal and 3' proximal of the enhancers), with around 14 pairs of guide RNAs per enhancer (referred as enhancer library 2 in Fig. 5b targeting 13,020 enhancers). In this way, all predicted enhancers in K562 cells by ENCODE were targeted".

We could also add this description to the main text if necessary.

5. "...two distinct promoter sequences should reduce the chance of potential recombination events in plasmids or infected cells." How was this formally tested? What was the average recombination rate?

We thank the referee for pointing out this, which was also shared by another referee. The statement you referred to was based on the literature suggesting reverse transcriptase-mediated recombination can lead to the uncoupling of molecular tags and barcodes, depending on the length between the homologous sequence, for instance, up to 28% for the distance of 720bp[1]. During the initial dual-CRISPR system design, a similar system was also made using two identical human U6 promoters. However, no PCR products could be made when amplifying the dual-crRNA sequences with the same primer-targeting sequence, and no subsequent dual-CRISPR libraries could be made. Therefore, we could not provide direct data supporting this statement, so we have removed it from the manuscript.

Instead, we try to state that direct sequencing of the two guide RNAs would remove other potential biases that have been reported. For instance, the swapping of sequences due to homologous recombination during lentivirus packaging, which could be up to ~50%[1-3]; or due to cloning, which could be ~20%[3]. After optimization, we were able to reduce the recombination rate to around 30%.

We revised the related statement in the revised manuscript as follows: "In addition, potential recombination bias from PCR, cloning, and template switching in pooled lentiviral production can be filtered out after sequencing" in lines 163-164.

6. "The replicates correlated well with each other, indicating the screen system is stable and reliable " Please report the correlation value in the main text. Additionally, upon examining Supplementary Figure 2b, it appears that the correlation is quite poor. Furthermore, "14 days rep1" seems to correlate better with "control rep1" rather than with "14 days rep2," suggesting potential batch effects or other technical issues.

We thank the referee for pointing this out. Following your suggestion, we further looked into this issue. Previously, we plotted the Pearson correlation coefficients in Supplementary Figure 2b, which might be inappropriate as the Pearson correlation was used to measure only linear relationships. To resolve this, we applied the Spearman correlation method to measure the correlation between biological replicates, which is used to evaluate the monotonic relationship. We also reported the Spearman correlation coefficients in the main text (control rep1 and rep2: 0.42, 14 days rep1 and rep2: 0.38) and adjusted the tone in interpreting these data, in lines 183-185.

Pooled genome-wide screenings, such as CRISPR gene knockout/activation screenings, are intrinsically noisy[15]. The dual-CRISPR screening is also similar. In the screenings we presented, we generated biological replicates starting from virus infection, which may also further the variation among samples. However, true biological hits would be identified using a proper statistical analysis package. We used the MAGeCK robust ranking algorithm for normalization and hits identification, which is tailored for CRISPR screenings and benchmarked and recommended[16]. Furthermore, multiple guide RNA pairs were designed to target each region, which also increased the power for detection. Most importantly, top hits were validated to confer the phenotype of interest.

7. "The cell growth phenotype was not due to the off-target effects of the individual paired guide RNAs, as editing using a single guide from the pair did not result in cell growth defects (Supplementary Fig. 3a)."

Supplementary Figure 3a shows an increase in proliferation for both individual guides, partially contradicting the authors' interpretation that these guides should not affect proliferation. It is not clear how the authors exclude off-target effects based on this experiment. This discrepancy should be explicitly reported and discussed.

Thank you for pointing this out. Indeed, the single guide RNA editing showed an increase in proliferation. As you suggested, we clearly described this phenotype and made proper discussion in the revised manuscript in lines 206-210.

“The cell growth phenotype was not due to the effects of the individual paired guide RNAs, as editing using a single guide from the pair did not result in cell growth defects but some growth advantage as shown from Foxp1_FI region (Supplementary Fig. 3a), which may be resulted from local editing effects of these single guides.”

8. “ However, CRISPRi targeting did not show any cell growth defects (Supplementary Fig. 3c), which may result from effector-range differences between CRISPRa and CRISPRi technologies.

“This dual CRISPR screening system has multiple advantages over existing similar systems, and we expect that this system will have broad applications in studying the functions of NCREs and other non-coding parts of the genome.”

The authors should first validate that CRISPRi works as expected on previously characterized enhancers in K562 by performing qPCR of genes that have already been associated with perturbation of these enhancers by CRISPRi. Excellent candidates are the MYC enhancers, which have been previously associated with cell proliferation by Fulco et al (<https://pubmed.ncbi.nlm.nih.gov/27708057/>). This validation/comparison is crucial to ensure the system is working as expected before making claims about advantages of the new proposed system. In addition is important to discuss and compare also with additional screens in K562: <https://www.ncbi.nlm.nih.gov/pmc/articles/PMC6886585/> and <https://pubmed.ncbi.nlm.nih.gov/34326544/>.

We thank the referee for this constructive suggestion. As you recommended, we used guide RNA targeting the MYC enhancer from the reference and performed additional experiments to test whether the CRISPRi system works in our hands. Expression of the MYC gene was significantly reduced, and cell growth slowed down (Figure 5h for the referee). These data indicate that the CRISPRi system works as expected.

And thank you for the suggestions on the two references. Compared to the dual-CRISPR system, using CRISPRi to tile the potential NCREs could pinpoint the exact functional units for gene regulation with higher base resolution. We think the dual-CRISPR system is complementary to these published systems, and discuss the advantage of the CRISPRi systems in the revised manuscript in lines 643-646.

“As the dual-CRISPR system aims to delete the entire NCREs, it is not able to pinpoint the exact functional units within the NCRE, compared to other complementary CRISPRi systems, which tile the defined regions using an array of single guide RNAs”.

9. “It has been challenging to dissect the roles of enhancers in their endogenous loci in a comprehensive manner, and therefore no related genome-wide systematic study has been done yet. ”

This statement is incorrect. Previous studies have used CRISPRi to study enhancers at scale (~6K regions), such as the one found here:

<https://www.ncbi.nlm.nih.gov/pmc/articles/PMC6690346>. It is crucial to compare the results presented in this paper with those from prior studies. A formal comparison is also needed for the section "Studying the functions of NCREs using the dual-CRISPR system at the single-cell level." From the data presented, it is not clear that the system described in this paper is superior to that of Gasperini et al. It would be beneficial to test the same validated elements in this study with the dual-CRISPR system to assess the sensitivity and specificity of the proposed assay.

Thank you for pointing this out, and we should have made this statement properly. Indeed, the reference from Gasperini et al provides an enhancer study at scale, though the enhancers were pre-selected to be associated with a subset of genes. To put the statement correctly, we have adjusted the related statement to: "There has not been a comprehensive study to examine all the potential enhancers within a defined cell line", in lines 339-341.

And thank you for the constructive suggestion to compare the single-cell dual-CRISPR system with the dataset of Gasperini et al, which was also raised by another referee. We aimed to present proof-of-concept experiments to show that it is possible to expand the dual-CRISPR system into the single-cell system. As suggested, we tried to compare with the dataset generated in K562 cells by Gasperini et al, which we previously referenced in the manuscript[7]. However, the target regions selected in our single-cell studies are largely different from those tested by Perturb-seq. Therefore, we are not able to compare these two systematically.

Nonetheless, we identified one enhancer that was jointly assayed by our dual-CRISPR system and the Perturb-seq system (Figure 6 for the Referee). Deleting this enhancer via the dual-CRISPR led to the down-regulation of the target gene FAM83A by around 11.3%, whereas Gasperini et al achieved 35% and 14.7% reduction of FAM83A expression by two different CRISPRi/guide RNAs respectively. This suggests that the dual-CRISPR system may be as good as the Perturb-seq in identifying cell-type-specific NCRE-gene pairs, and could serve as an alternative method compared to the currently available CRISPR-based single-cell systems.

10. When making statements about significance, it is important to the p-value, and the statistical test used in parentheses. For example, "RCN1 gene was significantly upregulated (p=?, test?)"

Exact number.

We thank the referee for pointing this out. We previously provided all the statistical details in the figure legends and methods. Now we have also added the statistical values in the text when appropriate in the revised manuscript.

For this specific case for RCN1 gene (adjusted p value=8.39e-18, the differential expression test was calculated by Wald test and p value was adjusted by Benjamini and Hochberg (BH) approach implemented by DESeq2)

11. What are the sequence features of the discovered silencers? It would be interesting to scan these regions for potential transcription factor (TF) binding sites for repressors.

We thank the referee for the suggestion. We scanned the 7 silencers that we verified by luciferase assays against 340 available ENCODE transcriptional factor binding sites from ChIP-seq experiments generated in K562 cells. (<https://genome.ucsc.edu/cgi-bin/hgTrackUi?db=hg19&q=encTfChipPk>).

Only 3 silencers (QKI_Jo, E8:124178 and E14:71791) overlapped with at least one TF binding site. For QKI_jo, two TFs POLR2A and HMBOX1 were found to bind the region. The HMBOX1 is a telomere binding protein, with an indication also to be a transcription repressor[17, 18]. For E8:124178 and E14:71791, there were many more TF binding sites, among which CREM, EGR1, EP300, GABPB1, HDAC1, MAX, MEIS2, NFIC and ZBTB7A are shared between the two silencers/repressors. Based on the literature, CREM, EGR1, MAX, NFIC and ZBTB7A were reported to function as transcriptional repressors[19-23] or HDAC1 reported as co-repressor[24].

12. “While constructing these libraries, we further optimized the cloning procedures and final structure of the dual CRISPR library system (Fig. 3b), which we named dual-CRISPR-2.0. The main improvement is that the distance between the two scaffolds was increased to 200 bp for optimal NGS sequencing efficiency “

From the text, it is not clear what the problem was with the previous system or how the new system is an improvement. It is important to discuss this and present some data to support this point.

We thank the referee for pointing this out, which we did not explain well. During the sequencing of the dual-CRISPR-2.0, both via Sanger sequencing and Next-generation sequencing, we observed that sometimes the quality of the reads dropped near the end of the guide RNA scaffold part (Figure 7 for the referee, starting around 45nt). We reasoned that the two close-by scaffold sequences may form some secondary structures during sequencing, which leads to the sequencing quality drop. Although this would not affect the overall analysis, as the 20bp crRNA sequences and the first part of the scaffold sequences are the readouts and not affected by this, we still would like to eliminate these unwanted effects. Therefore, we tested different insertion lengths between the two scaffolds and identified that 200 bp-insertion between the two scaffolds is ideal to achieve good reading qualities whilst still maintaining the optimal size of next-generation sequencing libraries for most Illumina sequencers (200bp to 800bp).

13. “To test how these essential enhancers regulate cell growth, we further analyzed the transcription factor binding enrichment in all the essential enhancer regions and identified ZNF263, PATZ1 and KLF4 among the top enriched motifs” It may be worth assessing the expression levels of these proposed factors. For example, KLF4 doesn't seem to be highly expressed in K562 cells, so other transcription factors from the same family, such as Klf1, which is highly expressed, may be the actual factors involved.

We thank the referee for the suggestion. We checked the expression levels of all these genes, including ZNF263, PATZ1, KLF4 and other KLF family members such as Klf1 (Figure 8 for the referee). In Figure 8a for the referee, the y axis represents the percentage of genes with respective transcripts, and x axis represents the log transformed TPM normalized transcripts count. As expected, a large percentage of the genes have very low or no expression in K562 cells (the left bar), whilst ZNF263, PATZ1 and KLF4 genes have reasonable expression levels. Indeed, as you pointed out, Klf1 has the highest expression among the Klf families as shown in Figure 8c for the referee, and these TFs share very similar motifs as shown in Figure 8b for the referee. Therefore it is possible that the more abundant TF family members could also be the potential regulators. We have updated this statement in the revised manuscript in lines 377-381.

“It is possible that these transcription factors or their close family members that share similar binding motifs are responsible for the function of these NCREs.”

14. “The NCREs whose loss confers Imatinib resistance were identified as regions that were positively selected (i.e., NCREs with beta scores $> \text{mean} + 2 \times \text{s.d.}$) in the 15-day imatinib-treated populations but are weakly selected in the 15-day culture populations.” It is also important to consider and report the p-values associated with the proposed threshold. Additionally, how many of these candidate regions are statistically significant?

We thank the referee for pointing it out. The analysis was performed using MAGeCK MLE, where the beta score was calculated using the Expectation-Maximization (EM) algorithm. This package is designed to analyze hits that confer drug resistance via pooled CRISPR screenings, and the beta scores describe how regions are selected: a positive beta score indicates positive selection, and a negative beta score indicates negative selection. All colored top center hits were considered significant hits that confer imatinib resistance, which is above the cut-off of normalized beta scores and recommended by the MAGeCK MLE. As the applied threshold was the means $\pm 2\text{SD}$ which defines the 95% confidence interval, all the highlighted regions should be statistically significant with P values < 0.05 . There are 81 regions in this group, which we have reported previously in Supplementary Table 3.

Minor

1. Line 182: "After filtering low coverage" could be better phrased as "After filtering for low coverage."

Thank you for this suggestion. We have corrected this accordingly.

2. Line 274 - “de_nono_1 ” -> de_novo_1

Thank you for pointing this out, and we're sorry for the errors. We have corrected these typos in the revised manuscript.

Figure 1: Efficiency of dual CRISPR-mediated genetic deletion.

(a,b) Quantification of PCR gel products shows deletion efficiency of dual CRISPR system targeting PBX3_Ci, FOXP1_Fi, PAX6_Ta and de_novo_1 regions in K562 cells (a) and 293T cells (b). The left panels show DNA gels of the cells transiently transfected with the respective dual-CRISPRs, with blue and red arrowheads indicating intact and knockout regions, respectively. The bar plot on the right represents the ratio between intact and knockout bands ($n=1$).

Figure 2: Correlation with published DISCOVER-Seq+ MRE11 ChIP-seq data.

(a) Pearson correlation of normalized MRE11 ChIP-seq read intensity in MRE11 peaks regions between the published dataset (Zou, Roger S., *et al.* 2023) with our dataset generated during this revision. **(b)** Venn diagram of MRE11 peaks overlapping with published MRE11 ChIP-seq in K562 (Zou, Roger S., *et al.* 2023) and our dataset generated during this revision.

Figure 3: Virtual 4C analysis of chromatin interactions around essential UCE/NCREs.

(a-d) Virtual 4C analysis anchored at 4 essential UCE/NCREs identified: PBX3_CI (a), FOXP1_FI (b), PAX6_Ta (c) and de_novo_1 (d). Only at PBX3_CI region, there were more additional peaks towards the nearby regions indicated by blue box.

Figure 4: qPCR analysis of genes in K562 cells with CRISPRa targeting the de_novo_1 and PAX6_Ta regions.

(a) The transcription of PTPRD gene in K562 cells with CRISPRa targeting the de_novo_1 region (de_novo_1 SAM) was upregulated, compared to the cells with CRISPRa non-targeting guide RNA (NTG) as measured by qPCR. ($n=3$ biological independent samples; values represent the mean value \pm s.d.; ** $P < 0.01$ calculated using two-tailed unpaired t-test). **(b)** The transcription of RCN1 gene in K562 cells with CRISPRa targeting the PAX6_Ta region (PAX6_Ta SAM) was upregulated, compared to the cells with CRISPRa non-targeting guide RNA (NTG) as measured by qPCR. ($n=3$ biological independent samples; values represent mean value \pm s.d.; *** $P < 0.001$ calculated using two-tailed unpaired t-test).

Figure 5: Cell viability of dual-CRISPR-editing and CRISPRi-editing cells.

(a) Cell growth of non-essential NCREs dual-CRISPR knockout (KO). No difference in growth was observed in ESRRG_Eros_KO and ESRRG_Isaac_KO cells. Left panel: cell proliferation assay was performed by mixing the KO cell lines with cells expressing GFP at a 1:1 ratio. The changes in GFP percentage were monitored at indicated time points by FACS. Cells with dual-CRISPR guide RNAs targeting GFP sequences served as negative controls (Ctrl_KO). The y axis represents the relative ratio of the GFP negative cells to the positive cells. The ratio of cells in the initial mixture was set as 100% ($n=3$ biological independent samples; values are shown as the mean \pm s.d.; $^{ns}P > 0.05$ calculated using two-way ANOVA). Middle and right panel: K562 cells were transfected with the respective dual-CRISPRs that target the indicated NCREs. The blue arrowhead indicates the intact genomic regions. The red arrow head indicates the NCRE deletions. **(b,c,d,e,f,g)** Cell growth effect of using CRISPRi to target either single (b,c,d) or multiple enhancers (e,f,g) within the respective enhancer clusters on chromosomes chr 6, chr 10 and chr 22. One guide RNA was used for each individual enhancer. Cell proliferation assay was performed by mixing cells expressing GFP with cells with CRISPRi targeting a single enhancer or multiple enhancers within the same enhancer cluster. The changes in GFP percentage were monitored at indicated time points by FACS. The y axis represents the relative ratio of the GFP negative cells to the positive cells, when compared to the cells with CRISPRi non-targeting guide RNA (NTG). The ratio of cells in the initial mixture was set as 100% ($n=3$ biological independent samples; values are shown as the mean \pm s.d.; for all CRISPRi editing cells $^{ns}P > 0.05$ calculated using two-way ANOVA). **(h)** Left panel: cell growth effect of CRISPRi targeting the proximal enhancer (e2) regulating *MYC* gene. K562 cell was transfected with guide RNA targeting e2 as reported in Fulco *et al.* 2016 (Myc-e2). Cell proliferation assay was performed as described above. ($n=3$ biological independent samples; values are shown as the mean \pm s.d.; Myc-e2 $^{**}P < 0.01$ calculated using two-way ANOVA). Right panel: The transcription of *MYC* gene in K562 cells with CRISPRi targeting e2 was downregulated, compared to the cells with CRISPRi non-targeting guide RNA (NTG) ($n=3$ biological independent samples; values represent mean value \pm s.d.; Myc-e2 $^{***}P < 0.001$ calculated using two-tailed unpaired t-test).

Figure 6: E8:124178 was detected to regulate *FAM83A* gene by both single-cell dual-CRISPR screening and CRISPRi-based perturb-seq.

(a) Transcriptional changes of genes surrounding candidate enhancer on chromosome 8 (E8:124178) measured by scRNA-seq upon E8:124178 deletion by dual-CRISPR. The y axis shows the logarithm fold change of the captured genes between single cells with the dual-CRISPR targeting E8:124178 and control K562 single cells. The x axis indicates the genomic coordinate. Dashed line indicates the location of E8:124178. Distinct colors represent different TADs. **(b)** Violin plot compared the normalized expression of *FAM83A* gene between single cells with dual-CRISPR targeting E8:124178 (Perturb) and control K562 single cells (NCT).

Figure 7: Distribution of base sequencing quality.

Base sequencing quality of dual-CRISPR 1.0 (ultraconserved element library and K562 enhancer library 1 respectively) and dual-CRISPR 2.0 (K562 enhancer library 2) are shown. The x axis represents the base location, the y axis represents the base quality, where higher scores indicate better base quality. The background of the graph divides the y-axis into very good quality calls (green), calls of reasonable quality (orange), and calls of poor quality (red). The central red line is the median value, the yellow box represents the interquartile range (25-75%), the upper and lower whiskers represent the 10% and 90% points and the blue line represents the mean quality. The structure of read was indicated in bar below: overhang part of the U6 promoter (blue), crRNA sequence (yellow) and tracrRNA sequence (red). Dashed line indicates the starting location of crRNA and tracrRNA part.

Figure 8: K562 essential enhancer-related TF gene expression level.

(a) The distribution of normalized gene expression in K562 cells based on RNA-seq data from Cancer Cell Line Encyclopedia Project (CCLE). The x axis represents the gene expression level normalized by logarithm transformed TPM reads count and the y axis represents the gene density. The expression levels of the 3 TF genes were indicated. **(b)** The TF motifs of KLF1 and KLF4 are shown (JASPAR database release 2024). **(c)** The expression levels of all KLF family genes in K562 retrieved from CCLE RNA-seq data. The x axis represents the genes and the y axis represents the gene expression level normalized by logarithm transformed TPM reads count.

References

1. Sack, L.M., et al., *Sources of Error in Mammalian Genetic Screens*. G3 (Bethesda), 2016. **6**(9): p. 2781-90.
2. Hill, A.J., et al., *On the design of CRISPR-based single-cell molecular screens*. Nature Methods, 2018. **15**(4): p. 271-274.
3. Xie, S., et al., *Frequent sgRNA-barcode recombination in single-cell perturbation assays*. PLoS One, 2018. **13**(6): p. e0198635.
4. Pang, B. and M.P. Snyder, *Systematic identification of silencers in human cells*. Nature Genetics, 2020. **52**(3): p. 254-263.
5. Fulco, C.P., et al., *Systematic mapping of functional enhancer-promoter connections with CRISPR interference*. Science, 2016.
6. Oh, S., et al., *Enhancer release and retargeting activates disease-susceptibility genes*. Nature, 2021. **595**(7869): p. 735-740.
7. Gasperini, M., et al., *A Genome-wide Framework for Mapping Gene Regulation via Cellular Genetic Screens*. Cell, 2019. **176**(1-2): p. 377-390.e19.
8. Subramanian, A., et al., *Gene set enrichment analysis: a knowledge-based approach for interpreting genome-wide expression profiles*. Proc Natl Acad Sci U S A, 2005. **102**.
9. Ozawa, M. and T. Muramatsu, *Reticulocalbin, a novel endoplasmic reticulum resident Ca(2+)-binding protein with multiple EF-hand motifs and a carboxyl-terminal HDEL sequence*. J Biol Chem, 1993. **268**(1): p. 699-705.
10. Barry, W.H. and J.H. Bridge, *Intracellular calcium homeostasis in cardiac myocytes*. Circulation, 1993. **87**(6): p. 1806-15.
11. Thakurela, S., et al., *Mapping gene regulatory circuitry of Pax6 during neurogenesis*. Cell Discovery, 2016. **2**(1): p. 15045.
12. Juven-Gershon, T., S. Cheng, and J.T. Kadonaga, *Rational design of a super core promoter that enhances gene expression*. Nature Methods, 2006. **3**(11): p. 917-922.
13. Hafemeister, C. and R. Satija, *Normalization and variance stabilization of single-cell RNA-seq data using regularized negative binomial regression*. Genome Biol, 2019. **20**(1): p. 296.
14. Hsu, P.D., et al., *DNA targeting specificity of RNA-guided Cas9 nucleases*. Nature Biotechnology, 2013. **31**(9): p. 827-832.
15. Joung, J., et al., *Genome-scale CRISPR-Cas9 knockout and transcriptional activation screening*. Nat. Protocols, 2017. **12**(4): p. 828-863.
16. Bodapati, S., et al., *A benchmark of algorithms for the analysis of pooled CRISPR screens*. Genome Biology, 2020. **21**(1): p. 62.
17. Chen, S., et al., *Isolation and functional analysis of human HMBOX1, a homeobox containing protein with transcriptional repressor activity*. Cytogenet Genome Res, 2006. **114**(2): p. 131-6.
18. Wu, L., et al., *HMBOX1, homeobox transcription factor, negatively regulates interferon- γ production in natural killer cells*. Int Immunopharmacol, 2011. **11**(11): p. 1895-900.
19. Stehle, J.H., et al., *Adrenergic signals direct rhythmic expression of transcriptional repressor CREM in the pineal gland*. Nature, 1993. **365**(6444): p. 314-20.
20. Feng, Y., et al., *EGR1 Functions as a Potent Repressor of MEF2 Transcriptional Activity*. PLoS One, 2015. **10**(5): p. e0127641.

21. Maeda, I., et al., *Max is a repressor of germ cell-related gene expression in mouse embryonic stem cells*. Nat Commun, 2013. **4**: p. 1754.
22. Liu, Y., H.U. Bernard, and D. Apt, *NFI-B3, a novel transcriptional repressor of the nuclear factor I family, is generated by alternative RNA processing*. J Biol Chem, 1997. **272**(16): p. 10739-45.
23. Liu, X.S., et al., *ZBTB7A acts as a tumor suppressor through the transcriptional repression of glycolysis*. Genes Dev, 2014. **28**(17): p. 1917-28.
24. Doetzlhofer, A., et al., *Histone deacetylase 1 can repress transcription by binding to Sp1*. Mol Cell Biol, 1999. **19**(8): p. 5504-11.